**#REVISED VERSION OF MANUSCRIPT HESSD-2018-400**

# Risks of seasonal extreme rainfall events in Bangladesh under 1.5 and 2.0 degrees' warmer worlds – How anthropogenic aerosols change the story

Ruksana H. Rimi[1], Karsten Haustein[1], Emily J. Barbour[1,2], Sarah N. Sparrow[3], Sihan Li[1,3], David C.H. Wallom[3], and Myles R. Allen[1].

[1]Environmental Change Institute, School of Geography and the Environment, University of Oxford, Oxford, OX1 3QY, UK.
[2]Commonwealth Scientific and Industrial Research Organisation, Land and Water, Canberra, ACT 2601, Australia.
[3]Oxford e-Research Centre, Department of Engineering Science, University of Oxford, Oxford, OX1 3QG, UK.

*Correspondence to:* Ruksana H. Rimi (ruksana.rimi@ouce.ox.ac.uk)

**Abstract.** Anthropogenic climate change is likely to increase risk (probability of occurrence of a hazard) of extreme weather events in the future. Previous studies have robustly shown how and where climate change has already changed the risks of weather extremes. However, developing countries have been somewhat underrepresented in these studies, despite high vulnerability and limited capacities to adapt. How additional global warming would affect the future risks of extreme rainfall events in Bangladesh needs to be addressed to limit adverse impacts. Our study focuses on understanding and quantifying the relative risks of extreme rainfall events in Bangladesh under the Paris Agreement temperature goals of 1.5°C and 2°C warming above pre-industrial levels. In particular, we investigate the influence of anthropogenic aerosols on these risks given their likely future reduction and resulting amplification of global warming. Using large ensemble regional climate model simulations from weather@home under different forcing scenarios, we compare the risks of rainfall events under pre-industrial (natural), current (actual), 1.5°C, and 2.0°C warmer and greenhouse gas (GHG)-only (with pre-industrial levels of anthropogenic aerosols) conditions. Both GHGs and anthropogenic aerosols influence changes in seasonal mean rainfall over this region. For extreme rainfall events, we find that the risk of a 1 in 100 year rainfall event has already increased significantly compared with pre-industrial levels across parts of Bangladesh, with additional increases likely for 1.5 and 2.0 degree warming (of up to 5.5 times higher, with an uncertainty range of 3.5 to 7.8 times). Climate change impacts on the probabilities of extreme rainfall events are found during both pre-monsoon and monsoon seasons, but the level of impacts are spatially variable across the country. Results also show that reduction in anthropogenic aerosols plays an important role in determining the overall future climate change impacts; by exacerbating the effects of GHG induced global warming and thereby increasing the rainfall intensity. We highlight that the net aerosol effect varies from region to region within Bangladesh, which leads to different outcomes of aerosol reduction on extreme rainfall statistics, and must therefore be considered in future risk assessments. Whilst there is a substantial reduction in the impacts resulting from 1.5°C compared with 2°C warming, the difference is spatially and temporally variable, specifically with respect to seasonal extreme rainfall events.

## 1 Introduction

One of the major goals of the 2015 Paris Agreement of the United Nations Framework Convention on Climate Change (UNFCCC), on "Holding the increase in the global average temperature to well below 2°C above pre-industrial levels and to pursue efforts to limit the temperature increase to 1.5°C" (UNFCCC, 2015), needs strong support from research on the nature, benefits and feasibility of this challenging goal. This Agreement calls for the quantification and comparison between the impacts of 1.5°C versus 2.0°C warmer global temperatures on different climate related aspects such as extreme weather events. While assessing both risks and vulnerabilities to incremental increases in global mean temperature, the discrimination of the

impacts of different radiative forcing contributions as well as the quantification of spatially varying changes in risk are crucially important. For example, highly unusual heat extremes that are virtually absent in the present climate in South Asia, would affect around 15% of land area of this region under 1.5°C and around 20% of land area under 2°C warming (The World Bank, 2012). The increase in heavy monsoon rainfall intensity for South Asia is projected to be 7% under 1.5°C and 10% under 2°C warming compared to pre-industrial conditions (Schleussner et al., 2016). Populations of this region largely depend on the stability of the monsoon, which provides water resources for agricultural production (The World Bank, 2012). It is projected that the years with above-normal monsoon rainfall will be more frequent (Endo et al., 2013; Kripalani et al., 2007). The seasonality of rainfall will be amplified with more rainfall during the wet season (Fung et al., 2011; Turner and Annamalai, 2012). The number of extreme rainfall events are projected to increase as well (Endo et al., 2012; Kumar et al., 2011 in Vinke et al., 2017). As a consequence of additional global warming, parts of East Asia and India are likely to have more frequent daily extreme rainfall events in monsoon season (Chevuturi et al., 2018). Here we assess whether these generalized projections are also valid using a large ensemble regional climate model framework focusing on Bangladesh.

Bangladesh is potentially a hotspot of climate change impacts as it is vulnerable to a combination of increasing challenges from record-breaking temperatures, extreme rainfall events, more intense river floods, tropical cyclones, and rising sea levels (The World Bank, 2012). Bangladesh has a tropical monsoon climate, flat and low-lying topography, and unique geographical location in the Ganges-Brahmaputra-Meghna Basin (Banglapedia, 2012; Rawlani and Sovacool, 2011). For these features, heavy rainfall events in the pre-monsoon (during Mar-Apr-May; MAM) and monsoon (during Jun-Jul-Aug-Sep; JJA) seasons are associated with a high risk of flooding and landslides almost every year. The frequencies of observed high-intensity rainfall events are increasing in the recent years (Murshed et al., 2011). For example, in 2017, heavy rainfall across the upstream Meghalaya hills in India and in Bangladesh caused pre-monsoon floods in March in the northeastern parts of the country. Consequently, vast areas of Haors (local name for lowland wetlands) and low-lying areas were inundated and most of the nearly-harvestable 'Boro' paddy crop (a local high yielding variety of paddy) was damaged (Nirapad, 2017). In June 2017, at southeastern parts of Bangladesh heavy rainfall caused devastating floods and multiple landslides killing at least 156 people (Paul and Hussain, 2017). National Aeronautics and Space Administration (NASA)'s near-real time Integrated Multi-satellitE Retrievals for Global Precipitation Measurement, GPM (IMERG) data estimated the heaviest rainfall accumulation of more than 510 mm in only 3 days (12-14 June 2017 (Gutro, 2017).

Considering the unfolding change in risk of heavy rain in the region under present-day conditions how would a 1.5°C and a 2.0°C warmer world change the probability of extreme rainfall events in Bangladesh? If climate change is already playing a role, then similar events are likely to occur even more frequently as global warming continues in the future (Faust, 2017). Reliable information regarding the relative changes in future risks of extreme rainfall events can help local decision makers to address the problem, develop appropriate adaptation strategies and allocate resources to minimize loss and damage associated with potential climate extremes. According to global climate model (GCM) ensemble based study, by 2090, the north-western part of Bangladesh would experience ~9% and ~18% increase in the pre-monsoon and monsoon mean rainfall respectively (Kumar et al., 2014). Caesar et al., (2015) used the high resolution (25 km) regional climate model (RCM), HadRM3P that is nested in the global HadCM3 model and projected a large increase in the very heavy daily rainfall events (>99th percentile, i.e., >23.8mm/day) and a decrease in the light-moderate rainfall events (<75th percentile, i.e., <12.3mm/day) during monsoon season (Jun-Sep) over Bangladesh by 2099. According to PRECIS (Providing REgional Climates for Impact Studies) model projection for 2080, the north-eastern Bangladesh would experience 0.42–75% more pre-monsoon rainfall compared to the baseline of 1971–2000 (Nowreen et al., 2015). While previous studies projected future changes in the seasonal mean or extreme rainfall events over a specific part or whole Bangladesh; none had the benefit of using very large model ensembles of high resolution RCM to examine exceptionally rare extreme rainfall events (e.g., events with 100 year of return period); explained whether or not anthropogenic

climate change played a role in changing the probabilities of those projected future rainfall events; and explored how anthropogenic aerosols changed the overall climate change impacts on rainfall events. The Fifth Phase Coupled Model Inter-comparison Project (CMIP5) models produced a broad range of temperature projections as a function of model sensitivity (van Vuuren et al., 2011). CMIP-style experiments are not ideal to provide with impact assessments specifically at 1.5°C or 2°C warming, because in these experiments, uncertainty increases with time and is dominated by responses and variability. While the UNFCCC in particular asking about the comparative risks associated with 1.5°C and 2°C warming, irrespective of what emission path is followed to achieve it. The Half a degree Additional warming, Prognosis and Projected Impacts (HAPPI) framework has been developed to address this call and provide with impact assessments specifically targeted for 1.5°C and 2.0°C warming (Mitchell et al., 2016, 2017). Hence the Half a degree additional warming, prognosis and projected impacts (HAPPI) framework has been developed, specifically targeted for 1.5°C and 2.0°C warming (Mitchell et al., 2016). There are only a few studies using CMIP5 (e.g., Fahad et al., 2017), or PRECIS ( e.g., Nowreen et al., 2015) simulations that investigated future changes in rainfall events over Bangladesh, but none of these have specifically addressed the warming targets of the Paris Agreement. The novelty of this study lies in meeting all these aforementioned challenges.

We considered anthropogenic aerosols in addition to greenhouse gases (GHGs) as a potential contributing factor in changing the risks of extreme rainfall events. Because aerosols can influence regional climate and change the risks of rainfall events by radiative forcing (Guo et al., 2013; Li et al., 2016). Furthermore, extreme rainfall events have higher sensitivity to aerosols removals, per degree of surface warming, in particular over the major aerosol emission regions like Asia (Samset et al., 2018). Therefore it is important to explore aerosol impacts while assessing the changes in the risks of extreme rainfall events under additional global warming scenarios of 1.5 and 2.0 degrees' of Paris Agreement.

Drawing on the large ensemble of regional climate model (RCM) runs generated with the weather@home system (Guillod et al., 2017; Massey et al., 2015) within the HAPPI experimental framework, we quantify changing rainfall risks for Bangladesh during MAM and JJAS. The risk of extreme rainfall events is evaluated for a counterfactual 'natural', current 'actual' and future 1.5 and 2.0 degrees warmer climate scenarios. The impact of anthropogenic aerosol emissions is quantified and discussed based on the GHG-only scenario.

We first introduce data and methods in Section 2, whilst a summary of model performance is presented in Section 3.1. We then assess percentage changes and standardized changes in the seasonal mean rainfall within five forcing scenarios (Natural, Actual, 1.5°C, 2.0°C and GHG-only) in Section 3.2. Finally, in Section 3.3 we detect the relative shifts in the probabilities of MAM and JJAS daily (and 5-day) rainfall extremes between the different forcing scenarios. The results are discussed in context of regional vulnerabilities and observed changes in Section 4.

## 2 Data and Methods

### 2.1 Observational data

The daily observational data sets that are used as a comparison against model results include: (i) Asian Precipitation Highly Resolved Observational Data Integration Towards Evaluation of Water Resources (APHRODITE) (Yatagai et al., 2012) and (ii) NOAA's Climate Prediction Center (CPC) global 0.5° analysis (Chen et al., 2008a). APHRODITE is a high-resolution daily gridded rainfall dataset for Asia (V1901, available for 1998-2015); created primarily with data obtained from a rain-gauge-observation network. CPC global daily rainfall dataset (available from 1979 to present) is constructed through a unified analysis of gauge-based daily rainfall over global land (Chen et al., 2008b). Basic facts about these two observational data sets are

presented in Table S1 in the supplementary information. Both model and observation data is re-gridded using bi-linear interpolation method to have similar and comparable grid structures.

## 2.2 Model setup and experimental design

The weather@home is part of the climateprediction.net programme (Stainforth et al., 2005) and is able to generate very large ensembles of climate model simulations by harnessing spare CPU time on a network of volunteers' personal computers (Allen, 1999; Stott et al., 2004; Massey et al., 2015). For this study, we use the high resolution (50 km) RCM, HadRM3P (over South Asia region) that is nested in the global atmosphere-only HadAM3P model of weather@home system and is driven by prescribed sea surface temperatures (SSTs) and radiative forcing (Massey et al. 2015; Guillod et al. 2017) to generate the required model

ensembles with initial condition perturbations. The model includes a sulphur cycle (Jones et al., 2001) and uses the updated Met Office Surface Exchange Scheme version 2 (MOSES2; Essery et al., 2003). Recent ECLIPSE v5a  global emissions dataset (Klimont et al., 2013) is used to prescribe the sulphur dioxide fields in the model. Information about the procedure of using this data in the model is given the supplementary material.

HAPPI experiments are designed to address research questions relating to 1.5°C and 2.0°C warming and as part of the experiments weather@home system is used to generate large model ensembles (Massey et al., 2015; Otto, 2017; Stainforth et al., 2005). Following the experimental set up of the HAPPI framework (for details see Mitchell et al., 2017), this study uses experiments of three decadal model ensembles:

1.  Actual climate (denoted as 'ACT') model ensemble with 98 members per year representing the current decade (2006–2015) with observed SST data from the Operational Sea Surface Temperature and Sea Ice Analysis (OSTIA) dataset (Donlon et al., 2012; Stark et al., 2007) and present-day atmospheric GHG and aerosol concentration.

        2.  HAPPI 1.5 model ensemble (2091–2100) with 98 members per year representing 1.5°C warmer than pre-industrial (1861–1880) climatic conditions, and

3.  HAPPI 2.0 model ensemble (2091–2100) with 98 members per year representing 2.0°C warmer than pre-industrial (1861–1880) climatic conditions.

For the HAPPI 1.5 model ensemble, the RCP 2.6 scenario is used to provide the model boundary conditions. In RCP 2.6 scenario, the mean global temperature reaches to ~1.55°C by 2100 (Mitchell et al., 2017). Since there is no analogous CMIP5 simulation

available which results in ~2°C warmer temperatures relative to preindustrial levels, a weighted combination of RCP2.6 and RCP 4.5 is used to provide the model boundary conditions of SST and sea ice for the HAPPI 2.0 model ensemble. The global mean temperature response reaches to ~2.05°C by the end of century in the HAPPI 2.0 model ensemble (Mitchell et al., 2017). Following the RCP2.6 protocol, anthropogenic aerosol concentrations are approximately one-third of the current levels (IPCC, 2013) in both HAPPI scenarios.


In addition, we use two model ensembles of hypothetical climate conditions:

        4.  Natural ('NAT') model ensemble with 98 members per year representing the current decade (2006–2015) climatic conditions, but here the modelled SST patterns of anthropogenic forcing (ΔSSTs) are removed from the OSTIA observed SSTs to simulate a counterfactual world. ΔSSTs are generated from the CMIP5 archive. In this case, HistoricalNat

simulations are subtracted from the Historical simulations as described in Schaller et al. (2016), thereby generating a representation of human influences on the SSTs that can be removed from the OSTIA SSTs. GHG and aerosol concentrations are set to pre-industrial levels.

5. 'GHG-only' model ensemble with 98 members per year representing the current decade (2006–2015) climatic conditions, but with anthropogenic aerosol concentrations reduced to pre-industrial levels to simulate a hypothetical climate, where impacts of aerosols are removed. The difference between ACT and GHG-only conditions simulates the net aerosol effect under current conditions assuming additive behaviour of different radiative ~~forces~~ forcing. The GHG-only model ensemble with anthropogenic aerosols reduction scenario in the HadRM3P model is satisfactorily representative when compared with the other GCMs (Haustein et al., in progress). Based on the very limited sample of CMIP5 aerosol only (AA) experiments, we found that the resulting ΔSST patterns are reasonably similar compared with ΔSSTs from ACT minus GHG-only (not shown).

## 2.3 Methods

To understand how seasonal mean rainfall changes from one climate condition to another, we looked at percent change (PC) and standardized precipitation index (SPI) change between different two forcing scenarios (from pre-industrial NAT to current ACT, ACT to HAPPI 1.5, HAPPI 1.5 to 2.0 and from ACT to GHG-only). PC and SPI analyses are done for rainfall changes over the central parts of South Asia and then over Bangladesh. For brevity, the supplementary text includes the details of the calculation methods for PC and SPI changes.

In Bangladesh, any MAM extreme rainfall events are known to cause flash floods and substantial crop damage (Ahmed et al., 2017). Bangladesh receives more than 75% of the annual total rainfall during JJAS (Shahid, 2010). An extreme rainfall event in this period can therefore cause wide-spread flooding and landslide eventually leading to loss of lives and livelihoods. A high impact post-monsoon (Oct-Nov; ON) rainfall event may be associated with the coastal floods that occur due to storm surges or tidal effects along the northern part of Bay of Bengal (Hossain, 1998).

Considering meteorological hazards and potential impacts, MAM and JJAS extreme rainfall events are analyzed in this study while, ON rainfall events are excluded. Winter (during Dec-Jan-Feb; DJF) season is also excluded because little or no rain occurs during DJF and we are interested in wet extremes.

We use sub-regions 1-4 located at north-west (88°-90°E, 24°-26°N), north-east (90.5°-92.5°E, 24°-25.5°N), south-west (89°-91°E, 21.5°-23.5°N) and south-east (91°-93°E, 20.5°-24°N) respectively. The two eastern sub-regions 2 and 4 are the rainier parts of the country compared to the other two western sub-regions of 1and 3. MAM and JJAS extreme rainfall events with up to 100 years of return periods are adequately well captured by the HadRM3P model over these same 4 sub-regions in Bangladesh when compared to high resolution gridded observation datasets (Rimi et al., 2019a). Such pre-evaluated model simulations provide confidence in analyzing comparative risks of extreme rainfall events under different forcing scenarios. The model is therefore considered to be fit for assessing climate change impacts on extreme rainfall events in Bangladesh.

In Bangladesh 1- to 10-day high impact rainfall events can trigger flooding and landslides. For example, at north-east Bangladesh (sub-region 2), more than 150 mm MAM rainfall over a 6-day period can lead to an early flash flood (Ahmed et al., 2017). In contrast, at south-east Bangladesh (sub-region 4), more than 350 mm JJAS rainfall over a 3-day period is enough to cause a landslide (Ali et al., 2014). For a wide-spread river flooding e.g., the Brahmaputra River Basin flooding in August 2017, 10-day extreme rainfall event is considered (Philip et al., 2018).

Considering such variations in rainfall magnitudes causing different hazards, we focused on daily and 5-day rainfall events to analyze the potential risks. The seasonal cycles of presented here are based on 5-day rainfall, which is used to represent the timescale responsible for river flooding as opposed to daily extremes that cause flash floods primarily in the pre-monsoon season.

"Return time" of an event, also known as the "return period" is the likelihood of an event occurring, defined by a particular variable exceeding a certain threshold during a given time interval. If variable X is equal to or greater than an event of magnitude $x_T$, occurs once in T years, then the probability of occurrence $P(X \geq x)$ in a given year of the variable is (Wilks, 2011):

$$P(X \geq x = \tfrac{1}{T}) \text{ or, } T = \tfrac{1}{1 - P(x \geq x_T)}$$

A "1 in 10 year event" is an event with a 10% chance of occurring. On the contrary, the rarest event is a "1 in 1000 year event",
with a 0.1% chance of occurring in a given year. The rainfall amounts associated with the 50- or 100-year return periods are extracted from the 98th and 99th percentiles, respectively, of a fitted distribution (i.e., $[1-0.98^{-year}]^{-1} = 50$ years and $[1-0.99^{-year}]^{-1}$ =100 years) (Wilks, 1993). The uncertainty of the return period is calculated using bootstrapping method. The time series of each ensemble is resampled a 1000 times using bootstrapping to derive 5 to 95% confidence intervals for return periods. We note that structural model uncertainty (such as parameter sensitivity) is not included in our uncertainty estimate.

Return time plots are used to explore the relative risks of rare events (like those with probabilities of <= 1 in 100 years). To construct the return time plots for MAM and JJAS daily (and 5-day) rainfall events, we use 98 plausible model realizations for the 10-year period of each model ensemble. For each year, three (MAM) and four (JJAS) months of data are used for pre-monsoon and monsoon season, respectively. The model uses a 360-day calendar with all 12 months spanning for 30 days. Therefore, we
have 3x30x10x98 = 88,200 and 4x30x10x98 = 117,600 simulated values to calculate the return periods of MAM and JJAS rainfall events, respectively.

Such large sample size allows us to estimate a range of physically plausible climate conditions with focus on the tails of the distribution, which can be robustly determined. We consider all days of a season for calculating the return periods or rainfall
events. In this way, we can look at low intensity rainfall events with minimum return period of 1 year, and also high intensity rainfall events with up to 98x10 = 980 years return period. However, we focused on rainfall events with high return periods that are relevant for impacts and adaptation planning (i.e., 10-100 year events).

To add a qualitative representation of the year-to-year natural variability from the ACT model ensemble, we use two wettest and
two driest years during the decade of 2006-2015. The spatiotemporal average for the corresponding sub-region and season over the 10-year simulation period has been used to identify the two wettest and driest years. ACT model ensemble has the same forcings in each year for the historical period of 2006-2015; the only variability playing a role in changing rainfall intensity is therefore the natural variability of SSTs.

For this reason, these two wettest and driest years of ACT model ensemble approximately indicate how much natural SST variation can contribute to changing rainfall intensities. By comparing these two subsampled model ensembles with the other ensembles of different forcing scenarios, we can estimate the signal-to-noise ratio in the return period plots. The supplementary material of Table S2 lists the wettest and driest years for MAM and JJAS over the four different sub-regions.

In order to examine robustness of the return time plots based on weather@home's RCM, HadRM3P model outputs; we compare similar results based on decadal simulations from additional four atmospheric general circulation models (AGCMs) from the HAPPI Tier-1 experiments (Mitchell et al., 2017). All forcings scenarios are available in the four AGCM simulations to do the comparison except for GHG-only scenario. These model ensembles are available with ≥100 members per year with initial

condition perturbations however; we have used 98 members per year to compare with HadRM3P results. The basic information about these four AGCMs is given at Supplementary Table S3, while more details and evaluation of these models is available in Chevuturi et al., (2018).

In order to quantify changes in the probability of occurrence of extreme rainfall event, we use Risk Ratio (RR), which is calculated as RR = $P_f$/ $P_{cf}$ (NAS, 2016). Here $P_f$ denotes the probability of the event in factual climate including climate change (ACT, HAPPI 1.5 and HAPPI 2.0) and $P_{cf}$ denotes the probability of an event of the same magnitude in a counterfactual climate without anthropogenic climate change (NAT). But, in case of RR for GHG-only scenario, it is calculated with regard to ACT instead of NAT. We quantified the changes in the RRs for four event thresholds during MAM and JJAS with return period of 10,
20, 50 and 100 year over the four sub-regions of Bangladesh.

In case of the model results, we have large enough an ensemble to calculate the probabilities of occurrence (P) of the event in question explicitly by means of the different forcing scenarios. For example, suppose a 200 mm/day precipitation event has a $P_{actual}$ of 50 years and a $P_{natural}$ of 100 years. The resulting change of probability of that event would simply be a doubling (RR= 2)
due to the change in forcing.

To calculate the upper and lower limits of the uncertainty of RR we have used the following formula (based on error propagation model for independent contributors): Upper limit of RR uncertainty = $\sqrt{(a^2 + c^2)}$; and Lower limit of RR uncertainty = $\sqrt{(b^2 + d^2)}$
Where, a = upper limit of uncertainty of $P_{ACT}$,      b = lower limit of uncertainty of $P_{ACT}$,
20         c = upper limit of uncertainty of $P_{NAT}$,      d = lower limit of uncertainty of $P_{NAT.}$

## 3 Results and Discussion

### 3.1 Model Evaluation

In Figure 1, seasonal cycles of 5-day rainfall as in the simulations of model ensembles under five different forcing scenarios (ACT, NAT, GHG-only, HAPPI 1.5 and HAPPI 2.0) and two observations (APHRODITE and CPC) are shown. The coloured
lines represent the ensemble means, with light-coloured shading representing the 10-90% percentile ranges (only shown for ACT model ensemble and the observations).

The seasonal cycles of 5-day rainfall from the different model ensembles are adequately representative of the observed seasonal cycles. Most of the observed rainfall is found to be within the 10-90% confidence intervals of the model data. We find an early
monsoon onset in the model simulations, which is also reported in previous studies (e.g., in Caesar et al., 2015; Fahad et al., 2017; Janes and Bhaskaran, 2012). However, JJAS rainfall is underestimated by 25-50% depending on the observational dataset and sub-regions. This bias is higher (up to 50% dry bias) in the wetter sub-regions of 2 and 4 (Figs. 1b & d) and lower (up to 30% dry bias) in the drier sub-regions of 1 and 3 (Figs. 1a & c). Underestimation of JJAS rainfall is reported in other model based studies over Indian monsoon region (Goswami et al., 2014; Kumar and Dimri, 2019; Saha et al., 2014) and specifically in Bangladesh
(Caesar and Janes, 2018; Islam, 2009; Macadam and Janes, 2017).

The bias is apparently present in all model scenarios used in this study; hence it is unlikely to affect the comparison between model scenarios. We also note that the signal of the change due to the changing climate is relatively small in comparison to the total rainfall. Therefore, the model is considered fit for purpose in assessing the potential impacts of climate change on extreme
rainfall events. The differences between the forcing scenarios throughout the seasonal rainfall cycle are discussed below.

## 3.2 Impact of Climate Change and Aerosol Reduction on Seasonal Mean Rainfall

Our results suggest that changes in mean rainfall due to global warming are significant for both MAM and JJAS, and that aerosols play an important role in determining the magnitude of future changes (Figs. 2 & 3). Based on PC, these changes are particularly evident during MAM, yet a smaller PC during JJAS can still have a significant impact given the magnitude of rainfall. Relative changes between pairs of forcing scenarios show large spatial variability over Bangladesh and the wider central South Asia region, although they suggest a general wetting trend across Bangladesh for both 1.5°C and 2.0°C warmer worlds.

During MAM, results show a non-linear response to temperature change in the PC over the eastern part of South Asia (Figs. 2a, b, & c) that is likely to be caused at least in part by the response for aerosols in Fig. 2d. The present-day ACT PC relative to NAT indicates that mean MAM rainfall is reduced by 15-30% over the eastern parts of South Asia and increased by 15-25% over the northern parts Bangladesh (Fig. 2a). Figure 2d shows the spatial distribution of the "omitted" aerosol induced rainfall over the South Asia region. Once aerosol levels drop to one-third of its current values (following the RCP2.6 protocol, IPCC, 2013), an increase of up to 20% in MAM rainfall is very likely to happen over most parts of South Asia region. Associated with this increased rainfall, PC relative to ACT in HAPPI 1.5 increases up to 20% over South Asia (Fig. 2b), with Bangladesh being the region where the aerosol effect dominates the total change (Figs. 2f & h). Across Bangladesh, our results indicate that MAM rainfall increases approximately linearly with temperature, suggesting a relevant role for thermodynamic effects and perhaps a smaller role for dynamic changes as far as our HadRM3P model results are concerned. By linear response, we meant steady and gradual increase in the climate change impact on rainfall from one forcing scenario to another due the warming effects starting from NAT to ACT, ACT to HAPPI 1.5 and HAPPI 1.5 to HAPPI 2.0. The additional warming effects in HAPPI 2.0 increase the mean MAM rainfall by an extra 10-20% over Bangladesh (Fig. 2g), in contrast to other parts of Asia. We note that our conjectures are speculative at this point, yet likely based on established research into monsoon dynamics (e.g., Bollasina et al. 2011).

Using other RCM projections (based on RCP8.5), Fahad et al. (2017) pointed out that MAM mean rainfall may increase by up to 20% relative to their baseline period (1971–2000) over the eastern mountainous region of Bangladesh, in line with our results for 1.5 and 2.0°C warming. However, the fact that the northern parts of India show very non-linear behaviour with regard to rainfall PC in response to the combined GHG and aerosol-related radiative forcing (Figs. 2a-d) is indicative of circulatory, dynamic shifts with stronger warming.

The PC of mean JJAS rainfall (Fig. 3a) in ACT relative to NAT indicates a weakening monsoon over central India and strengthening of the monsoon over Bangladesh and north-east India (10-15% increase). Evidence for reduced JJAS rainfall amounts over the last few decades in South Asia region is also found in the observational records (Bollasina et al., 2011; Srivastava et al., 2010; Turner and Annamalai, 2012; Wang et al., 2012). In contrast, the CMIP5 models simulate about 2.3% increase in rainfall per degree of warming for the Indian summer monsoon (Menon et al., 2013) due to an increase in moisture availability in a warmer world. These conflicting results can be attributed to an underestimated aerosol effect in many CMIP5 models. Subsampling those models that include indirect aerosol effects helps to resolve the discrepancy (Bollasina et al., 2011; Turner and Annamalai, 2012).

The most important change in the PC occurs in HAPPI1.5 relative to ACT (Fig. 3b). Comparing HAPPI 1.5 and 2.0, we find an additional increase in mean JJAS rainfall but of lower magnitude (a further 10 to 20% increase; Fig. 3c). We find a very strong drying tendency during JJAS due to anthropogenic aerosols over most parts of South Asia (Fig. 3a). Correspondingly, the "committed" rainfall increase, which will be realised once aerosol emissions are reduced (Fig. 3d), is in the order of 15-30%. This means that the observed drying is entirely caused by the aerosols that have overcompensated, the GHG induced rainfall increase.

In Bangladesh (Figs. 3e-h), the aerosol effect is less strong and GHG induced intensification of summer monsoon rainfalls have already increased the risk of more intense rain. The Standardized Precipitation Index (SPI) analysis for the pre-monsoon and monsoon seasons (Supplementary Figs. S1 & 2) corroborates our results from the PC analysis.

In addition to PC and SPI analyses, we looked at the absolute rainfall (Figs. 4 & 5) for all five forcing scenarios during MAM and JJAS (median, as well as the 25-75[th] percentiles of seasonal mean rainfall) to explain the variability in the mean of absolute rainfall relative to the change between scenarios over the four sub-regions in Bangladesh (for locations of the sub-regions, see boxes in Figs. 2e & 3e). Changes in mean absolute rainfall are much more pronounced over sub-regions 1 and 2, where both MAM and JJAS rainfall exhibit clear shifts from one forcing to another forcing scenario (Fig. 4). On the other hand, over sub-
regions 3 and 4, only JJAS rainfall exhibited a robust shift (Fig. 5 b & d). The absolute aerosol effect is strongest in summer during JJAS, yet the relative change is smaller which is in line with lower rainfall PC as discussed above. Most importantly, aerosols play a dominant role in all sub-regions and seasons, except for MAM rainfall over sub-region 3 and 4. Despite more effective aerosol removal from the atmosphere by means of wet deposition during JJAS, high regional emission rates prevent drastic reductions of the aerosol optical depth. Consequently, direct and indirect aerosol effects, accompanied by feedbacks such
as reduced lapse rate, reduced boundary layer turbulence, or a modified land-sea circulation, remain to be a potent driver for changing monsoonal rainfall amounts.

For future warming scenarios of HAPPI 1.5 and HAPPI 2.0 compared to ACT, we find robust linear increase in (absolute) rainfall in almost all sub-regions and seasons. We notice a persistent change with increase in absolute mean rainfall from ACT to HAPPI
1.5 and HAPPI 1.5 to HAPPI 2.0. Conversely, we find no clear shifts between NAT, ACT and HAPPI 1.5 during MAM over sub-region 3 and 4 (Figs. 5a & c). While aerosol effects are consistent with those in other regions, the GHG induced rainfall is hampered, likely due to dynamic changes such as a delayed onset of the monsoon in response to warming (Bollasina et al., 2011; Zhao et al., 2019). The proximity to the Indian Ocean may also be a contributing factor. Even though the atmosphere can hold more moisture, the slower ocean warming stabilises the atmosphere over sea in the same way aerosols stabilise the atmosphere
over land.

Impact of climate change and aerosol reduction on seasonal mean rainfall (as in PC, SPI and absolute) is in agreement with the findings in the seasonal cycles (Fig. 1) presented before. As shown in Fig. 1, the monsoon onset in sub-region 3 and 4 (Figs. 1c & d) does not change notably under different forcing scenarios as far as 5-day rainfall is concerned. Otherwise, the aerosol and GHG
induced response is consistent with the conclusions based on the spatial maps across the four sub-regions. Sub-regions 1 and 2 show considerable changes in rainfall strength during MAM, with an earlier onset in the HAPPI 2.0 scenario over sub-region 2. The most pronounced change is simulated at the peak of monsoon season, in early June over sub-region 2, with an associated increase in magnitude of almost one-third between NAT and HAPPI 2.0. It is noteworthy that this increase in rainfall is very linear with progressively warmer climate conditions (Fig. 1b).


### 3.3 Extreme Rainfall Events

An analysis of changes in extreme rainfall events suggests that Bangladesh is likely to experience significantly higher magnitude of 1 and 5-day rainfall events (Figs. 6-9 and S3-S6) during both pre-monsoon and monsoon seasons across all sub-regions for a 1.5°C change. The only exceptions included MAM rainfall events over sub-region 4 (Fig. 7b) and JJAS rainfall events over sub-
region 3 (Fig. 9a). In contrast, changes between HAPPI 1.5 and HAPPI 2.0 are only significant in JJAS over sub-regions 1 and 2 (Figs. 8a & b). Overall, the signal-to-noise ratio is higher across all sub-regions, during JJAS compared to that during MAM. During MAM, the highest and lowest signal-to-noise ratio is over sub-region 1 and 3, respectively (Figs. 6a & 7a). On the other

hand, during JJAS, we find the highest and lowest signal-to-noise ratio is over sub-region 3 and 1, respectively (Figs. 9a & 8a). The lower the ratio, the more difficult it is to establish causality as natural variability due to ENSO or circulation anomalies is higher.

The most linear rainfall response to warming is simulated in sub-region 2 in MAM and JJAS (Figs. 6b & 8b), with aerosols masking approximately 50% of the increased risk with regard to 1-in-100-year NAT return time. Hence future rainfall in sub-region 2 continues to increase, with accelerated pace once aerosol levels drop significantly. Sub-region 1 is likely to receive more extreme rainfall as well with continued warming, with strong increases once aerosol levels drop. Sub-regions 1 and 2 are equally sensitive to aerosols, yet dynamic feedback processes might partially counter the thermodynamic increase in rainfall risk with
continued warming.

Figures 10 and 11 give simple illustration for the change in risk ratios, which remarkably vary with seasons (pre-monsoon and monsoon) as well as locations (sub-regions of 1-4) and also indicate how aerosols impact risk ratios. Supplementary material of Table S4 and S5 present the risk ratios with associated uncertainty ranges for MAM and JJAS rainfall over four sub-regions,
respectively. Figure 10 demonstrates that there is noticeable masking effect of aerosols during MAM that repress the change between NAT and ACT worlds at sub-region 1. Hence, present-day risk for MAM rainfall event has not changed (see RR for ACT/NAT in Fig. 10a). But then the risk of extreme rainfall event with respect to 1-in-100-year NAT return time increase by a factor of 4 (with uncertainty range 2.0-7.0) in a 1.5°C world (see RR for HAPP1.5/NAT in Fig. 10a). In contrast, the aerosol masking effect during MAM is minimal at sub-region 2; leading to a robust change between NAT and ACT worlds (see RR for
ACT/NAT in Fig. 10b).

We find persistent increase in the RRs for JJAS extreme rainfall events at the sub-regions 2 and 4. At sub-region 2, risk of JJAS extreme rainfall event with respect to 1-in-100-year NAT return time increases 3-fold (with uncertainty range 1-4) in a 1.5°C world and then 4.6-fold (with uncertainty range 2.9-7.2) in a 2.0°C world (see RRs for HAPP1.5/NAT and HAPP2.0/NAT in Fig.
10d). At sub-region 4 (Fig. 11d), where current risks of JJAS extreme rainfall events are already increased 3.9 times (with uncertainty range 2.6-5.8) with respect to 1-in-100-year NAT return time; the risk for similar event increases 4.1 times (with uncertainty range 2.2-5.3) in a 1.5°C world and 5.5 times in a 2.0°C world (with uncertainty range 3.5-7.8).

**4 Conclusions**

Results of the weather@home HadRM3P South Asia regional model suggest that both, 1.5°C and 2.0°C warming are projected to
increase seasonal mean and extreme rainfall probabilities during the pre-monsoon and monsoon seasons across Bangladesh. The magnitude of change exceeds the current internal year-to-year variability in the associated sub-regions 1 and 2 during both pre-monsoon and monsoon seasons. These increases are likely to be amplified by a reduction in aerosols, consistent with previous findings (e.g., Samset et al., 2018).

We find that there are large spatial variations in the patterns of changes in the relative risks of extreme rainfall in Bangladesh. Sub-regions 1 and 2 shows an enhanced susceptibility to aerosols during the pre-monsoon season; whereas, sub-regions 3 and 4 show a smaller aerosol sensitivity during the monsoon season. Aerosols have reduced the absolute daily rainfall amount by up to 1mm (~ 5-10%) during the monsoon season in sub-region 1 and 2, comparable to the simulated rainfall change in a future 2.0°C warming scenario. This is in line with a growing array of research that has shown that anthropogenic aerosols play a substantial
role in modulating the strength of the monsoon in South Asia (Bollasina et al., 2011, 2013; Lau and Kim, 2010; Ramanathan et al., 2005).

As far as other regions in South Asia are concerned, our results imply that the present-day decline in the mean monsoon seasonal rainfall can be explained by the existing atmospheric aerosols impacts, which offsets the GHG-induced global warming effects. Future aerosol removal from the atmosphere will unmask the GHG induced rainfall increase with surprisingly fast changes in risk due to the non-linear nature of the imposed external forcing contributions (e.g., over sub-region 1 in pre-monsoon season). For that reason we emphasize that the impacts of aerosol reductions on the changing risks of extreme rainfall events should be considered for future risk assessments.

Considering the importance of the climate change impacts found in sub-region 1 and 2 for MAM and JJAS rainfall extremes, Figs. 12 and 13 demonstrate how robust the HadRM3P results are compared to four other HAPPI AGCMs of MIROC5, ETH_CAM4, CanAM4 and NorESM1 models. Similar comparisons for other sub-regions and seasons are presented at the supplementary material for brevity (in Figs. S7 - S12). For MAM rainfall extremes over sub-region 1, HadRM3P, MIROC5 and NorESM1 models suggest increasing risks but ETH_CAM4 and CanAM4 models show increasing risks only for a 2.0°C warmer climate condition. MIROC5 model shows a linear progression of MAM rainfall risks, increasing from NAT to Act and then from ACT to 1.5° and 2.0°C warmer conditions (Fig. 12b). In contrast, HadRM3P model shows a non-linear change in the MAM rainfall risks with nearly no increase between NAT and ACT but then a drastic increase in warmer scenarios (Fig. 12a). Since the other HAPPI AGCMs do not provide a GHG-only scenario, we can only infer causality of potential non-linear changes between the warming scenarios from HadRM3P model in this case. For JJAS rainfall extremes over sub-region2, all five models suggest increasing risks in warmer conditions (see Figs. 13a-e). However intensities of rainfall events vary within different models. For JJAS rainfall extremes over sub-region 2, HadRM3P model results show better agreement with NorESM1 model in terms of both increasing risks and rainfall intensities (compare Figs. 13a &e); probably because this model includes updated module for aerosols (Kirkevåg et al., 2013).

This study for the first time presents a multi-model assessment of both current and future changes in the risks of seasonal extreme rainfall events, considering anthropogenic climate change drivers of GHGs and aerosols, at sub-regional local scales in Bangladesh. These projected changes have important implications for agricultural yields and associated economic losses, particularly during the pre-monsoon season. For example, at north-east Bangladesh, in 2017, pre-monsoon extreme rainfall caused the earliest flash flood since 2000. Consequent damage of nearly harvestable rice crop in turn, caused the highest price hike in the following year. Anthropogenic climate change drivers are found to make this extreme rainfall event twice as likely (Rimi et al., 2019b). In contrast, during the monsoon season, property damage is more likely to occur when large inhabited areas are inundated on a regular basis. Finding of this study implies that policy-makers and relevant stakeholders not only need to take distinctively different regional responses in extreme rainfall into account, but also the non-linearity in the response. Relying on observed changes can be deeply misleading, creating an unwarranted sense of security. Our study highlights that preparedness for more frequent extremes is key in the northern part of Bangladesh during both the pre-monsoon and the monsoon season. While additional regional model experiments are needed to confirm the weather@home model results, our analysis of available data from other HAPPI GCMs point in the same direction for seasonal extreme rainfall events in Bangladesh. Similar findings are also reported for the larger South Asia region (e.g., Chevuturi et al., 2018; Lee et al., 2018). However, since they do not allow for a quantification of the aerosol effect, we call for more nuanced experiments in that regard in the future.

**Data/Code availability**

Observation data set of APHORODITE is available for download at http://aphrodite.st.hirosaki-u.ac.jp/products.html and CPC can be downloaded from https://climexp.knmi.nl/select.cgi?id=someone@somewhere&field=prcp_cpc_daily. The additional four

HAPPI AGCM simulation data can be accessed at http://portal.nersc.gov/c20c/data/. Analysis codes in R programming language can be made available upon request from the corresponding author.

**Supplement**

The supplement related to this article is available online at: ……………………………………………………….

**Author contributions**

Ruksana H. Rimi's contribution towards this work was performed as part of her DPhil research project. Sihan Li and Sarah N. Sparrow prepared and distributed the computational simulations to generate the data used in the study onto the weather@home system. David C.H. Wallom manages operation of the weather@home/climateprediction.net infrastructure which was used to generate the data. Karsten Haustein generated and extracted the weather@home HadRM3P model data. Md. Zahidul Abedin contributed to the discussion about social implications of the results. All results were analysed and plotted by Ruksana H. Rimi with advice from Myles R. Allen, Karsten Haustein, and Emily Barbour. The paper was written by Ruksana H. Rimi, with edits from all co-authors.

**Competing interests**

The authors declare that they have no conflict of interests.

**Acknowledgments**

We would like to thank the Met Office Hadley Centre PRECIS team for their technical and scientific support for the development and application of weather@home. We would like to thank our colleagues at the Oxford eResearch Centre for their technical expertise. We acknowledge the use of the observation datasets of APHRODITE and CPC. We would like to thank the HAPPI project team and the different modeling centers, who contributed to simulations. Finally, we would like to thank all the volunteers who have donated their computing time to climateprediction.net and weather@home.

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

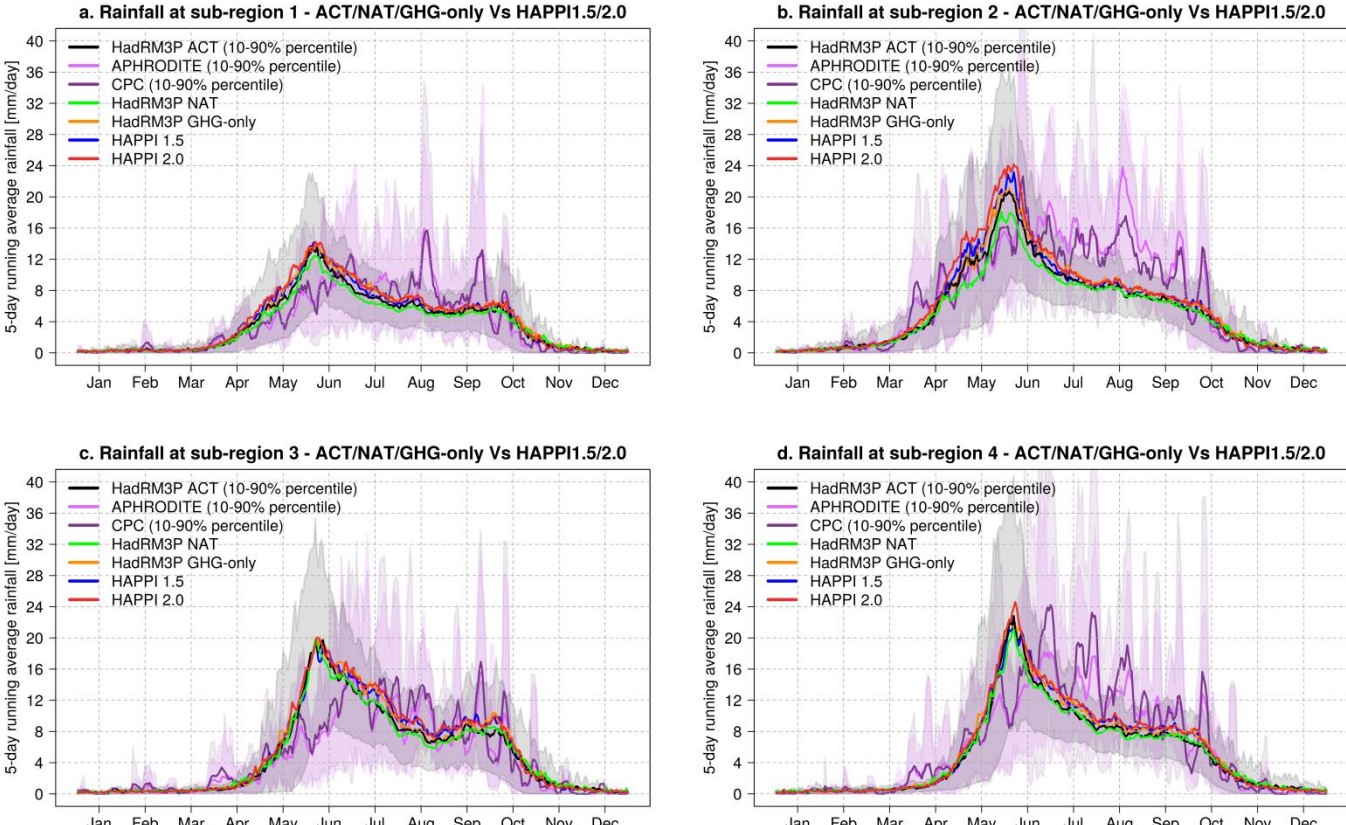

**Figure 1. Seasonal cycles of five day mean rainfall under different forcing scenarios over the four sub-regions of Bangladesh. The ACT (black), NAT (green) and GHG-only (orange), HAPPI 1.5 (blue) and 2.0 (red) ensembles are compared with the observations from APHRODITE (dark purple) and CPC (dark purple). The model adequately captures the seasonal cycle of rainfall compared to observation but underestimates monsoon rainfall. Only over sub-region 2, rainfall is clearly shifting from NAT to ACT, from ACT to GHG-only, from ACT to HAPPI 1.5 and from HAPPI 1.5 to 2.0 forcing scenarios.**

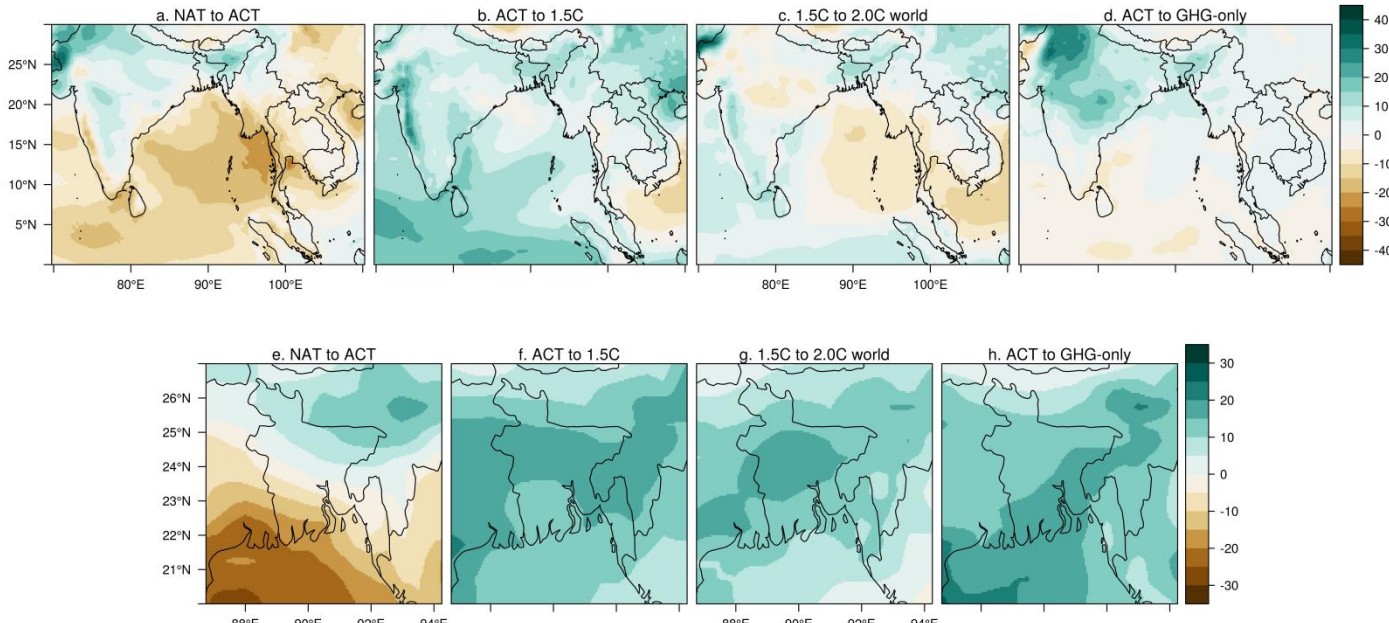

Figure 2. Percentage change (PC) in MAM mean rainfall between different forcing scenarios. The top row (panels a-d) shows the regional PC over central parts of the South Asia. a. ACT rainfall PC relative to NAT over SA b. ACT rainfall PC relative to HAPPI 1.5°C over SA c. HAPPI 1.5°C rainfall PC relative to HAPPI 2.0°C over SA d. ACT rainfall PC relative to GHG-only. Bottom row (panels e-h) shows PC in the same way but over Bangladesh. The four boxes (1-4) on top of the panel e represent the four sub-regions of Bangladesh.

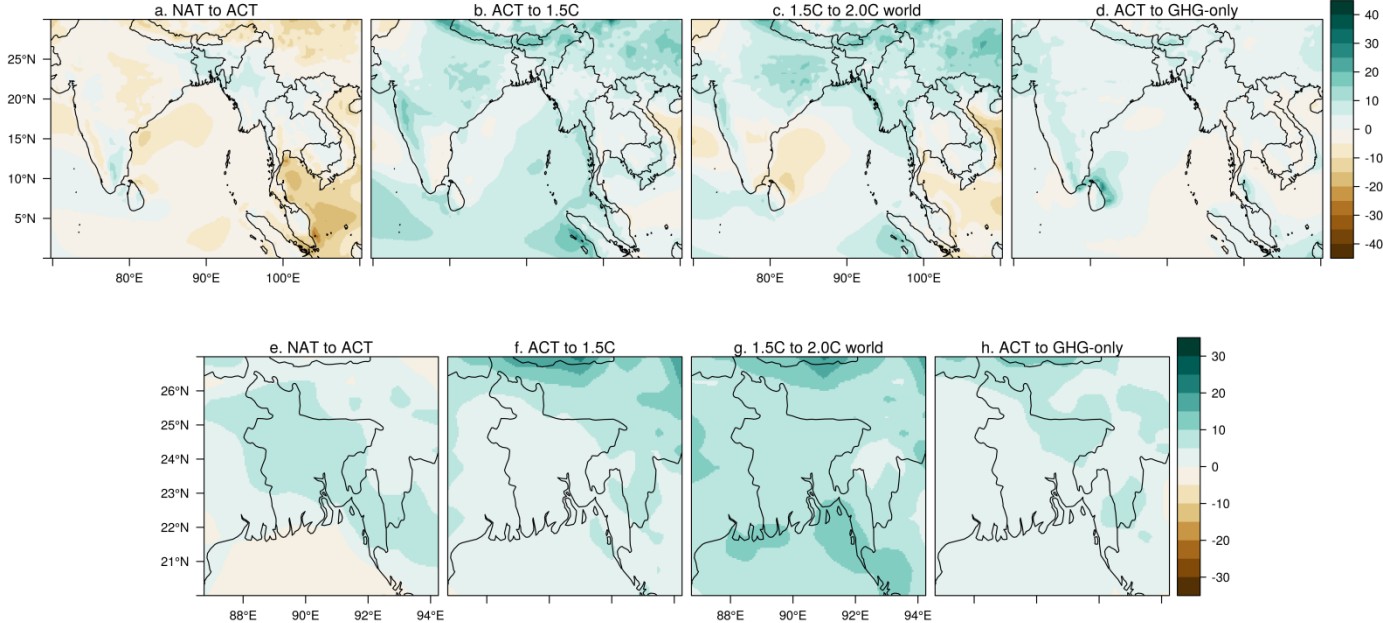

**Figure 3. Same as Fig. 2, but for JJAS rainfall PC. This figure shows that the apparently non-linear response between panels of a, b, and c (or, e, f, g) can be explained by the response for GHG-only (anthropogenic aerosols reduced to pre-industrial levels) in the panel d (or, h).**

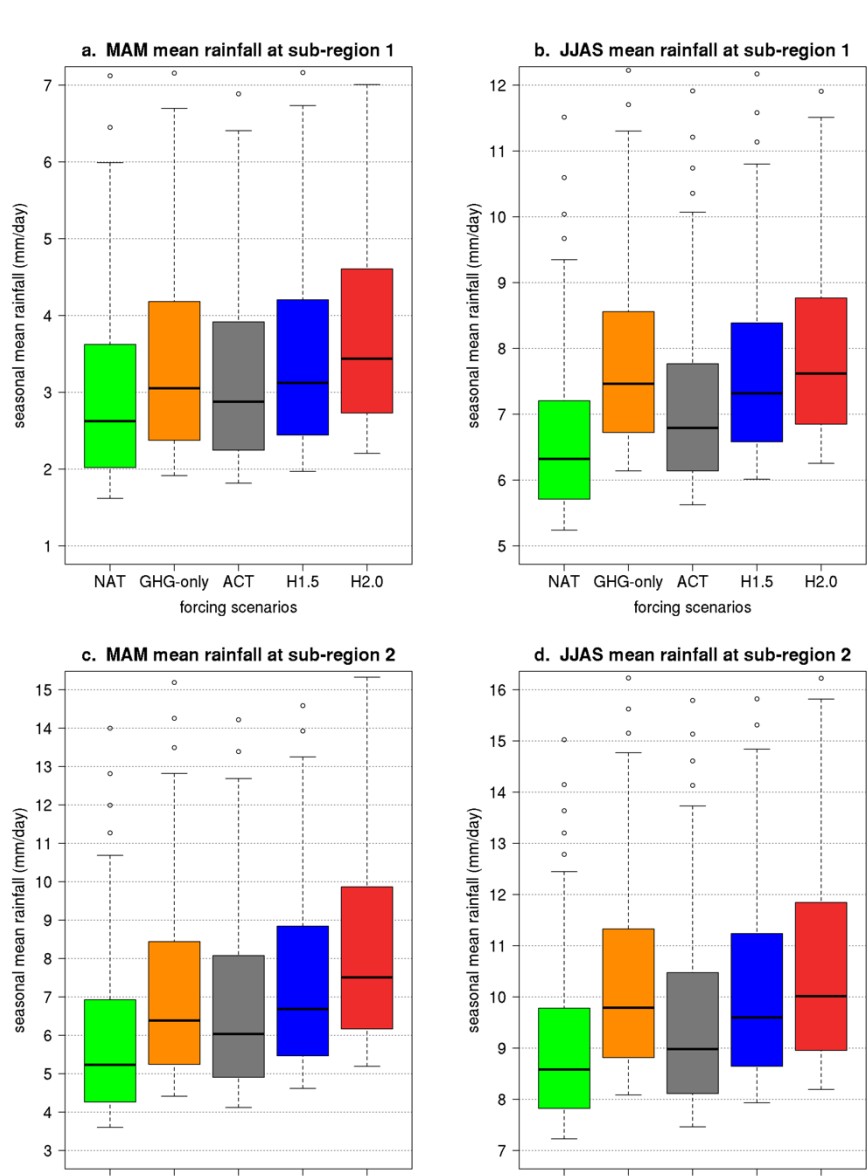

**Figure 4. Seasonal mean rainfall in MAM (left column) and JJAS (right column) over the sub-regions 1 and 2 (top and bottom row) of Bangladesh. Green, orange, grey, blue and red colours represent NAT, GHG-only, ACT, HAPPI 1.5 and HAPPI 2.0 ensembles respectively. Each panel has different y-scale range to clearly indicate the details of changes in the median values between different model ensembles. The horizontal black line in each box indicates the median value, the bottom and top limits of the box represents the 25th and 75th percentiles respectively. The figure shows that aerosol impacts over both sub-regions 1 and 2 are larger in MAM dry season than that in JJAS wet season.**

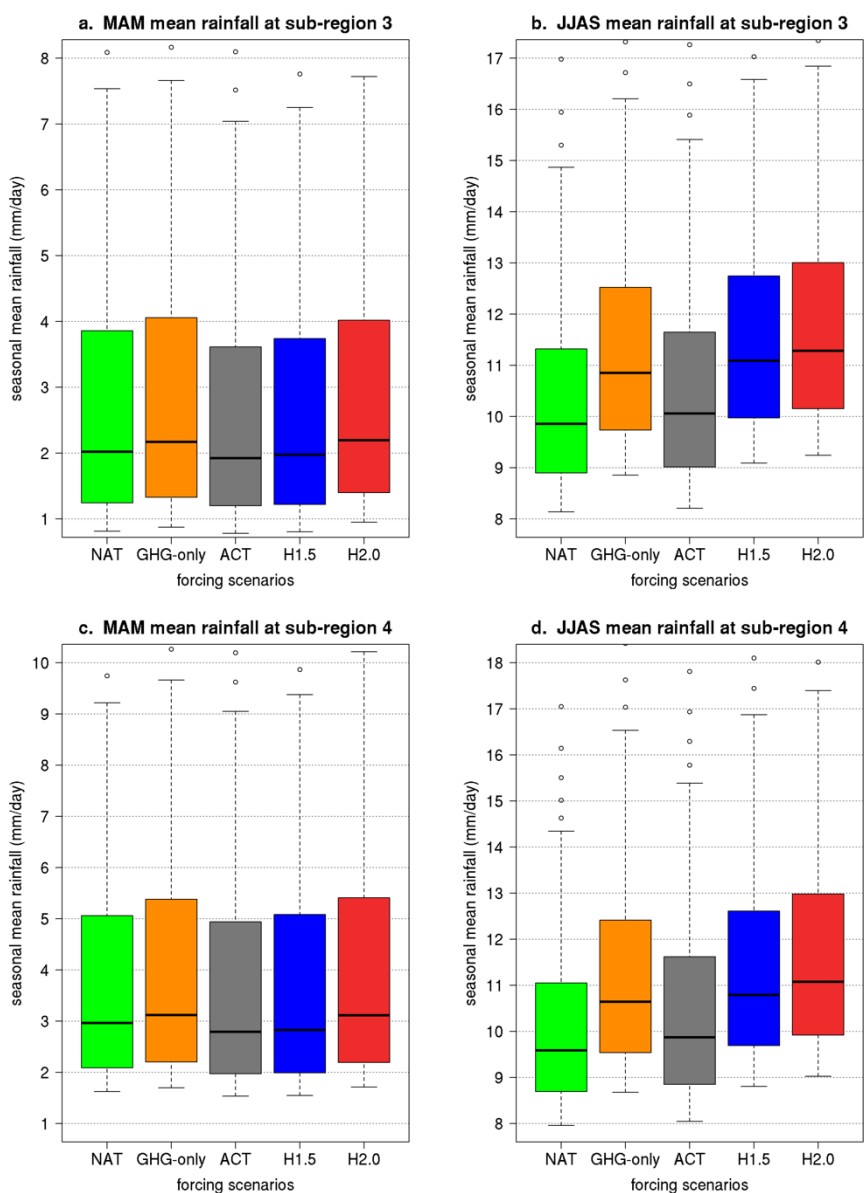

**Figure 5. Same as Fig. 4, but for sub-regions 3 and 4. During MAM over both sub-regions 3 and 4, aerosol effects repress the mean rainfall change between NAT and ACT (i.e., ACT rainfall is lower than NAT). On the other hand, during JJAS over both sub-regions 3 and 4, with lesser aerosol masking effects, ACT has higher mean rainfall than NAT and GHG-only would have noticeably much higher mean rainfall.**

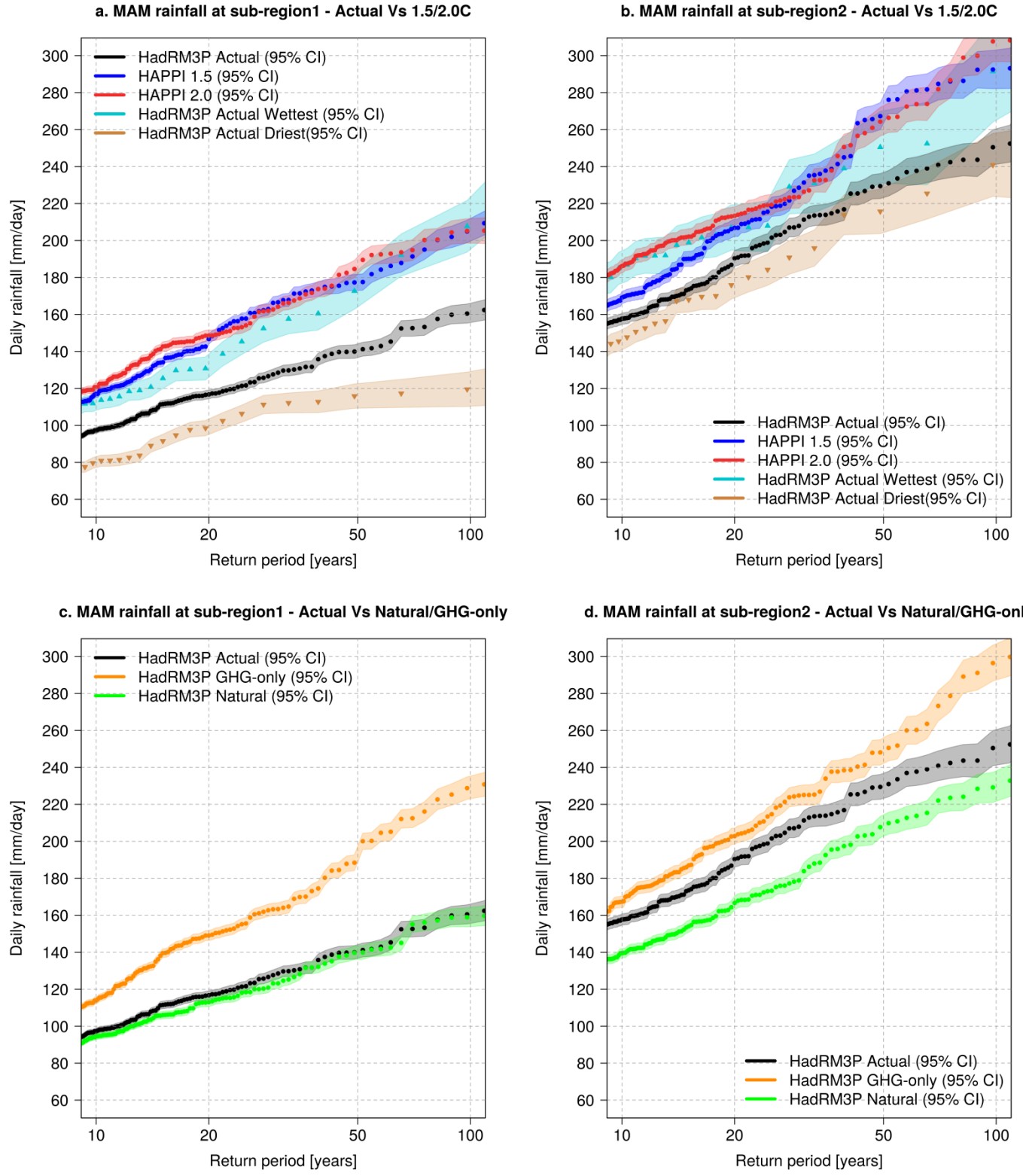

**Figure 6. Return time plots for MAM daily rainfall under different forcing scenarios over the sub-regions 1 and 2 of Bangladesh. The ACT (black), ACT highest (sky-blue), ACT lowest (grey), NAT (green) and GHG-only (orange) ensembles are compared with the HAPPI 1.5 (blue) and HAPPI 2.0 (red) ensembles. Anthropogenic warming effects have not strongly influenced the present-day risks of extreme MAM rainfall over sub-region 1 (Fig. 6a). With a 1.5 or 2.0 degrees' world, this sub-region might see extreme rainfall events with four-fold higher risks.**

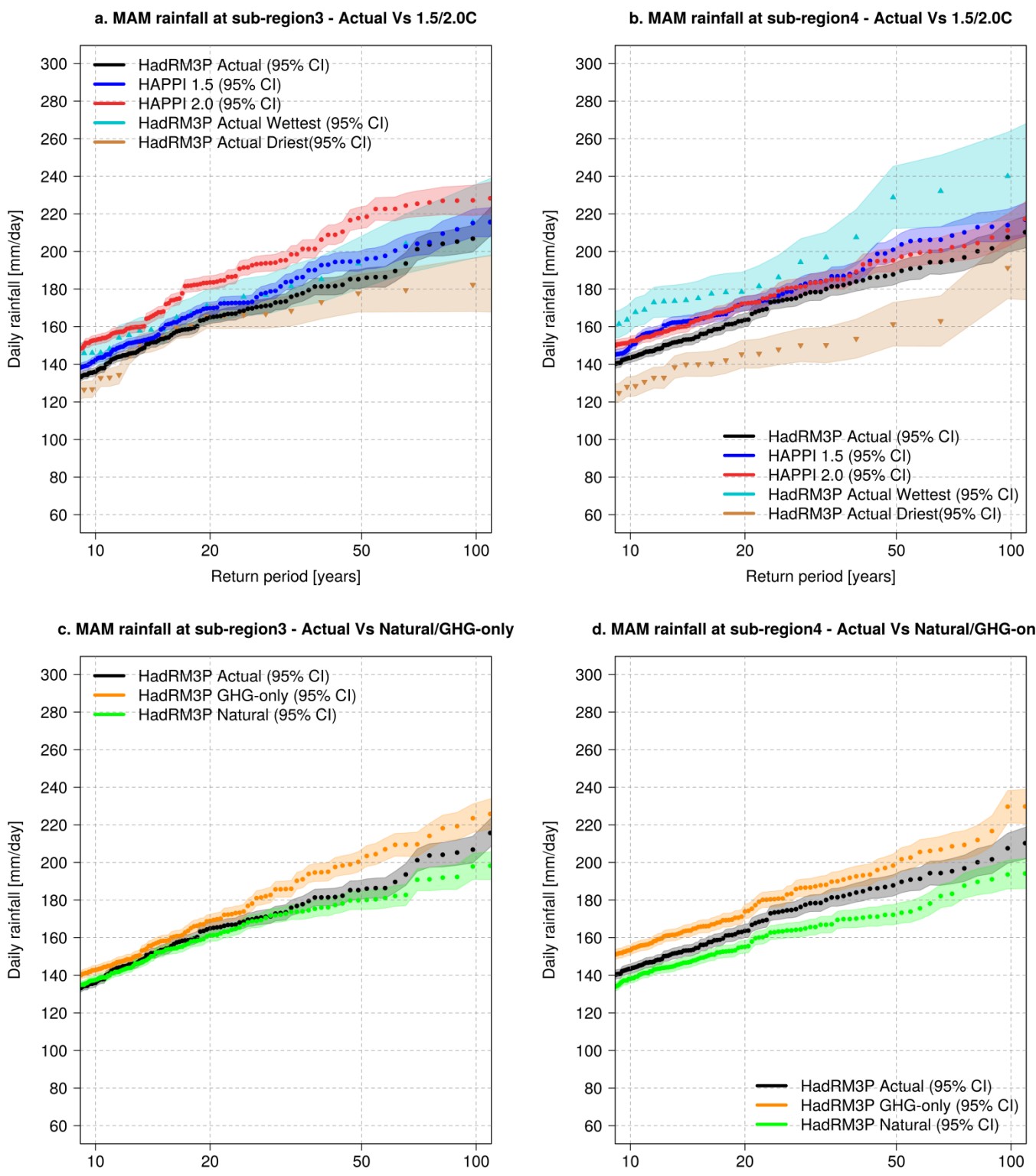

**Figure 7. Same as Fig 6, but showing return time plots for MAM daily rainfall under different forcing scenarios over the sub-regions 3 and 4 of Bangladesh.**

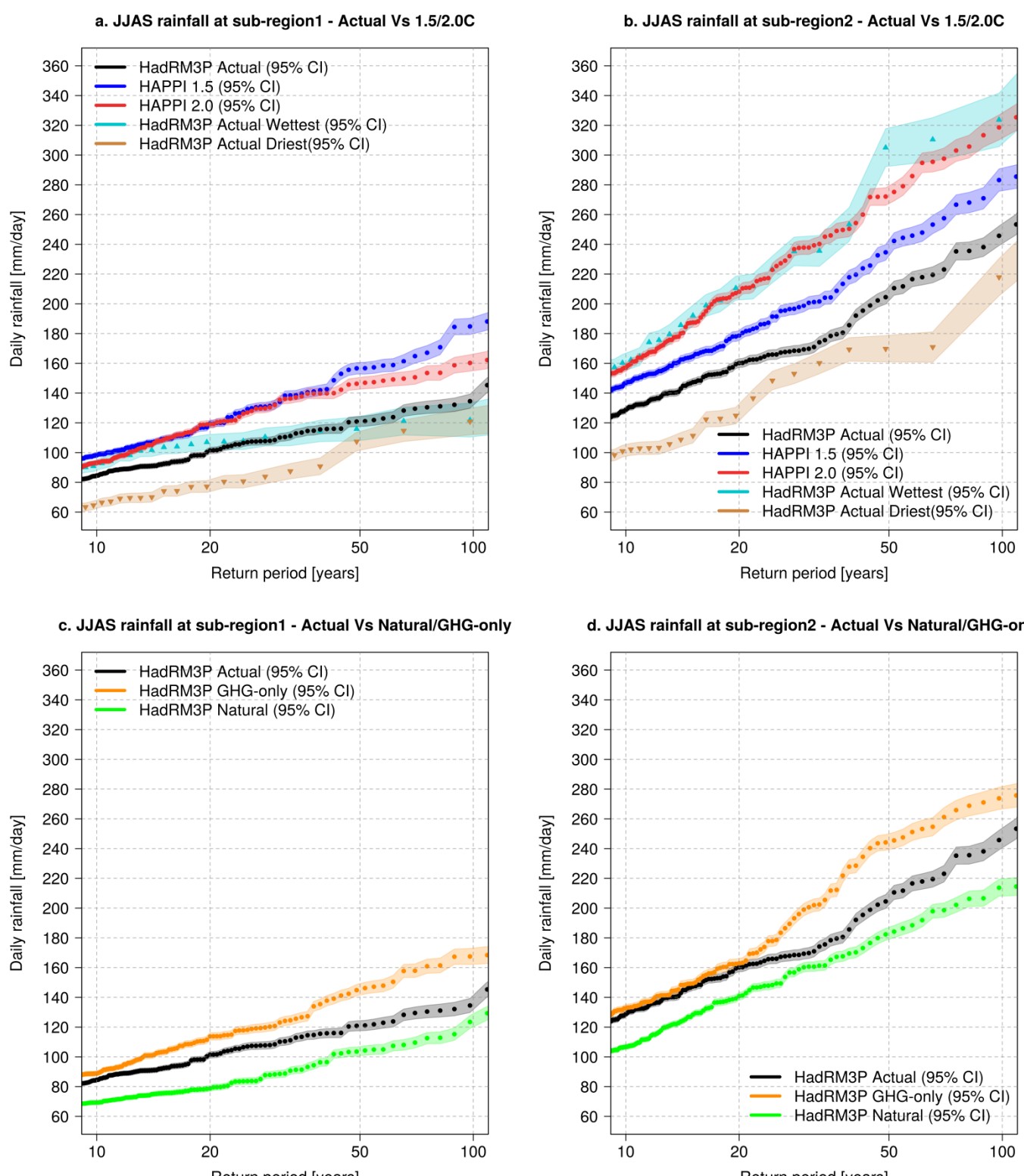

**Figure 8. Return time plots for daily rainfall during monsoon (JJAS) season in different forcing scenarios over the sub-regions of 1 and 2 of Bangladesh. The HadRM3P ACT (black), ACT highest (upper grey with upward triangles), ACT lowest (lower grey with downward triangles), NAT (green) and GHG-only (orange) ensembles are compared with the HAPPI 1.5 (blue) and 2.0 (red) ensembles. The most significant changes in the risks of extreme monsoon rainfall take place in the sub-region2, which is already the wettest part of Bangladesh.**

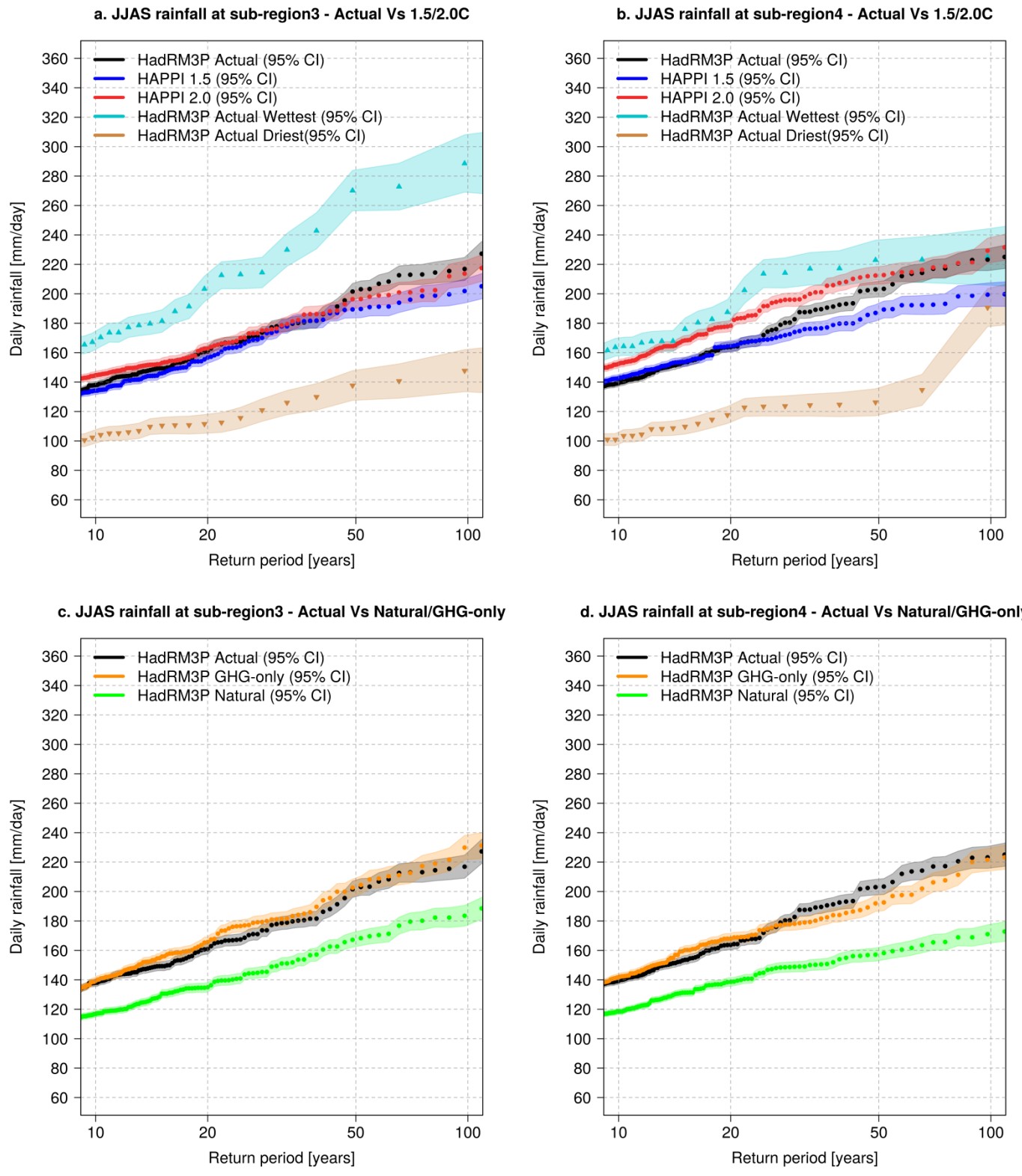

**Figure 9. Same as Fig 8, but showing return time plots for daily rainfall during monsoon (JJAS) season in different forcing scenarios over the sub-regions of 3 and 4 of Bangladesh.**

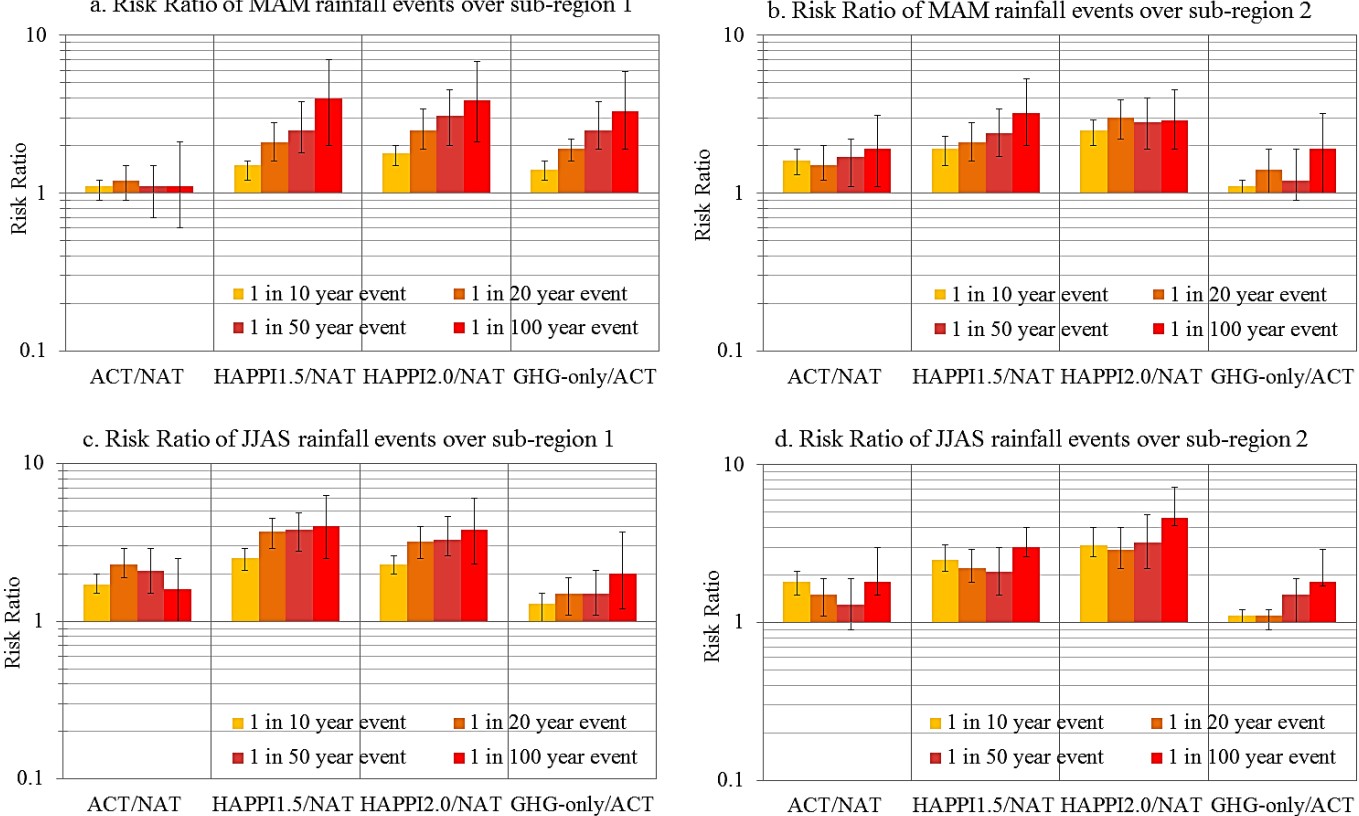

**Figure 10. The risk ratios of four specific rainfall events with return periods of 10, 20, 50, and 100 years between ACT/NAT, HAPPI 1.5/NAT, HAPPI 2.0/NAT and GHG-only /ACT over the two northern sub-regions 1 and 2 during MAM (top two panels of a. & b.) and JJAS (bottom two panels of c. & d.). The error bars indicate the associated uncertainty range with 95% confidence level for individual event.**

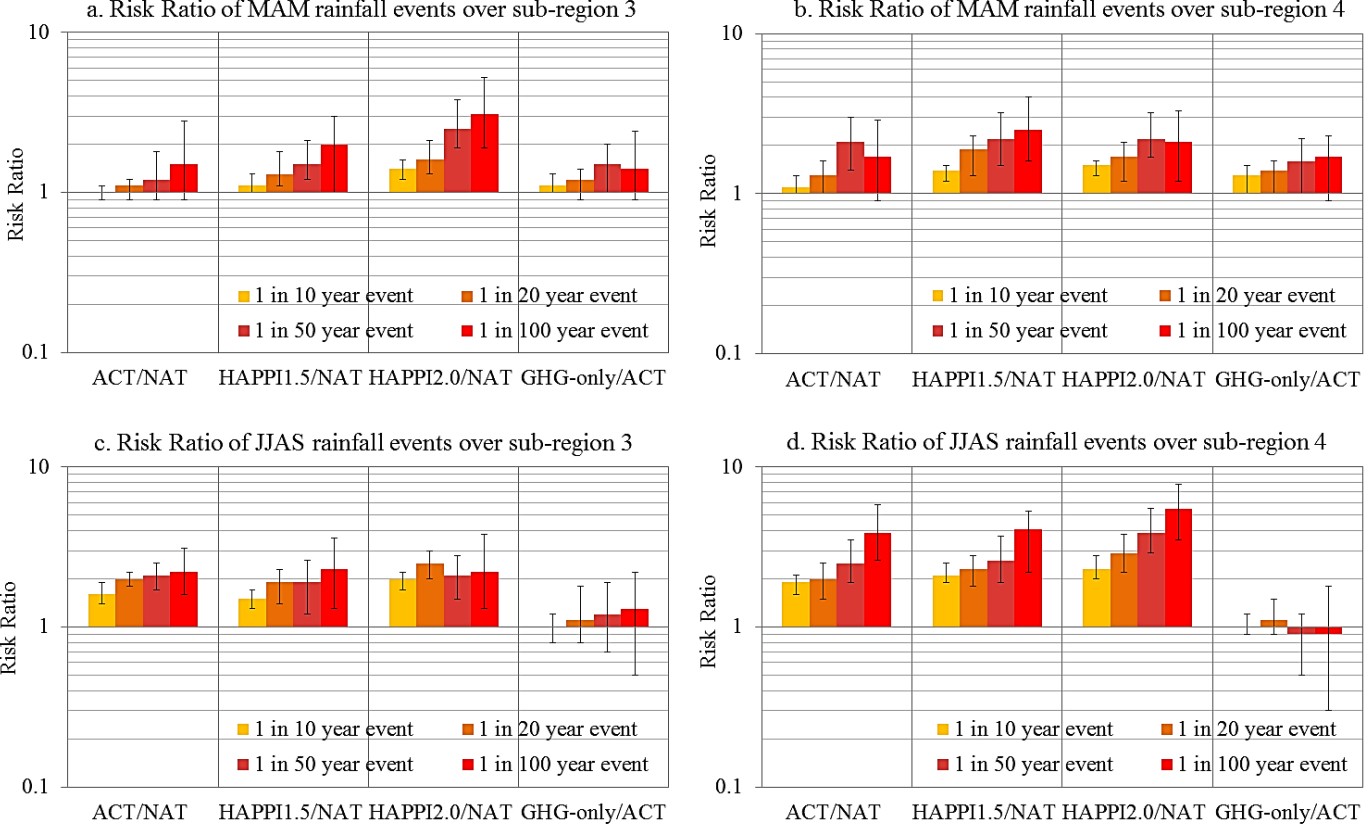

**Figure 11. Same as Fig 10, but for MAM and JJAS risk ratios over the two southern sub-regions 3 and 4.**

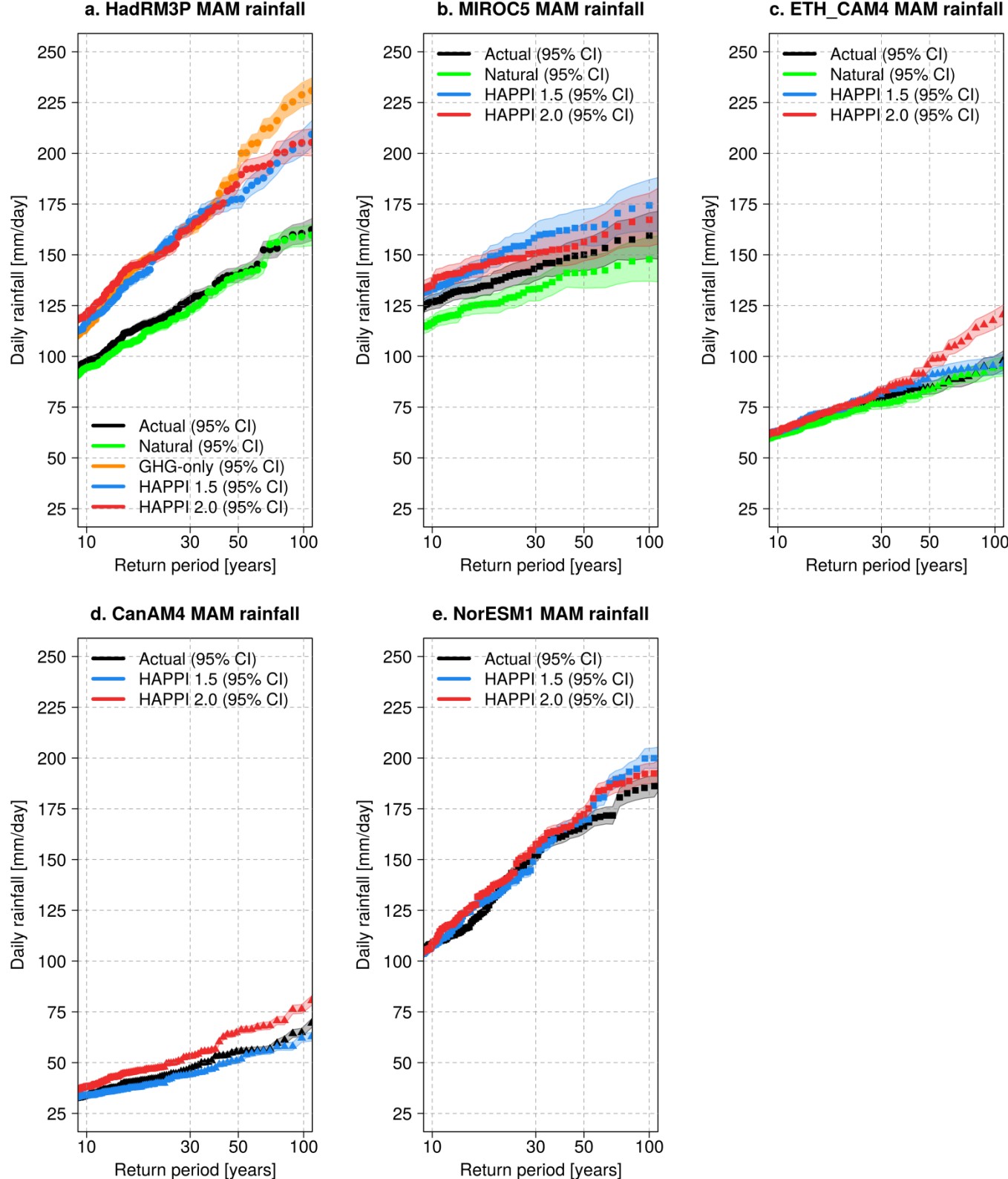

**Figure 12. Comparative return periods (10-100 year events) of MAM daily rainfall (mm/day) over sub-region 1 during 1986-2015 as per (a) HadRM3P, (b) MIROC5, (c) ETH_CAM4, (d) CanAM4 and (e) NorESM1 models. ACT, NAT, GHG-only, plus 1.5°C and 2.0°C model ensembles are shown in black, green, orange, blue and red colours respectively.**

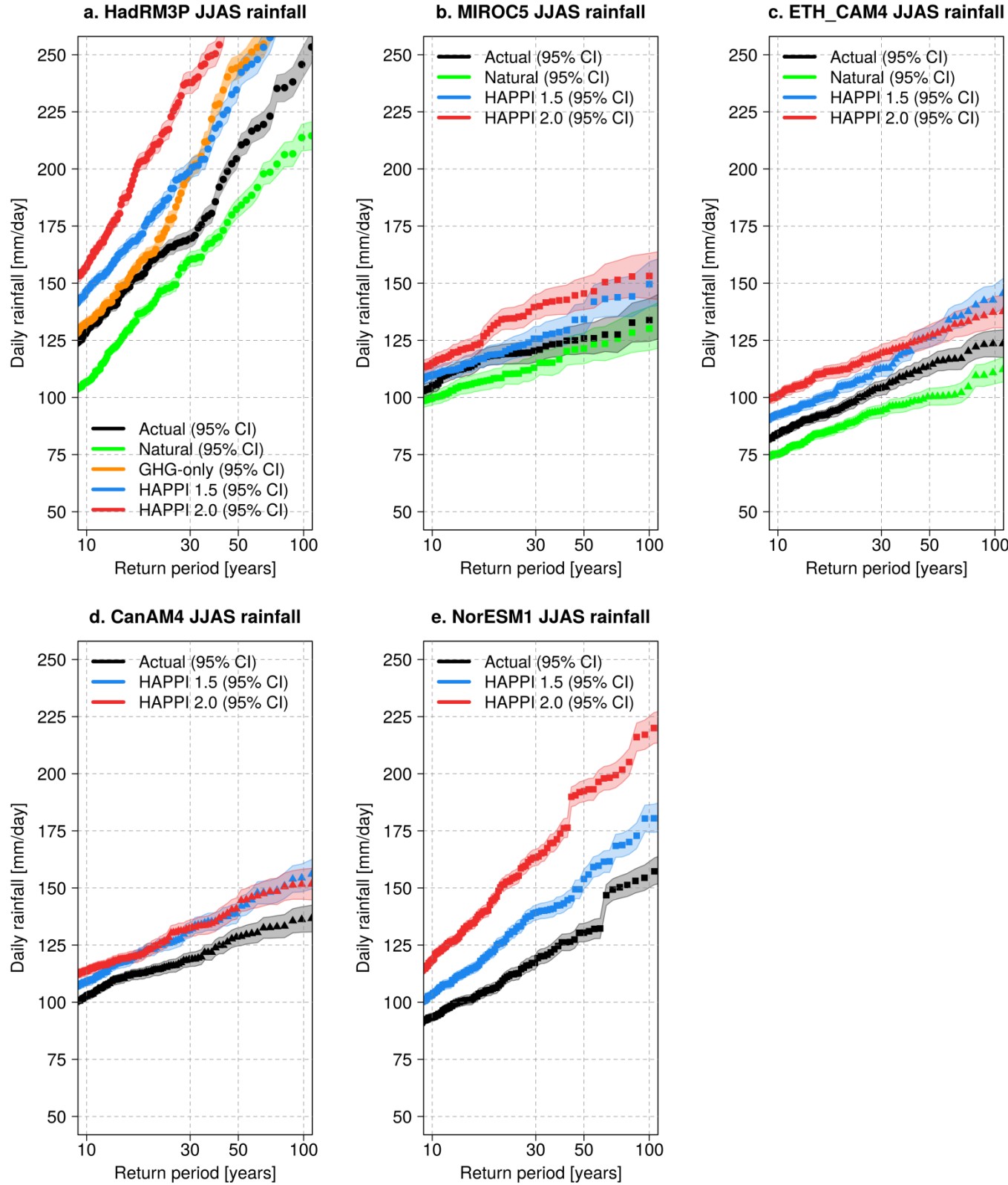

**Figure 13. Comparative return periods (10-100 year events) of JJAS daily rainfall (mm/day) over sub-region 2 during 1986-2015 as per (a) HadRM3P, (b) MIROC5, (c) ETH_CAM4, (d) CanAM4 and (e) NorESM1 models. ACT, NAT, GHG-only, plus 1.5°C and 2.0°C model ensembles are shown in black, green, orange, blue and red colours respectively.**

