# Peer review of "Risks of seasonal extreme rainfall events in Bangladesh under 1.5 and 2.0 degrees' warmer worlds – How anthropogenic aerosols change the story."

_Hydrology and Earth System Sciences, 2018_

## Referee Comment (RC1) · Anonymous Referee #1 · 1 Dec 2018

**Risks of seasonal extreme rainfall events in Bangladesh under 1.5 and 2.0degrees' warmer worlds – How anthropogenic aerosols change the story by Rimi et al.**

**Referee comments**

**Introduction**

The introduction provides a general background on the research topic with clearly stated objectives and research question. The author divided whole Bangladesh into four sub-regions (page 3, line11-14) to evaluate the risk of extreme rainfall events. Are there other specific reasons for to divide into four reasons like climatological variation or any other previous study used these sub-regions. A standalone Figure of Bangladesh including the four sub-regions could be useful with mean seasonal rainfall. Line 16 page 2 – In June 2017, heavy rainfall killed at least 156 people (needs a citation).

**Data and Methods**
**Model setup and experiment design**

The wettest and driest years are not well presented in the Table S2. Is it range of years or individual year? Classification of wet years and dry is not clearly mentioned. It has been mentioned that spatiotemporal average of rainfall has been used here. Is there any threshold for the classification of wet period and dry period? In Bangladesh flooding years are considered as wet year during monsoon. This need to be made clear.

**Results and Discussion**
**Section 3.1 Model Evaluation for Five day mean rainfall**

"Five day mean rainfall is used to represent the timescale responsible for river flooding as opposed to daily extremes that cause flash floods primarily in the pre-monsoon season." Is the 5 days rainfall causes flooding or 1-day extreme rainfall causes flash flood in Bangladesh and what is the intensity of rainfall termed as extreme (what is the amount of rainfall mm/day considered as extreme value)? Citation may clear this statement.

Fig.1 represents annual cycle of the four sub-regions in Bangladesh and results of the five models (ACT, NAT, GHG, HAPPI 1.5 and HAPPI 2.0) show maximum rainfall occurs in June. June to September is the monsoon month and June is the month of monsoon onset. Usually, July is the maximum rainfall month in Bangladesh. Do the results indicate any shifting of monsoon timing Due to the monsoon climate, the overall variation(interannual) of rainfall in JJAS months (seasonal) is not quite high. Bangladesh has almost similar pattern of monsoon

precipitation in the JJAS months. Underestimation by 25-65% is quite high. The bias and uncertainty within these values is very high. The authors need to explain the reasoning for this a bit more.

**Section 3.2: Impact of Climate Change and Aerosol Reduction on Seasonal Mean Rainfall**
Provides important information regarding rainfall change due to warming 1.5˚C to 2˚C and aerosol impact. However, the change has been computed using model based on simulated observed data. The actual changes can be presented by using observation data (e.g., Aphrodite).

**Page 7 line23 -24**

"While aerosol effects are consistent with other regions, the GHG induced rainfall is hampered, likely due to dynamic changes such as a delayed onset of the monsoon in response to warming", It can be supported with other relevant studies (e.g. the variation of interannual rainfall may depend on the onset of monsoon).

**3.3 Rainfall extreme:**
**Line 40, papge7**
"The signal-to-noise ratio is higher in the monsoon season across all sub-regions with the lowest and highest ratio in sub-region 1 and 3, respectively (Figs. 8a & 9a)". This statement may be needed further explanation.

**Reference**
The author referred Banglapedia, 2012 as citation in page 2 line 10. However, did not provide in the reference list.

**Other comments:**

(a) The title of the paper says risks of seasonal extreme rainfall events and presented rainfall extreme using daily and five day mean rainfall. One day max and 5day max would be better presentation of rainfall extreme. It is also necessary to have a better description why daily and five day mean rainfall has been used.

(b) Inconsistent in figure indexing spacing: In the results (e.g. Sect. 3.2) it is needed to be consistent with spacing when referencing to figures. For example, Fig.2 d and (Fig. 2d), (Figs. 8a & 8b) and (Fig. 4 a & c).

(c) The Figure captions are too long. The author started to describe the results in some of them (e.g. Fig. 5). The results or discussion should be in text. Caption should be concise and just define what the Figure shows with the necessary information to gather information from it.

(d) Line 6 (page 4) – Evaluation of the model for the region was conducted by Rimi et al. (under review) and demonstrated a reasonable agreement between model results and observational datasets for extreme rainfall events.

What is a reasonable agreement? Which statistical skills show general agreement (e.g. $r^2$, KGE). For example, 60% of stations achieved values greater than 0.6 between modelled and observed data.

The discussion article '**Risks of seasonal extreme rainfall events in Bangladesh under 1.5 and 2.0degrees' warmer worlds – How anthropogenic aerosols change the story**' by Ruksana H. Rimi et al. is very interesting focused on the extreme rainfall events due climate change particularly 1.5 and 2 degrees warmer world. This is a comprehensive analysis of future projection of multi model rainfall over several sub-region of Bangladesh. The author provides sufficient graphs and maps in the article which explained the results. The major findings of the article are related with the global warming and its implication extreme weather events for Bangladesh.

Finally, I suggest that the author will consider the above comments in finalizing the script. The article is recommended to publish with minor correction.

---

## Referee Comment (RC2) · Anonymous Referee #2 · 14 Dec 2018

The authors investigate changes in mean and extreme precipitation in Bangladesh for five different forcing scenarios. They divide Bangladesh in four regions and analyse the magnitude of events with different return times. They find increases in pre-monsoon and monsoon precipitation due to higher global mean temperatures but also due to a decrease in aerosols.

While the paper is clear in scope and the analysis in principle straightforward, it needs more work, especially the text. As crucial information is missing from the methods section I can only recommend publication of the paper after major revisions.

[Figure]

**General & Technical Comments**

**Text**

* The text could profit from more work. Some sections seem rather long, while others miss some essential information (see below). Also, there are numerous small mistakes that give the impression of sloppy proofreading. The conclusion, on the other hand are very well written and concise.

* It is very hard to really pin down, but I had the impression that some information is is repeated over and over again, see e.g. my comment concerning the abbreviations below. Another example is the first two sentences in your introduction - they are different but they basically say the same. Of course, sometimes it is good to repeat things (e.g. in the conclusions), I felt it rather hindered the flow while reading.

* When you use the word significant do you mean 'statistically significant' or large? Significant is a reserved word – please only use it if you conducted a statistical test!

* Similarly, you have to be careful when using the word 'risk'. Risk is often formalized as the combination of exposure and vulnerability to weather and climate events. Therefore please check if 'probability' of 'magnitude' would be more appropriate.

* You introduce abbreviations for the simulations, but you often refer to the simulations by the full name; e.g. on P6: L11 (twice); L15, L16, L21; L30 (twice). There are many more examples throughout the manuscript.

* What is the abbreviation for the simulation with current-day GHGs and pre-industrial aerosols? GHGonly? GHG? GHG only? AR? All of them are used throughout the manuscript. Be consistent! Also make sure that it can be differentiated from the abbreviation for greenhouse gas (GHG).

**Methods**

* You should explicitly write why you use different time periods for your two observa-

tional datasets.

* You use bi-linear interpolation to regrid your data. For the future I would recommend to use a conservative remapping scheme to make sure the precipitation amount is conserved.

* The description of the model simulations is very long and overly detailed. I recommend to shorten it.

* On the other hand, I miss a description of your statistical analysis, which makes it difficult to assess it. In particular I need the following questions answered: - How do you calculate the return time? How do you calculate the uncertainty of the return time? - For the RR, do you assume an Extreme Value distribution? If not how is the probability calculated? - How do you calculate the uncertainty of RR?

* Do you consider all days or only days with rainfall larger a threshold? This is particularly relevant for MAM.

* Given that you have data from 98 * 10 years, should your maximum return time not be at 980 years instead of 1000 as in Figure 6?

* Similarly, when analysing the two wettest/ driest years should you not end at 196 years instead of 200?

* You might want to mention that you look at SPI in the methods.

**Results**

* The two observational datasets show quite some differences. Do you have an guess if this is due to the different periods they span, or would they also be different covering the same periods? How different are they during the two overlapping years?

* What is the point of using the two lowest and highest years? It basically says 'during a wet year the magnitude of a 1 in X years event is larger than during an average year', which is a relatively trivial result. Also, these results are barely used/ described in the
results sections. My proposition would be to either remove this entirely, or move it to the appendix. This would help to clean up the figures and/ or reduce the amount of required subplots.

* Evaluation is done with 5-day precipitation but return time plots and RR in the main text are with 1-day precipitation – why is that?

* You sometimes talk about a linear rainfall response. What do you mean with linear? How can you know it's linear (as you have only 5 data points).

**Figures**

* I had to rasterize the Figures in order to print them. Not sure what the problem is but please make sure this does not happen for the final paper.

* However, even in your original pdf the figures are all blurry and don't have a good quality, this makes them very difficult to analyse. Please save them as *.pdf and ensure fonts are embedded.

* Avoid mixing green and red in the figures.

* The text (labels, legend) is generally too small.

* The caption are very long and describe results. Please make them shorter and move all results to the main text.

* The naming of simulations is inconsistent in the legend/ labels. E.g. in Figure 1 you call it 'HadRM3P ACT' but 'HAPPI 2.0'. I recommend to remove 'HadRM3P'. In Figure 2 and 3 it is called present instead of ACT. In Figure 4 you introduce a new abbreviation!

* In the same category: sometimes it is called MAM and sometimes pre-monsoon

**Minor Comments**

P1 L11: the future

P1 L15: risk -> probability
P1 L16: 2°C global warming

P1 L20: risk -> probability or magnitude

P1 L23: in terms . . . impact. -> remove, you do not look at impacts

P1 L25: GHG abbreviation not introduced

P2 L12: Is it really an increasing trend – do you mean a positive trend?

P2 L14: names -> name

P2 L15: Currently the sentence reads as if the "low-lying areas" damaged the rice.

P2 L15: Should that be "Boro (...) and paddy crops"? Or is should it be "Boro paddy crops (...)"?

P2 L16: Which dataset is this?

P2 L25: remove 'multi'. Also I would recommend to rewrite this sentence.

P2 L25: for the northwestern part of

P2 L27: the high resolution

P2 L28: in the global

P2 L31: north-eastern

P2 L34: remove 'of'

P2 L35: Even if they did not calculate RRs other studies did look at the influence of anthropogenic climate change -> please reformulate

P2 L41: runs -> simulations

P2 L41: remove 'the'

P3 L4-L5: also . . . warming: I don't understand what you mean here.

P3 L10: remove 'here'

P3 L11-L14: I recommend to move this sentence to the methods.

P3 L18: 2.2 -> 3.1

P3 L20: You did not mention Section 3.2

P3 L27: remove the second 'observational'

P3 L30: remove 'grids'

P3 L31: in Table S1

P3 L39: Move the reference behind 'program'.

P3 L41: remove 'of'

P4 L2: remove 'the model'

P4 L5: GHG was introduced before

P4 L6-L9: This belongs in the Results Section.

P4 L10: remove 'the'

P4 L27: one third

P4 L29: remove 'world'

P4 L31: hereinafter?

P4: Formula (i) and (ii): These are unnecessary.

P4 L39: force -> forcing

P4 L41: the other GCMs -> other GCMs

P5 L5 and L6: I would write this as 3x30x10x98 and 4x30x10x98. In addition, make a remark that all months have 30 days in the used model.

[Figure]

P5 L15: presented -> present

P5 L15: This sentence does not make sense, please rewrite.

P5 L17: the probability

P5 L17: remove 'scenarios'

P5 L18: an event of the same magnitude

P5 L18-L20: Abbreviations!

P5 L26 and L34: annual -> seasonal

P5 L35: dataset it is compared with and sub-regions. -> dataset and the sub-region.

P5 L36: at -> in

P5 L36: There is no way to know if the bias is the same for the different scenarios! This is an assumption. It's ok to make the assumption, but 'note' is not the appropriate word here.

P6 L2: Significant?

P6 L6: although they suggest

P6 L28-L29: Did you show this somewhere in your paper? Do you have a reference for this?

P6 L34-L35: Is that global or local warming?

P6 41: Significant?

P7 L4: remove 'and hence efficiently masked'

P7 L6: SPI: abbreviation not introduced

P7 L15: in all -> in almost all

P7 L36: frequencies of occurrence -> magnitude

Figure 1

\* Annual -> seasonal

Figure 2 and Figure 3

\* I recommend to change the title to '(ACT - NAT) / NAT', '(HAPPI1.5 – ACT) / ACT', etc. so it is absolutely clear what you are doing.

\* The maps in the top row look a bit distorted – do you use a projection for the map plot?

\* Why is it only 'approximately' the sub-regions? Remove approximately.

Figure 6

\* 'sub-regions of 1 and 2' -> remove of

Figure 10 and 11

\* Please use a logarithmic y-axis so that the plot is symmetric with respect to 1.

\* Remove the bars, the RR is not really something that starts a 0.

\* Reverse the order of the bars, start with the 1 in 10 year event.

---

## Referee Comment (RC3) · Anonymous Referee #3 · 19 Dec 2018

This study focuses on the investigation of changes in total and extreme precipitation in Bangladesh due to changes in greenhouse gas and aerosol concentrations. Large ensembles of regional climate simulations are used to represent regional dynamics and aerosol effects with sufficient detail and at the same time obtain statistical robust results also for extreme events with long return periods. In my opinion, the research presented in this manuscript is generally sound and provides novel insights (although not outstandingly new/innovative) into future rainfall changes in a highly impact-relevant region. However, I think the presentation and discussion of the research needs a

substantial revision before the manuscript could be published. In my view, the interpretation of the results is too superficial at several places throughout the manuscript. Furthermore, the language and wording are not always adequate. With respect to the second point, I list a few issues below, but this list is not complete, and actually the native speakers among the co-authors should be able to fix this in a better way than I am.

Specific comments:

Page 1 Line 22: As this is a model study, I'd avoid the term "impacts were observed"

P 1 L 28: "specifically with respect to...": I was confused when reading this as I though the whole study would focus on extreme events. It is not clear from the abstract that also seasonal mean rainfall is analyzed.

P 2 L 16: Nirapad (2017) is not in the list of references

P 2 L 23: "help to provide...": wording issue

P 3 L 12-14: I think these detailed information regarding the sub-regions would fit better in the methods section.

P 3 L27: "observational" appears too often in this sentence

P 3 L 35: This is my only purely methodological comment: I am a bit skeptical with respect to the usage of bi-linear interpolation, as this does not conserve the area-average rainfall amount and also biases the extremes compared to the original grid point values. I would ask the authors to at least test the sensitivity of their approach using a more appropriate conservative interpolation method (see, e.g., Chen and Knutson, 2008, doi:10.1175/2007JCLI1494.1).

P 4: The fact that some experiments are mentioned twice (in the first paragraph and further below) leads to some repetitions.

P 4 L 16: I'd mention already here in which way the ensemble members differ from

each other.

P 4 L 33-34: This notation is awkward. I'd either write this in text form or as a "real" equation, but not mix these things up.

P L 12-15: This description of the aerosol affect is too short and not very clear. The term "omitted aerosol induced rainfall" should be explained. I am also confused by the sign of the signal and the figure caption: The caption of Fig. 2 says that the figure shows present-day relative to GHG only; positive values would thus mean that the present-day rainfall including the aerosol effect is larger than the rainfall due to GHG only, which is not consistent. Finally, before directly linking this result to potential future decreases in the aerosol effect already in the second sentence, the actual content of the figures should be described and explained.

P 6 L 19-28: I think this whole discussion is too superficial. There are many speculations on how thermodynamic and dynamic effects could influence the precipitation changes which, in my view, are speculative and should be based on a more quantitative and solid analysis. For instance, I am not sure how an "approximately linear" scaling is deduced from the data presented in this study. If this just refers to the differences between the 1.5 and 2.0 simulations, I could well imagine a case in which precipitation increases due to both increase in the atmospheric moisture content and in the monsoon circulation, and this increase is amplified in the 2.0 case, which may also produce a linear change over these simulations. Moreover, is a linear scaling really what we expect thermodynamically? The Clausius-Clapeyron relation is non-linear. The conclusion that dynamic changes play a secondary role should be manifested in a quantitative way. Also the statement that the thermodynamic response "usually scales with 20-40% of Clausius Clapeyron" is vague and, as such, not comprehensible.

P 7 L 1-2: I cannot follow here: In the region with the strongest decrease in Fig. 3a, the aerosol effect is small.

P 7 L6: The abbreviation "SPI" has not been introduced.

P 7 L 11-3: Again I cannot follow. For instance, the changes in 5b,d are similar to those in 4b,d

P 7 L 14: It is difficult to assess the relative change at this point, as the figures show absolute values.

P 7 L 16-17: This doesn't fit with the interpretation in the figure caption. In general, I find it difficult to shift parts of the discussion to the figure captions.

P 7 L 18: lapse rate and stability changes are not different feedbacks

P 7 L 21: Again I don't understand what "linear" increase means here.

Section 3.3, first paragraph: There is an imbalance between the amount of text/discussion and the number of figures. The reader is left alone with much of the material shown in Figs. 6-9. Either expand this discussion, or, if you think that the results are not that important, move parts of the figures to the supplement.

P 8 L 4: "appear to counter": I cannot see how you come to this conclusion.

P 8 L 24: "to a lesser extent": Really? Aren't the relative changes larger for the extremes?

P 8 L 31: "we conclude that the drier subregions...": I don't think this has been demonstrated. To show this, the masking effect has to be quantified. Furthermore, it is not clear to me which region and season you're referring to.

Fig. 1: I think this figure is too busy. I cannot distinguish the different shadings and also the lines of the observations are hard to see.

Fig. 3: shorten caption (as Fig. 2, but for the monsoon season)

Fig. 4: I cannot follow the interpretation in the caption. For instance, I don't see such large differences in the masking effect between the regions. More in general, it is hard for me to understand how the masking effect is quantified here.

Fig. 10: Again, the interpretation in the caption is unclear (e.g., which region and season are you referring to?)

[Figure]

---

## Author Comment (AC1) · 2 Jul 2019

We thank for the constructive comments from the anonymous Reviewer 1. We have carefully revised the manuscript to incorporate necessary amendments as per suggestions. Responses to the Referee Comments 1(RC1) are presented in the following paragraph in the Author's Comments 1(AC1):

Introduction

RC1: The introduction provides a general background on the research topic with clearly

stated objectives and research question. The author divided whole Bangladesh into four sub-regions (page 3, line11-14) to evaluate the risk of extreme rainfall events. Are there other specific reasons for to divide into four seasons like climatological variation or any other previous study used these sub-regions.

AC1: These 4 sub-regions of Bangladesh and the different seasons are used for a reason. The three wet seasons with substantial climatological variations include: pre-monsoon (during Mar-Apr-May; MAM), monsoon (during Jun-Jul-Aug; JJA) and post-monsoon (Sep-Oct-Nov; SON). Little or no rain occurs in winter (during Dec-Jan-Feb; DJF). This dry winter season is excluded from our analysis because we are interested in wet extremes. Any MAM extreme rainfall events are known to cause flash floods and substantial crop damage (Ahmed et al., 2017). Bangladesh receives more than 75% of the annual total rainfall during JJAS (Shahid, 2010). An extreme rainfall event in this period can therefore cause wide-spread flooding and landslide eventually leading to loss of lives and livelihoods. A high impact SON rainfall event may be associated with the coastal floods that occur due to storm surges or tidal effects along the northern part of Bay of Bengal (Hossain, 1998). Considering both meteorological hazards and potential impacts, we have looked at extreme rainfall events of pre-monsoon and monsoon seasons and excluded the post-monsoon season. The same 4 sub-regions are used in Rimi et al., (2019) where, we evaluate the model's performance in simulating extreme events (see Fig. AC1.a). We believe that the pre-evaluated model simulations over the same 4 sub-regions provide confidence in analyzing the extreme rainfall events under different forcing scenarios.

RC1: A standalone Figure of Bangladesh including the four sub-regions could be useful with mean seasonal rainfall.

AC1: Because a standalone figure of Bangladesh including the four sub-regions with mean seasonal rainfall has already been shown in Rimi et al., 2019 (see Fig AC1.b); we are not using the same kind of figure in this paper. Nevertheless, the sub-regions are indicated in panel e of Fig 2 & 3 of this manuscript.

RC1: Line 16 page 2 – In June 2017, heavy rainfall killed at least 156 people (needs a citation).

AC1: The following citation is added for this information: Paul, R. and Hussain, Z.: Landslide, floods kill 156 in Bangladesh, India; toll could rise, Reuters, 14th June [online] Available from: https://uk.reuters.com/article/uk-bangladesh-landslides/landslide-floods-kill-156-in-bangladesh-india-toll-could-rise-idUKKBN1950AG, 2017.

Data and Methods Model setup and experiment design

RC1: The wettest and driest years are not well presented in the Table S2. Is it range of years or individual year? Classification of wet years and dry is not clearly mentioned. It has been mentioned that spatiotemporal average of rainfall has been used here. Is there any threshold for the classification of wet period and dry period? In Bangladesh flooding years are considered as wet year during monsoon. This needs to be made clear.

AC1: We identified the two wettest and two driest years during 2006–2015 over each of the four sub-regions of Bangladesh using ACT data. This identification process involved comparing the magnitude along with return periods of rainfall events during MAM and JJAS in each of the years throughout 2006 to 2015 (ACT model ensemble with 200 members per year used). Then a pair of wettest and driest years is used to approximately indicate the noise-to-signal ratio. For example, according to ACT model ensemble, during monsoon season at sub-region 2, the wettest years with extreme rainfall events are 2008 and 2012 (see Fig. AC1.c, the red and blue dots for individual model runs and shadings for 5-95% confidence intervals) and the driest years are 2006 and 2013 (yellow and cyan dots for individual model runs and shadings for 5-95% confidence intervals). Using the pair of these wettest and driest years from the model ensemble, we could demonstrate the plausible spread of the rainfall events within this model ensemble as a measure of noise-to-signal ratio. Because this model ensemble has same forcings in each year for the historical period of 2006-2015, the only variability

playing a role in changing the intensity of rainfall is natural variability of sea surface temperatures (SSTs). The use of two wettest and driest years, therefore explains how much natural variability of SSTs contributes to the variability of rainfall intensities over this study area. We agree with your comment and changed the presentation of the years in the supplementary Table 2 to make it clear that they are the two individual years with either wettest or, driest conditions in a 10-year period over a specific sub-region.

Results and Discussion Section 3.1 Model Evaluation for Five day mean rainfall

RC1: "Five day mean rainfall is used to represent the timescale responsible for river flooding as opposed to daily extremes that cause flash floods primarily in the pre-monsoon season." Is the 5 days rainfall causes flooding or 1-day extreme rainfall causes flash flood in Bangladesh and what is the intensity of rainfall termed as extreme (what is the amount of rainfall mm/day considered as extreme value)? Citation may clear this statement.

AC1: By extreme rainfall event, we mean a high impact rainfall event (i.e., sufficient to cause flooding or landslides) with up to 100 year return period (a rare event with low frequency of occurrence). The intensity of the rainfall event can vary depending on both location and season. For example, at north-east Bangladesh, in pre-monsoon season, more than 150 mm rainfall event over a 6-day period can lead to an early flash flood (Ahmed et al., 2017). On the other hand, at south-east parts of Bangladesh, in monsoon season, more than 350 mm rainfall events over a 3-day period is enough to cause a landslide (Ali et al., 2014). Whereas, for a wide-spread river flooding e.g., the Brahmaputra River Basin flooding in August 2017, 10-day extreme rainfall event is considered (Philip et al., 2018). Considering the range from 1- to 10-day high impact rainfall events that can trigger flooding and landslides in Bangladesh, we focused on daily and 5-day events because these events can provide a typical idea about the potential risks.

RC1: Fig.1 represents annual cycle of the four sub-regions in Bangladesh and results of the five models (ACT, NAT, GHG, HAPPI 1.5 and HAPPI 2.0) show maximum rainfall occurs in June. June to September is the monsoon month and June is the month of monsoon onset. Usually, July is the maximum rainfall month in Bangladesh. Do the results indicate any shifting of monsoon timing Due to the monsoon climate, the overall variation (inter-annual) of rainfall in JJAS months (seasonal) is not quite high? Bangladesh has almost similar pattern of monsoon precipitation in the JJAS months. Underestimation by 25-65% is quite high. The bias and uncertainty within these values is very high. The authors need to explain the reasoning for this a bit more.

AC1: No, as per observational data (APHRODITE and CPC) that are used in the annual cycles in Fig 1; there are no indications of temporal shifting of monsoon. We see an early monsoon onset in the model simulations. Such early onset of monsoon is also reported in other model based studies (e.g., in Caesar et al., 2015; Fahad et al., 2017; Janes and Bhaskaran, 2012). At the time of writing this paper, APHRODITE was only available until 2007, so we used 1998-2007 data for comparison. Recently, APHRODITE has been updated until 2015, allowing us to use 2006-2015 data for comparison (which is now same as CPC). As a result, APHRODITE and CPC observations are now in better agreement (see Fig. AC1.d) and model bias is also smaller than before (highest bias level of 65% dropped to 50%). The 25-50% underestimation of monsoon precipitation is quantified based on the model ensemble mean compared to observations. Most of the observed rainfall is found to be within the 10-90% confidence interval of the model data. Overall, the weather@home model simulates the annual cycle of rainfall with satisfactory agreement, despite the dry monsoon rainfall bias. Furthermore, Rimi et al. (2019) shows that the weather@home model gives a reasonable representation of extreme rainfall events with return periods of 50-100 years. Therefore, we are confident that these biases will not affect our risk assessments for extreme rainfall events.

Section 3.2: Impact of Climate Change and Aerosol Reduction on Seasonal Mean

Rainfall

RC1: Provides important information regarding rainfall change due to warming 1.5ËŽC to 2ËŽC and aerosol impact. However, the change has been computed using model based on simulated observed data. The actual changes can be presented by using observation data (e.g., Aphrodite).

AC1: While it is definitely useful to look at the observed changes between pre-industrial and present-day rainfall, present day and future warmer (1.5 and 2.0 degrees) scenarios or the aerosol impacts; we can only do this kind of comparison using model simulated data. Because no high resolution gridded observational dataset offers the pre-industrial records and the future scenarios of 1.5 and 2.0 degrees warming. APHRODITE data was only available from 1963 to 2007 at the time of writing this paper. Recently they have updated their data up to 2015. CPC observation dataset extends from 1979-present.

Page 7 line23 -24

RC1: "While aerosol effects are consistent with other regions, the GHG induced rainfall is hampered, likely due to dynamic changes such as a delayed onset of the monsoon in response to warming", It can be supported with other relevant studies (e.g. the variation of interannual rainfall may depend on the onset of monsoon).

AC1: While it is beyond the scope of this study to identify the exact mechanisms that lead to future changes associated with aerosol removal (or the change due to contemporary emissions for that matter), we point to the literature where this issue has been investigated in some detail already. The seminal paper by Bollasina et al. (2011), as well as more recent work by Zhao et al. (2019) are excellent resources that support our point.

3.3 Rainfall extreme: Line 40, page7 RC1: "The signal-to-noise ratio is higher in the monsoon season across all sub-regions with the lowest and highest ratio in sub-region

1 and 3, respectively (Figs. 8a & 9a)". This statement may be needed further explanation.

AC1: This statement is rewritten as "Overall, the signal-to-noise ratio is higher across all sub-regions, during JJAS compared to that during MAM. During MAM, the highest and lowest signal-to-noise ratio is over sub-region 1 and 3, respectively (Figs. 6a & 7a). On the other hand, during JJAS, we find the highest and lowest signal-to-noise ratio is over sub-region 3 and 1, respectively (Figs. 9a & 8a). The lower the ratio, the more difficult it is to establish causality as natural variability due to ENSO or circulation anomalies is higher."

References

RC1: The author referred Banglapedia, 2012 as citation in page 2 line 10. However, did not provide in the reference list.

AC1: We have now added this citation as "Banglapedia: River and Drainage system, Banglapedia- Natl. Encycl. Bangladesh [online] Available from: http://en.banglapedia.org/index.php?title=River_and_Drainage_System, 2012." Other comments:

(a) The title of the paper says risks of seasonal extreme rainfall events and presented rainfall extreme using daily and five day mean rainfall. One day max and 5day max would be better presentation of rainfall extreme. It is also necessary to have a better description why daily and five day mean rainfall has been used.

AC1: This is because our focus is on unusual rare rainfall events with the potential to cause high impacts on the ground in pre-monsoon and monsoon seasons. We have used daily and 5-day running mean rainfall events throughout a season and then looked at events crossing a threshold (e.g. 250mm/day that can trigger floods or, landslides) with a high return period (e.g. 100 years). In particular, we have explored whether or not, and to what extent, the probability of having that same magnitude event (i.e.

250mm/day) changes in that particular season from one forcing scenario to another (e.g. from ACT to HAPPI 1.5).

(b) Inconsistent in figure indexing spacing: In the results (e.g. Sect. 3.2) it is needed to be consistent with spacing when referencing to figures. For example, Fig.2 d and (Fig. 2d), (Figs. 8a & 8b) and (Fig. 4 a & c).

AC1: Figure indexing spacing issue has been resolved. Now this is uniformly done throughout the manuscript using Fig. 1a; Figs. 1a & b; Figs. 1a-d and Figs. 1b & 2b style.

(c) The Figure captions are too long. The author started to describe the results in some of them (e.g. Fig. 5). The results or discussion should be in text. Caption should be concise and just define what the Figure shows with the necessary information to gather information from it.

AC1: We have now moved parts of the figure captions to result and discussion section to reduce the length, and make them concise.

(d) Line 6 (page 4) – Evaluation of the model for the region was conducted by Rimi et al. (2019) and demonstrated a reasonable agreement between model results and observational datasets for extreme rainfall events. What is a reasonable agreement? Which statistical skills show general agreement (e.g. r2, KGE). For example, 60% of stations achieved values greater than 0.6 between modelled and observed data.

AC1: We present here Fig. AC1.a adapted from Rimi et al., (2019) for reference. In Figure AC1a., the black line represents the model simulated rainfall events; while, red, blue, orange and sky-blue colours indicate APHRODITE, GPCC, CPC gridded observations and TRMM satellite data, respectively. Based on these figures, Rimi et al., (2019) reports that the pre-monsoon and monsoon extreme rainfall events with up to 100 years of return periods are adequately well captured by the model over Bangladesh when compared to APHRODITE, GPCC, CPC gridded observations and TRMM satellite data. Although the observation data sets used in that model evaluation paper had short lengths of records; by fitting a Generalized Extreme Value distribution, the authors demonstrated that the model simulated extreme events are in good agreement with the observations.

RC1: The discussion article 'Risks of seasonal extreme rainfall events in Bangladesh under 1.5 and 2.0degrees' warmer worlds – How anthropogenic aerosols change the story' by Ruksana H. Rimi et al. is very interesting focused on the extreme rainfall events due climate change particularly 1.5 and 2 degrees warmer world. This is a comprehensive analysis of future projection of multi model rainfall over several sub-region of Bangladesh. The author provides sufficient graphs and maps in the article which explained the results. The major findings of the article are related with the global warming and its implication extreme weather events for Bangladesh. Finally, I suggest that the author will consider the above comments in finalizing the script. The article is recommended to publish with minor correction.

AC1: The authors highly appreciate your careful review with constructive comments. We have updated the manuscript following most of your suggestions. In case of any disagreement, we have provided our explanation behind that. We hope that now the manuscript is ready to be accepted for publication.

References Ahmed, M. R., Rahaman, K. R., Kok, A. and Hassan, Q. K.: Remote sensing-based quantification of the impact of flash flooding on the rice production: A case study over Northeastern Bangladesh, Sensors (Switzerland), 17(10), doi:10.3390/s17102347, 2017.

Ali, R. M. E., Tunbridge, L. W., Bhasin, R. K., Akter, S., Khan, M. M. H. and Uddin, M. Z.: Landslides susceptibility of chittagong city, bangladesh and development of landslides early warning system, in Landslide Science for a Safer Geoenvironment, vol. 1, pp. 423–429., 2014.

Bollasina, M.A., Ming, Y. and Ramaswamy, V.: Anthropogenic Aerosols and the Weakening of the South Asian Summer Monsoon, Science 334 (6055), 502-505, doi: 10.1126/science.1204994, 2011. Caesar, J., Janes, T., Lindsay, A. and Bhaskaran, B.: Temperature and precipitation projections over Bangladesh and the upstream Ganges, Brahmaputra and Meghna systems, Environ. Sci. Process. Impacts, 17(6), 1047–1056, doi:10.1039/C4EM00650J, 2015.

Fahad, M. G. R., Saiful Islam, A. K. M., Nazari, R., Alfi Hasan, M., Tarekul Islam, G. M. and Bala, S. K.: Regional changes of precipitation and temperature over Bangladesh using bias-corrected multi-model ensemble projections considering high-emission pathways, Int. J. Climatol., doi:10.1002/joc.5284, 2017.

Hauser, M., Gudmundsson, L., Orth, R., Jézéquel, A., Haustein, K., Vautard, R., van Oldenborgh, G. J., Wilcox, L. and Seneviratne, S. I.: Methods and Model Dependency of Extreme Event Attribution: The 2015 European Drought, Earth's Futur., 5(10), 1034–1043, doi:10.1002/2017EF000612, 2017.

Hossain, A. N. H. A.: Flood management. [online] Available from: http://www.apfm.info/publications/casestudies/cs_bangladesh_sum.pdf, 1998.

Janes, T. and Bhaskaran, B.: Evaluation of regional model performance in simulating key climate variables over Bangladesh, Met Office, Exeter, United Kingdom., 2012.

Kumar, D., Arya, D. S., Murumkar, A. R. and Rahman, M. M.: Impact of climate change on rainfall in Northwestern Bangladesh using multi-GCM ensembles, Int. J. Climatol., 34(5), 1395–1404, doi:10.1002/joc.3770, 2014.

NAS: Attribution of Extreme Weather Events in the Context of Climate Change, National Academies Press., 2016.

Nirapad: Bangladesh: Flash Flood Situation - April 19, 2017, Situat. Rep., 7 [online] Available from: https://reliefweb.int/sites/reliefweb.int/files/resources/Flash_Flood%2C Updated %28April 19%29%2C 2017.pdf (Accessed 14 November 2017), 2017.

Philip, S., Sparrow, S., Kew, S. F., van der Wiel, K., Wanders, N., Singh, R., Hassan, A.,

Mohammed, K., Javid, H., Haustein, K., Otto, F. E. L., Hirpa, F., Rimi, R. H., Islam, A. S., Wallom, D. C. H. and van Oldenborgh, G. J.: Attributing the 2017 Bangladesh floods from meteorological and hydrological perspectives, Hydrol. Earth Syst. Sci. Discuss., 1–32, doi:10.5194/hess-2018-379, 2018.

Rimi, R. H., Haustein, K., Barbour, E. J., Jones, R. G., Sparrow, S. N. and Allen, M. R.: Evaluation of a large ensemble regional climate modelling system for extreme weather events analysis over Bangladesh, Int. J. Climatol., 39(6), 2845–2861, doi:10.1002/joc.5931, 2019.

Shahid, S.: Rainfall variability and the trends of wet and dry periods in Bangladesh, Int. J. Climatol., 30(15), 2299–2313, doi:10.1002/joc.2053, 2010.

Zhao, A.D., Stevenson, D.S. and Bollasina, M.A., The role of anthropogenic aerosols in future precipitation extremes over the Asian Monsoon Region, Climate Dynamics 52:6257-6278, doi: 10.1007/s00382-018-4514-7, 2019

[Figure]

**Fig. 1.** AC1.a: Left panels show 5-day rainfall events in JJA and MAM over Bangladesh while right panels show the same but for sub-region 2.

[Figure]

**Fig. 2.** AC1.b: Left panel shows South Asia domain of the weather@home regional climate model and right panel shows the four sub-regions of Bangladesh with APHRODITE based mean monsoon rainfall (mm/day).

[Figure]

**Fig. 3.** AC1.c: Return period plots for JJAS daily precipitation over sub-region 2 in different years (2006-2015).

[Figure]

**Fig. 4.** AC1.d: The left column shows the old version of the annual cycles of 5-day rainfall over the four sub-regions of Bangladesh. The right column shows the same but uses updated APHRODITE data.

---

## Author Comment (AC3) · 2 Jul 2019

This study focuses on the investigation of changes in total and extreme precipitation in Bangladesh due to changes in greenhouse gas and aerosol concentrations. Large ensembles of regional climate simulations are used to represent regional dynamics and aerosol effects with sufficient detail and at the same time obtain statistical robust results also for extreme events with long return periods. In my opinion, the research resented in this manuscript is generally sound and provides novel insights (although not outstandingly new/innovative) into future rainfall changes in a highly impact-relevant

region. However, I think the presentation and discussion of the research needs a substantial revision before the manuscript could be published. In my view, the interpretation of the results is too superficial at several places throughout the manuscript. Furthermore, the language and wording are not always adequate. With respect to the second point, I list a few issues below, but this list is not complete, and actually the native speakers among the co-authors should be able to fix this in a better way than I am.

We thank for the constructive comments from the anonymous Reviewer 3. We have carefully revised the manuscript to incorporate necessary amendments as per suggestions. Responses to the Referee Comments 3 (RC3) are presented in the following Author's Comments 3 (AC3):

RC3 Specific Comments

RC3: Page 1 Line 22: As this is a model study, I'd avoid the term "impacts were observed"

AC3: Rewritten the line as "Climate change impacts on the probabilities of extreme rainfall events are found during both pre-monsoon and monsoon seasons, but the level of impacts are spatially variable across the country."

RC3: P 1 L 28: "specifically with respect to...": I was confused when reading this as I though the whole study would focus on extreme events. It is not clear from the abstract that also seasonal mean rainfall is analyzed.

AC3: To make it clear that we have also looked at climate change impacts on seasonal mean rainfall, we have now added the following line in abstract: "Analysis of percent change, standardized precipitation index and absolute change in seasonal mean rainfall revealed that there both GHGs and anthropogenic aerosols play important roles in determining the overall climate change impact over this region."

RC3: P 2 L 16: Nirapad (2017) is not in the list of references

AC3: The reference for Nirapad (2017) is now added to the Bibliography.

RC3: P 2 L 23: "help to provide...": wording issue

AC3: Deleted 'to provide'

RC3: P 3 L 12-14: I think these detailed information regarding the sub-regions would fit better in the methods section.

AC3: Agreed and moved to method section.

RC3: P 3 L27: "observational" appears too often in this sentence

AC3: Rewritten as "The daily observational data sets that are used as a comparison against model results include: (i) Asian Precipitation Highly Resolved Observational Data Integration Towards Evaluation of Water Resources (APHRODITE) (Yatagai et al., 2012) and (ii) NOAA's Climate Prediction Center (CPC) global 0.5° analysis (Chen et al., 2008a)."

RC3: P 3 L 35: This is my only purely methodological comment: I am a bit sceptical with respect to the usage of bi-linear interpolation, as this does not conserve the area-average rainfall amount and also biases the extremes compared to the original grid point values. I would ask the authors to at least test the sensitivity of their approach using a more appropriate conservative interpolation method (see, e.g., Chen and Knutson, 2008, doi:10.1175/2007JCLI1494.1).

AC3: Thank you for the comment. We have checked our ACT precipitation data over Bangladesh in this regard. As per figure AC3.a, we can argue that changing the method from bilinear interpolation to conservative have no effect on the high intensity precipitation events.

RC3: P 4: The fact that some experiments are mentioned twice (in the first paragraph and further below) leads to some repetitions.

AC3: We have now aimed at avoiding repetitions in the manuscript as good as possible.

RC3: P 4 L 16: I'd mention already here in which way the ensemble members differ from each other.

AC3: The model ensemble members differ in their initial conditions. They are either slightly perturbed, or the atmospheric field to restart the next model run is slightly different (i.e. it originates from a different member that has been run earlier). In contrast to the scenarios, all forcing parameters are the same. As far as NAT, GHG-only and the HAPPI scenarios are concerned, we use 11 different delta SST pattern (prescribed SSTs to define the lower boundary conditions). Those 11 patterns correspond to the same forcing scenario in CMIP5 (where the delta SSTs are derived from), but they do show slightly different spatial SST anomalies and add therefore additional variability to the weather@home ensemble of the counterfactual and future model scenarios.

RC3: P 4 L 33-34: This notation is awkward. I'd either write this in text form or as a "real" equation, but not mix these things up.

AC3: Agreed, we have now removed the equations and only kept text to describe this model ensemble.

RC3: P L 12-15: This description of the aerosol affect is too short and not very clear. The term "omitted aerosol induced rainfall" should be explained. I am also confused by the sign of the signal and the figure caption: The caption of Fig. 2 says that the figure shows present-day relative to GHG only; positive values would thus mean that the present-day rainfall including the aerosol effect is larger than the rainfall due to GHG only, which is not consistent. Finally, before directly linking this result to potential future decreases in the aerosol effect already in the second sentence, the actual content of the figures should be described and explained.

AC3: The positive values for percent change in MAM mean rainfall shown in Fig. 2d in present-day ACT relative to GHG-only indicates the additional rainfall that could happen in the present-day if only GHGs were the dominant forcing and if anthropogenic aerosols (reduced to pre-industrial levels) were not effecting rainfall. But, instead of
such additional rainfall, we see a drying effect in present-day ACT relative to NAT because existing aerosols over-compensate the GHG warming effects over this region.

RC3: P 6 L 19-28: I think this whole discussion is too superficial. There are many speculations on how thermodynamic and dynamic effects could influence the precipitation changes which, in my view, are speculative and should be based on a more quantitative and solid analysis. For instance, I am not sure how an "approximately linear" scaling is deduced from the data presented in this study. If this just refers to the differences between the 1.5 and 2.0 simulations, I could well imagine a case in which precipitation increases due to both increase in the atmospheric moisture content and in the monsoon circulation, and this increase is amplified in the 2.0 case, which may also produce a linear change over these simulations. Moreover, is a linear scaling really what we expect thermodynamically? The Clausius-Clapeyron relation is non-linear.

AC3: We are indeed speculating based on work by others (e.g. Bollasina et al. 2011). It is beyond the scope of this paper to investigate the dynamic response in detail. We are planning on doing that in a more advanced study with additional model simulations from HAPPI, but for now all we do is to "indicate" or "suggest" that a combination of mechanisms might be at play. None of what we say is conclusive, but it provides a potential explanation as to which factors could be at play. We can simply delete this paragraph, yet we believe that this would severely affect the integrity/content of this section. We would therefore pledge to keep the gist of the paragraph. We have added a sentence clarifying that our statements are rather speculative.

RC3: The conclusion that dynamic changes play a secondary role should be manifested in a quantitative way. Also the statement that the thermodynamic response "usually scales with 20-40% of Clausius Clapeyron" is vague and, as such, not comprehensible.

AC3: We did reformulate in order to make a less strong conjecture as to what could be going on. We deleted the last statement (although we are not sure why it is not

comprehensible).

RC3: P 7 L 1-2: I cannot follow here: In the region with the strongest decrease in Fig. 3a, the aerosol effect is small.

AC3: This sentence is now deleted to avoid confusion.

RC3: P 7 L6: The abbreviation "SPI" has not been introduced.

AC3: Added now.

RC3: P 7 L 11-3: Again I cannot follow. For instance, the changes in 5b,d are similar to those in 4b,d

AC3: This is rewritten as: "Changes in mean absolute rainfall are much more pronounced over sub-regions 1 and 2, where both MAM and JJAS rainfall exhibit clear shifts from one forcing to another forcing scenario (Fig. 4). On the other hand, over sub-regions 3 and 4, only JJAS rainfall exhibited a robust shift (Fig. 5 b & d)."

RC3: P 7 L 14: It is difficult to assess the relative change at this point, as the figures show absolute values

AC3: Not sure we can exactly follow their point. Fig. 2 and 3 (as well as S1 and S2 for SPI) show relative percent changes. We now discuss absolute changes. Can the referee elaborate on what is meant with that comment? We would highly appreciate that.

RC3: P 7 L 16-17: This doesn't fit with the interpretation in the figure caption. In general, I find it difficult to shift parts of the discussion to the figure captions.

AC3: Figure caption reformulated as "...... aerosol impacts over both sub-regions 1 and 2 are larger in MAM dry season than that in JJAS wet season." Part of the figure caption is now also discussed here (as opposed to the figure caption).

RC3: P 7 L 18: lapse rate and stability changes are not different feedbacks

AC3: We agree. Instead we have clarified the point and added effects on boundary layer turbulence. Rewritten as: "Consequently, direct and indirect aerosol effects, accompanied by feedbacks such as reduced lapse rate, reduced boundary layer turbulence, or a modified land-sea circulation, remain to be a potent driver for changing monsoonal rainfall amounts."

RC3: P 7 L 21: Again I don't understand what "linear" increase means here.

AC3: By linear response, we meant steady and gradual increase in the climate change impact on rainfall from one forcing scenario to another due the warming effects, for example, from ACT to HAPPI 1.5 and HAPPI 1.5 to HAPPI 2.0. In other words, a 'linear' response is when we impacts (e.g. drying, wettening, warming, cooling) continue as a function of increased warming, i.e. scaling with global mean surface temperature.

RC3: Section 3.3, first paragraph: There is an imbalance between the amount of text/discussion and the number of figures. The reader is left alone with much of the material shown in Figs. 6-9. Either expand this discussion, or, if you think that the results are not that important, move parts of the figures to the supplement.

AC3: This section is expanded to discuss results from figures 6-9.

RC3: P 8 L 4: "appear to counter": I cannot see how you come to this conclusion.

AC3: Rephrased: "Might partially" instead of "appear"

RC3: P 8 L 24: "to a lesser extent": Really? Aren't the relative changes larger for the extremes?

AC3: Revised to: [are projected to increase seasonal mean and extreme rainfall probabilities during] "probabilities".

RC3: P 8 L 31: "we conclude that the drier subregions ...": I don't think this has been demonstrated. To show this, the masking effect has to be quantified. Furthermore, it is not clear to me which region and season you're referring to.

AC3: This part was indeed very confusing. Please accept our apologies. We have revised this paragraph including a more quantitative statement.

RC3 Comments on Figures

RC3: Fig. 1: I think this figure is too busy. I cannot distinguish the different shadings and also the lines of the observations are hard to see.

AC3: Figure 1 is redone with higher resolution and better visual clarity (see Fig. AC3 b).

RC3: Fig. 3: shorten caption (as Fig. 2, but for the monsoon season)

AC3: All figure captions are now reasonably shortened.

RC3: Fig. 4: I cannot follow the interpretation in the caption. For instance, I don't see such large differences in the masking effect between the regions. More in general, it is hard for me to understand how the masking effect is quantified here.

AC3: You are right; the masking effects vary only with wet and dry seasons. The figure interpretation is rewritten as: "The figure shows that aerosol impacts over both sub-regions 1 and 2 are larger in MAM dry season than that in JJAS wet season." We compare the NAT, ACT and GHG-only (green, gray and orange boxplots) results to quantify the aerosol masking effects for both sub-regions (see Figure AC3.c).

RC3: Fig. 10: Again, the interpretation in the caption is unclear (e.g., which region and season are you referring to?)

AC3: Rewritten the figure caption as follows (see Figure AC3.d): Same as Figure AC3.c but for sub-region 3 and 4. During MAM over both sub-regions 3 and 4, aerosol effects suppress the mean rainfall change between NAT and ACT (i.e., ACT rainfall is lower than NAT). On the other hand, during JJAS over both sub-regions 3 and 4, with lesser aerosol masking effects, ACT has higher mean rainfall than NAT and GHG-only would have noticeably much higher mean rainfall.

References:

Bollasina, M.A., Ming, Y. and Ramaswamy, V.: Anthropogenic Aerosols and the Weakening of the South Asian Summer Monsoon, Science 334 (6055), 502-505, doi: 10.1126/science.1204994, 2011.

[Figure]

[Figure]

**Fig. 1.** AC3 a: Comparison between two interpolation methods applied for seasonal mean precipitation during JJAS and MAM over Bangladesh.

[Figure]

**a. Rainfall at sub-region 1 - ACT/NAT/GHG-only Vs HAPPI1.5/2.0**

**b. Rainfall at sub-region 2 - ACT/NAT/GHG-only Vs HAPPI1.5/2.0**

**c. Rainfall at sub-region 3 - ACT/NAT/GHG-only Vs HAPPI1.5/2.0**

**d. Rainfall at sub-region 4 - ACT/NAT/GHG-only Vs HAPPI1.5/2.0**

**Fig. 2.** AC3 b: Seasonal cycles of five day mean rainfall under different forcing scenarios over the four sub-regions of Bangladesh.

[Figure]

**Fig. 3.** AC3 c: Seasonal mean rainfall in MAM (left column) and JJAS (right column) over the sub-regions 1 and 2 (top and bottom row) of Bangladesh.

[Figure]

[Figure]

**Fig. 4.** AC3 d: Same as Figure AC3.c but for sub-region 3 and 4.

---

## Author Comment (AC4) · 7 Jul 2019

**Risks of seasonal extreme rainfall events in Bangladesh under 1.5 and 2.0 degrees' warmer worlds – How anthropogenic aerosols change the story?**

Ruksana H. Rimi[1], Karsten Haustein[1], Emily J. Barbour[1,2], Sarah N. Sparrow[3], Sihan Li[1,3], David C.H. Wallom[3] and Myles R. Allen[1].

[1]Environmental Change Institute, School of Geography and the Environment, University of Oxford, Oxford, OX1 3QY, UK.

[2] Commonwealth Scientific and Industrial Research Organisation, Land and Water, Canberra, ACT 2601, Australia.

[3]Oxford e-Research Centre, Department of Engineering Science, University of Oxford, Oxford, OX1 3QG, UK.

*Correspondence to:* Ruksana H. Rimi (ruksana.rimi@ouce.ox.ac.uk)

**Abstract.** Anthropogenic climate change is likely to increase  risk of extreme weather events in the future. The term 'risk' here means the probability of occurrence of a hazard, e.g., an extreme rainfall event that can trigger sudden flash-flood, landslide or flood. Previous studies have robustly shown how and where climate change has already changed the risks of weather extremes. However, developing countries have been somewhat underrepresented in these studies, despite high vulnerability and limited capacities to adapt. How additional global warming would affect the future risks of extreme rainfall events in Bangladesh needs to be addressed to limit adverse impacts. Our study focuses on understanding and quantifying the relative risks of extreme rainfall events in Bangladesh under the Paris Agreement temperature goals of 1.5°C and 2°C warming above pre-industrial levels. In particular, we investigate the influence of anthropogenic aerosols on these risks given their likely future reduction and resulting amplification of global warming. Using large ensemble regional climate model simulations from weather@home under different forcing scenarios, we compare the risks of rainfall events under pre-industrial (natural), current (actual), 1.5°C, and 2.0°C warmer and greenhouse gas (GHG)-only (with pre-industrial levels of anthropogenic aerosols ) conditions. Analysis of percent change, standardized precipitation index and absolute change in seasonal mean rainfall revealed that there both GHGs and anthropogenic aerosols play important roles in determining the overall climate change impact 
[revised manuscript text omitted]

Du, J., Fang, J., Xu, W. and Shi, P.: Analysis of dry/wet conditions using the standardized precipitation index and its potential usefulness for drought/flood monitoring in Hunan Province, China, Stoch. Environ. Res. Risk Assess., 27(2), 377–387, doi:10.1007/s00477-012-0589-6, 2013.

Endo, H., Kitoh, A., Ose, T., Mizuta, R. and Kusunoki, S.: Erratum: Future changes and uncertainties in Asian precipitation simulated by multiphysics and multi-sea surface temperature ensemble experiments with high-resolution Meteorological Research Institute atmospheric general circulation models (MRI-AGCMs) (Jo, J. Geophys. Res. Atmos., 118(5), 2303, doi:10.1002/jgrd.50267, 2013.

Fahad, M. G. R., Saiful Islam, A. K. M., Nazari, R., Alfi Hasan, M., Tarekul Islam, G. M. and Bala, S. K.: Regional changes of precipitation and temperature over Bangladesh using bias-corrected multi-model ensemble projections considering high-emission pathways, Int. J. Climatol., doi:10.1002/joc.5284, 2017.

Faust, E.: Rapid attribution: Is climate change involved in an extreme weather event?, [online] Available from: https://www.munichre.com/topics-online/en/2017/topics-geo/rapid-attribution, 2017.

Fung, F., Lopez, A. and New, M.: Water availability in +2 C and +4 C worlds, Philos. Trans. R. Soc. A Math. Phys. Eng. Sci., 369(1934), 99–116, doi:10.1098/rsta.2010.0293, 2011.

Guillod, B. P., Jones, R. G., Bowery, A., Haustein, K., Massey, N. R., Mitchell, D. M., Otto, F. E. L., Sparrow, S. N., Uhe, P., Wallom, D. C. H., Wilson, S. and Allen, M. R.: Weather@home 2: Validation of an improved global-regional climate modelling system, Geosci. Model Dev., 10(5), 1849–1872, doi:10.5194/gmd-10-1849-2017, 2017.

Guo, L., Highwood, E. J., Shaffrey, L. C. and Turner, A. G.: The effect of regional changes in anthropogenic aerosols on rainfall of the East Asian Summer Monsoon, Atmos. Chem. Phys., 13(3), 1521–1534, doi:10.5194/acp-13-1521-2013, 2013.

Gutro, R.: Bangladesh's Heavy Rainfall Examined With NASA's IMERG, Nasa Gpm [online] Available from: https://www.nasa.gov/feature/goddard/2017/bangladeshs-heavy-rainfall-examined-with-nasas-imerg (Accessed 14 November 2017), 2017.

Hauser, M., Gudmundsson, L., Orth, R., Jézéquel, A., Haustein, K., Vautard, R., van Oldenborgh, G. J., Wilcox, L. and Seneviratne, S. I.: Methods and Model Dependency of Extreme Event Attribution: The 2015 European Drought, Earth's Futur., 5(10), 1034–1043, doi:10.1002/2017EF000612, 2017.

Haustein, K., Uhe, P. F., Rimi, R. H., Islam, A. S. and Otto, F. E. L.: Is the wettest place on Earth getting wetter?, Geophys. Res. Lett., n.d.

IPCC, 2013: The Physical Science Basis. Contribution of Working Group I to the Fifth Assessment Report of the Intergovernmental Panel on Climate Change, Cambridge University Press, Cambridge, United Kingdom and New York, NY, USA. [online] Available from: http://scholar.google.nl/scholar?hl=nl&q=climate+change&btnG=&lr=&oq=climate+ca#2, 2013.

Kripalani, R. H., Oh, J. H., Kulkarni, A., Sabade, S. S. and Chaudhari, H. S.: South Asian summer monsoon precipitation variability: Coupled climate model simulations and projections under IPCC AR4, Theor. Appl. Climatol., 90(3–4), 133–159, doi:10.1007/s00704-006-0282-0, 2007.

Kumar, D., Arya, D. S., Murumkar, A. R. and Rahman, M. M.: Impact of climate change on rainfall in Northwestern Bangladesh using multi-GCM ensembles, Int. J. Climatol., 34(5), 1395–1404, doi:10.1002/joc.3770, 2014.

Kumar, K. K., Kamala, K., Rajagopalan, B., Hoerling, M. P., Eischeid, J. K., Patwardhan, S. K., Srinivasan, G., Goswami, B. N. and Nemani, R.: The once and future pulse of Indian monsoonal climate, Clim. Dyn., 36(11–12), 2159–2170, doi:10.1007/s00382-010-0974-0, 2011.

Lau, W. K. M. and Kim, K. M.: Fingerprinting the impacts of aerosols on long-term trends of the Indian summer monsoon regional rainfall, Geophys. Res. Lett., 37(16), n/a-n/a, doi:10.1029/2010GL043255, 2010.

Lee, D., Min, S.-K., Fischer, E. M., Shiogama, H., Bethke, I., Lierhammer, L. and Scinocca, J.: Impacts of half a degree additional warming on the Asian summer monsoon rainfall characteristics, Environ. Res. Lett., doi:10.1088/1748-9326/aab55d, 2018.

Li, S., Mote, P. W., Rupp, D. E., Vickers, D., Mera, R. and Allen, M.: Evaluation of a regional climate modeling effort for the western United States using a superensemble from weather@home, J. Clim., 28(19), 7470–7488, doi:10.1175/JCLI-D-14-00808.1, 2015.

Li, W., Fu, R., Juarez, R. I. N. and Fernandes, K.: Observed change of the standardized precipitation index, its potential cause and implications to future climate change in the Amazon region, Philos. Trans. R. Soc. B Biol. Sci., 363(1498), 1767–1772, doi:10.1098/rstb.2007.0022, 2008.

Li, Z., Lau, W. K. M., Ramanathan, V., Wu, G., Ding, Y., Manoj, M. G., Liu, J., Qian, Y., Li, J., Zhou, T., Fan, J., Rosenfeld, D., Ming, Y., Wang, Y., Huang, J., Wang, B., Xu, X., Lee, S. S., Cribb, M., Zhang, F., Yang, X., Zhao, C., Takemura, T., Wang, K.,

Xia, X., Yin, Y., Zhang, H., Guo, J., Zhai, P. M., Sugimoto, N., Babu, S. S. and Brasseur, G. P.: Aerosol and monsoon climate interactions over Asia, Rev. Geophys., 54(4), 866–929, doi:10.1002/2015RG000500, 2016.

Mahfouz, P., Mitri, G., Jazi, M. and Karam, F.: Investigating the Temporal Variability of the Standardized Precipitation Index in Lebanon, Climate, 4(2), 2016.

5  Massey, N., Jones, R., Otto, F. E. L., Aina, T., Wilson, S., Murphy, J. M., Hassell, D., Yamazaki, Y. H. and Allen, M. R.: Weather@Home-Development and Validation of a Very Large Ensemble Modelling System for Probabilistic Event Attribution, Q. J. R. Meteorol. Soc., 141(690), 1528–1545, doi:10.1002/qj.2455, 2015.

Mckee, T. B., Doesken, N. J. and Kleist, J.: The relationship of drought frequency and duration to time scales, in AMS 8th Conference on Applied Climatology, vol. 17, pp. 179–184, American Meteorological Society Boston, MA., 1993.

10  McKee, T. B., Doesken, N. J. and Kleist, J.: Drought Monitoring with Multiple Time Scales, in Proceedings of the 9th AMS Conference on Applied Climatology, pp. 233–236, American Meteorological Society Dallas, Boston, MA., 1995.

Menon, A., Levermann, A., Schewe, J., Lehmann, J. and Frieler, K.: Consistent increase in Indian monsoon rainfall and its variability across CMIP-5 models, Earth Syst. Dyn. Discuss., 4(1), 1–24, doi:10.5194/esdd-4-1-2013, 2013.

Mitchell, D., James, R., Forster, P. M., Betts, R. A., Shiogama, H. and Allen, M.: Realizing the impacts of a 1.5 °C warmer world,
15  Nat. Clim. Chang., 6(8), 735–737, doi:10.1038/nclimate3055, 2016.

Mitchell, D., AchutaRao, K., Allen, M., Bethke, I., Beyerle, U., Ciavarella, A., Forster, P. M., Fuglestvedt, J., Gillett, N., Haustein, K., Ingram, W., Iversen, T., Kharin, V., Klingaman, N., Massey, N., Fischer, E., Schleussner, C. F., Scinocca, J., Seland, Ø., Shiogama, H., Shuckburgh, E., Sparrow, S., Stone, D., Uhe, P., Wallom, D., Wehner, M. and Zaaboul, R.: Half a degree additional warming, prognosis and projected impacts (HAPPI): Background and experimental design, Geosci. Model Dev.,
20  10(2), 571–583, doi:10.5194/gmd-10-571-2017, 2017.

Murshed, S. B., Islam, A. K. M. and Khan, M. S. A.: Impact of climate change on rainfall intensity in Bangladesh, Dhaka, Bangladesh. [online] Available from: http://teacher.buet.ac.bd/akmsaifulislam/reports/Heavy_Rainfall_report.pdf, 2011.

Naresh Kumar, M., Murthy, C. S., Sesha sai, M. V. R. and Roy, P. S.: On the use of Standardized Precipitation Index (SPI) for drought intensity assessment, Meteorol. Appl., 16(3), 381–389, doi:10.1002/met.136, 2009.

[revised manuscript text omitted]

**Supplementary Figure captions:**

**Figure S1. Relative change in  SPI of MAM mean rainfall between different forcing scenarios. The top row (panels a-d) shows the regional SPI changes over central parts of the South Asia (SA) while, bottom row (panels e-h) shows the SPI changes over Bangladesh. The four boxes (1-4) on top of the panel e  represent the four sub-regions of Bangladesh.  a.  ACT rainfall  SPI relative to  NAT over SA b.  ACT rainfall  SPI relative to HAPPI1.5 over SA c. HAPPI1.5 rainfall  SPI relative to HAPPI 2.0 over SA d.  ACT rainfall  SPI relative to GHG-only  over SA.**

**Figure S2. ~~Relative changes in the SPI of monsoon (JJAS) seasonal mean rainfall between different forcing scenarios. The top row (panels a-d) shows the regional SPI over central parts of the South Asia (SA) while, bottom row (panels e-h) shows the SPI over Bangladesh. The four boxes (1-4) on top of the panel e approximately represent the four sub-regions of Bangladesh. These four sub-regions (1-4) are used later for the relative quantification of risks of extreme monsoon rainfall events. a. present-day rainfall PC relative to natural pre-industrial climate over SA b. present-day rainfall PC relative to 1.5°C world over SA c. 1.5°C world rainfall PC relative to 2.0°C world over SA d. present-day rainfall PC relative to GHG-only climate over SA.The~~ This figure shows that the apparently non-linear response between panels of a, b, and c (or, e, f, g) can be explained by the response for aerosols in the panel d (or, h).**

**Figure S3. Return time plots for MAM five day mean rainfall  under different forcing scenarios over the sub-regions  1 and 2 of Bangladesh. The  ACT (black), ACT highest ( sky-blue), ACT lowest ( grey), NAT (green) and GHG-only (orange) ensembles are compared with the HAPPI 1.5 (blue) and HAPPI 2.0 (red) ensembles.**

**Figure S4. Same as Fig S3 but for  return time plots for MAM five day mean rainfall  under different forcing scenarios over the sub-regions  3 and 4 of Bangladesh.**

**Figure S5. Return time plots for JJAS five day mean rainfall  different forcing scenarios over the  sub-regions  1 and 2 of Bangladesh. The  ACT (black), ACT highest ( sky-blue), ACT lowest ( grey), NAT (green) and GHG-only (orange) ensembles are compared with the HAPPI 1.5 (blue) and HAPPI2.0 (red) ensembles. The risks of extreme rainfall events are evidently increasing between different forcing scenarios over sub-region 2.**

**Figure S6. Same as Fig S5 but for  return time plots for JJAS five day mean rainfall  under different forcing scenarios over the sub-regions  3 and 4 of Bangladesh.**

**Supplementary Text**

**Analysis methods:**

1. **Percentage Change (PC)** in seasonal mean precipitation is calculated for one forcing scenario relative to another forcing scenario to indicate the magnitude of change between the scenarios across the study region. This approach enables the identification of areas at risk of becoming wetter or drier. For instance, the PC for ACT relative to NAT in monsoon (JJAS) season is calculated as:

$$PC_{\text{ACT relative to NAT}} = \left[ \frac{\text{JJAS precipitation in ACT} - \text{JJAS precipitation in NAT}}{\text{Mean JJAS precipitation in ACT}} \right] \times 100$$

The multi-year monthly means of JJAS months for each decadal model ensemble is used to calculate the PC in all cases. The PC for pre-monsoon (MAM) season is calculated using the same approach.

2. The **Standardized Precipitation Index (SPI)** (Mckee et al., 1993; McKee et al., 1995) is a simple, flexible index which is powerful to effectively analyse both wet and dry periods. SPI is widely used for assessing wetting/drying effects (e.g., Du et al., 2013; Li et al., 2015, 2008; Mahfouz et al., 2016). Precipitation data is the only required input parameter to calculate the SPI and this can be computed for multiple timescales from 1 to 24 months (WMO, 2012). For example, SPI for monsoon precipitation during JJAS months in GHG only climate model ensemble (denoted as GHG-only) relative to actual climate model ensemble (denoted as ACT) is calculated by the following equation:

$$\text{SPI}_{\text{GHG-only relative to Act}} = \frac{\text{JJAS precipitation in GHG-only - JJAS precipitation in ACT}}{\text{Standard deviation of JJAS precipitation in ACT}}$$

The multi-year monthly means of JJAS months for each model ensemble is used to calculate the SPI in all cases. An SPI index value greater than 2.0 indicates areas are extremely wet, 1.5 to 1.99 indicates very wet; 1.0 to 1.49 moderately wet; -0.99 to 0.99 near normal; -1.0 to -1.49 moderately dry; -1.5 to -1.99 severely dry; and -2 and less indicate areas to be extremely dry (WMO, 2012).

---

## Author Response (AR1)

**Final Authors' response for the manuscript titled "Risks of seasonal extreme rainfall events in Bangladesh under 1.5 and 2.0 degrees' warmer worlds – How anthropogenic aerosols change the story" by Ruksana H. Rimi et al., 2019.**

5 We would like to thank the Editor and all Reviewers for their constructive comments and suggestions.

In this document, we sequentially present the replies to all reviewers (as added at the interactive discussion) and a marked-up version of the manuscript with author's edits in response to all comments.

Dear Editor,

We have attached the detailed replies to the reviewers' comments that are also available online in the open discussion. Afterwards, we have added a marked-up version of the manuscript with author's edits in response to all three reviewers.

In addition to the reply to the reviewers, we made some extra changes in the paper. For instance we have included in the revised manuscript

1. In the introduction we explain the context of HAPPI project development.
2. Additional four HAPPI model data analyses are added to compare with HadRM3P model results. Such comparison can test the robustness of results from HadRM3P model and significantly strengthen this paper by getting over the problem of model uncertainty.
3. In conclusion, a more emphasis is given on the fact that for the first time, a multi-model assessment of both present and future risks of extreme rainfall events considering anthropogenic climate change drivers of GHGs and aerosols is done at sub-regional local scale in Bangladesh in this paper.
4. Information about prescribed fields of $SO_2$ emission in the HadRM3P model is added because of the importance of the role played by the aerosols. But for the sake of better readability of this paper, this information is added in the supplementary material.

We hope that the current version of the manuscript contains sufficient amendments to address the comments made by the reviewers. We hope that this manuscript version will get a positive decision.

Best regards,
Ruksana Rimi

**Response to Reviewer 1**

We thank for the constructive comments from the anonymous Reviewer 1. We have carefully revised the manuscript to incorporate necessary amendments as per suggestions. Responses to the Referee Comments 1(RC1) are presented in the following paragraph in the Author's Comments 1(AC1):

**Introduction**

RC1: *The introduction provides a general background on the research topic with clearly stated objectives and research question. The author divided whole Bangladesh into four sub-regions (page 3, line11-14) to evaluate the risk of extreme rainfall events. Are there other specific reasons for to divide into four seasons like climatological variation or any other previous study used these*

10 *sub-regions.*

AC1: These 4 sub-regions of Bangladesh and the different seasons are used for a reason. The three wet seasons with substantial climatological variations include: pre-monsoon (during Mar-Apr-May; MAM), monsoon (during Jun-Jul-Aug-Sep; JJA) and post-monsoon (Oct-Nov; SON). Little or no rain occurs in winter (during Dec-Jan-Feb; DJF). This dry winter season is excluded from

15 our analysis because we are interested in wet extremes. Any MAM extreme rainfall events are known to cause flash floods and substantial crop damage (Ahmed et al., 2017).

Bangladesh receives more than 75% of the annual total rainfall during JJAS (Shahid, 2010). An extreme rainfall event in this period can therefore cause wide-spread flooding and landslide eventually leading to loss of lives and livelihoods. A high impact

20 SON rainfall event may be associated with the coastal floods that occur due to storm surges or tidal effects along the northern part of Bay of Bengal (Hossain, 1998).

Considering both meteorological hazards and potential impacts, we have looked at extreme rainfall events of pre-monsoon and monsoon seasons and excluded the post-monsoon season. The same 4 sub-regions are used in Rimi et al., (2019) where, we

25 evaluate the model's performance in simulating extreme events (see Fig. AC1.a). We believe that the pre-evaluated model simulations over the same 4 sub-regions provide confidence in analyzing the extreme rainfall events under different forcing scenarios.

In the revised manuscript, we have added the following lines at **page 5 lines 16-24**: "In Bangladesh, any MAM extreme rainfall

30 events are known to cause flash floods and substantial crop damage (Ahmed et al., 2017). Bangladesh receives more than 75% of the annual total rainfall during JJAS (Shahid, 2010). An extreme rainfall event in this period can therefore cause wide-spread flooding and landslide eventually leading to loss of lives and livelihoods. A high impact post-monsoon (Oct-Nov; ON) rainfall event may be associated with the coastal floods that occur due to storm surges or tidal effects along the northern part of Bay of Bengal (Hossain, 1998). Considering meteorological hazards and potential impacts, MAM and JJAS extreme rainfall events are

35 analyzed in this study while, ON rainfall events are excluded. Winter (during Dec-Jan-Feb; DJF) season is also excluded because little or no rain occurs during DJF and we are interested in wet extremes."

And added the following lines at **page 5 lines 28-30**: "The model's performance in simulating extreme rainfall events over these same 4 sub-regions is evaluated in Rimi et al., (2019). Such pre-evaluated model simulations provide confidence in analyzing

40 comparative risks of extreme rainfall events under different forcing scenarios."

RC1: *A standalone Figure of Bangladesh including the four sub-regions could be useful with mean seasonal rainfall.*

AC1: Because a standalone figure of Bangladesh including the four sub-regions with mean seasonal rainfall has already been shown in Rimi et al., 2019 (see Fig AC1.b); we are not using the same kind of figure in this paper. Nevertheless, the sub-regions are indicated in panel e of Fig 2 & 3 of this manuscript.

*RC1: Line 16 page 2 – In June 2017, heavy rainfall killed at least 156 people (needs a citation).*

AC1: The following citation is added for this information in the revised bibliography at **page 15 lines 13-15**: Paul, R. and Hussain, Z.: Landslide, floods kill 156 in Bangladesh, India; toll could rise, Reuters, 14th June [online] Available from: https://uk.reuters.com/article/uk-bangladesh-landslides/landslide-floods-kill-156-in-bangladesh-india-toll-could-rise-idUKKBN1950AG, 2017.

**Data and Methods**
**Model setup and experiment design**

*RC1: The wettest and driest years are not well presented in the Table S2. Is it range of years or individual year? Classification of wet years and dry is not clearly mentioned. It has been mentioned that spatiotemporal average of rainfall has been used here. Is there any threshold for the classification of wet period and dry period? In Bangladesh flooding years are considered as wet year during monsoon. This needs to be made clear.*

AC1: We identified the two wettest and two driest years during 2006–2015 over each of the four sub-regions of Bangladesh using ACT data. This identification process involved comparing the magnitude along with return periods of rainfall events during MAM and JJAS in each of the years throughout 2006 to 2015 (ACT model ensemble with 200 members per year used). Then a pair of wettest and driest years is used to approximately indicate the noise-to-signal ratio.

For example, according to ACT model ensemble, during monsoon season at sub-region 2, the wettest years with extreme rainfall events are 2008 and 2012 (see Fig. AC1.c, the red and blue dots for individual model runs and shadings for 5-95% confidence intervals) and the driest years are 2006 and 2013 (yellow and cyan dots for individual model runs and shadings for 5-95% confidence intervals). Using the pair of these wettest and driest years from the model ensemble, we could demonstrate the plausible spread of the rainfall events within this model ensemble as a measure of noise-to-signal ratio.

Because this model ensemble has same forcings in each year for the historical period of 2006-2015, the only variability playing a role in changing the intensity of rainfall is natural variability of sea surface temperatures (SSTs). The use of two wettest and driest years, therefore explains how much natural variability of SSTs contributes to the variability of rainfall intensities over this study area.

We agree with your comment and changed the presentation of the years in the supplementary Table 2 to make it clear that they are the two individual years with either wettest or, driest conditions in a 10-year period over a specific sub-region.

In the revised manuscript, we have added the following lines at **page 6 lines 20-23**: "ACT model ensemble has the same forcings in each year for the historical period of 2006-2015; the only variability playing a role in changing rainfall intensity is therefore the natural variability of SSTs. For this reason, these two wettest and driest years of ACT model ensemble approximately indicate how much natural SST variation can contribute to changing rainfall intensities."

**Results and Discussion**

**Section 3.1 Model Evaluation for Five day mean rainfall**

*RC1: "Five day mean rainfall is used to represent the timescale responsible for river flooding as opposed to daily extremes that cause flash floods primarily in the pre-monsoon season." Is the 5 days rainfall causes flooding or 1-day extreme rainfall causes flash flood in Bangladesh and what is the intensity of rainfall termed as extreme (what is the amount of rainfall mm/day considered as extreme value)? Citation may clear this statement.*

AC1: By extreme rainfall event, we mean a high impact rainfall event (i.e., sufficient to cause flooding or landslides) with up to 100 year return period (a rare event with low frequency of occurrence). The intensity of the rainfall event can vary depending on both location and season.

For example, at north-east Bangladesh, in pre-monsoon season, more than 150 mm rainfall event over a 6-day period can lead to an early flash flood (Ahmed et al., 2017). On the other hand, at south-east parts of Bangladesh, in monsoon season, more than 350 mm rainfall events over a 3-day period is enough to cause a landslide (Ali et al., 2014). Whereas, for a wide-spread river flooding e.g., the Brahmaputra River Basin flooding in August 2017, 10-day extreme rainfall event is considered (Philip et al., 2018). Considering the range from 1- to 10-day high impact rainfall events that can trigger flooding and landslides in Bangladesh, we focused on daily and 5-day events because these events can provide a typical idea about the potential risks.

We have added the lines in the revised manuscript at **page 5 lines 32-40**: "In Bangladesh 1- to 10-day high impact rainfall events can trigger flooding and landslides. For example, at north-east Bangladesh (sub-region 2), more than 150 mm MAM rainfall over a 6-day period can lead to an early flash flood (Ahmed et al., 2017). In contrast, at south-east Bangladesh (sub-region 4), more than 350 mm JJAS rainfall over a 3-day period is enough to cause a landslide (Ali et al., 2014). For a wide-spread river flooding e.g., the Brahmaputra River Basin flooding in August 2017, 10-day extreme rainfall event is considered (Philip et al., 2018). Considering such variations in rainfall magnitudes causing different hazards, we focused on daily and 5-day rainfall events to analyze the potential risks.

The seasonal cycles of presented here are based on 5-day rainfall, which is used to represent the timescale responsible for river flooding as opposed to daily extremes that cause flash floods primarily in the pre-monsoon season."

*RC1: Fig.1 represents annual cycle of the four sub-regions in Bangladesh and results of the five models (ACT, NAT, GHG, HAPPI 1.5 and HAPPI 2.0) show maximum rainfall occurs in June. June to September is the monsoon month and June is the month of monsoon onset. Usually, July is the maximum rainfall month in Bangladesh. Do the results indicate any shifting of monsoon timing Due to the monsoon climate, the overall variation (inter-annual) of rainfall in JJAS months (seasonal) is not quite high? Bangladesh has almost similar pattern of monsoon precipitation in the JJAS months. Underestimation by 25-65% is quite high. The bias and uncertainty within these values is very high. The authors need to explain the reasoning for this a bit more.*

AC1: No, as per observational data (APHRODITE and CPC) that are used in the annual cycles in Fig 1; there are no indications of temporal shifting of monsoon. We find an early monsoon onset in the model simulations. Such early onset of monsoon is also reported in other model based studies (e.g., in Caesar et al., 2015; Fahad et al., 2017; Janes and Bhaskaran, 2012).

At the time of writing this paper, APHRODITE was only available until 2007, so we used 1998-2007 data for comparison. Recently, APHRODITE has been updated until 2015, allowing us to use 2006-2015 data for comparison (CPC data is also used for 2006-2015 duration). As a result, APHRODITE and CPC observations are now in better agreement (see Fig. AC1.d) and

model bias is also smaller than before (highest bias level of 65% dropped to 50%). The 25-50% underestimation of monsoon precipitation is quantified based on the model ensemble mean compared to observations. Most of the observed rainfall is found to be within the 10-90% confidence interval of the model data.

Overall, the weather@home model simulates the annual cycle of rainfall with satisfactory agreement, despite the dry monsoon rainfall bias. Furthermore, Rimi et al. (2019) shows that the weather@home model gives a reasonable representation of extreme rainfall events with return periods of 50-100 years. Therefore, we are confident that these biases will not affect our risk assessments for extreme rainfall events.

We have rewritten the paragraph with **lines 17- 31 in page 7** as: "The seasonal cycles of 5-day rainfall from the different model ensembles are adequately representative of the observed seasonal cycles. Most of the observed rainfall is found to be within the 10-90% confidence intervals of the model data. We find an early monsoon onset in the model simulations, which is also reported in previous studies (e.g., in Caesar et al., 2015; Fahad et al., 2017; Janes and Bhaskaran, 2012). However, JJAS rainfall is underestimated by 25-50% depending on the observational dataset and sub-regions. This bias is higher (up to 50% dry bias) in the wetter sub-regions of 2 and 4 (Figs. 1b & d) and lower (up to 30% dry bias) in the drier sub-regions of 1 and 3 (Figs. 1a & c). Underestimation of JJAS rainfall is reported in other model based studies over Indian monsoon region (Goswami et al., 2014; Kumar and Dimri, 2019; Saha et al., 2014) and specifically in Bangladesh (Caesar and Janes, 2018; Islam, 2009; Macadam and Janes, 2017). The bias is apparently present in all model scenarios used in this study; hence it is unlikely to affect the comparison between model scenarios. We also note that the signal of the change due to the changing climate is relatively small in comparison to the total rainfall. Therefore, the model is considered fit for purpose in assessing the potential impacts of climate change on extreme rainfall events."

**Section 3.2: Impact of Climate Change and Aerosol Reduction on Seasonal Mean Rainfall**

*RC1: Provides important information regarding rainfall change due to warming 1.5˚C to 2˚C and aerosol impact. However, the change has been computed using model based on simulated observed data. The actual changes can be presented by using observation data (e.g., Aphrodite).*

AC1: While it is definitely useful to look at the observed changes between pre-industrial and present-day rainfall, present day and future warmer (1.5 and 2.0 degrees) scenarios or the aerosol impacts; we can only do this kind of comparison using model simulated data. Because we have no high resolution gridded observational dataset that offers the pre-industrial records as well as the future scenarios of 1.5 and 2.0 degrees warming. APHRODITE data was only available from 1963 to 2007 at the time of writing this paper. Recently they have updated their data up to 2015. CPC observation dataset extends from 1979-present.

*RC1: "While aerosol effects are consistent with other regions, the GHG induced rainfall is hampered, likely due to dynamic changes such as a delayed onset of the monsoon in response to warming", It can be supported with other relevant studies (e.g. the variation of interannual rainfall may depend on the onset of monsoon).*

AC1: While it is beyond the scope of this study to identify the exact mechanisms that lead to future changes associated with aerosol removal (or the change due to contemporary emissions for that matter), we point to the literature where this issue has been investigated in some detail already. The seminal paper by Bollasina et al. (2011), as well as more recent work by Zhao et al. (2019) are excellent resources that support our point. We have added these two references to support this statement at **page 9 lines 13-14**.

**3.3 Rainfall extreme:**

**Line 40, page7**

*RC1: "The signal-to-noise ratio is higher in the monsoon season across all sub-regions with the lowest and highest ratio in sub-region 1 and 3, respectively (Figs. 8a & 9a)". This statement may be needed further explanation.*

AC1: This statement is rewritten at **page 9 lines 33-37** as "Overall, the signal-to-noise ratio is higher across all sub-regions, during JJAS compared to that during MAM. During MAM, the highest and lowest signal-to-noise ratio is over sub-region 1 and 3, respectively (Figs. 6a & 7a). On the other hand, during JJAS, we find the highest and lowest signal-to-noise ratio is over sub-region 3 and 1, respectively (Figs. 9a & 8a). The lower the ratio, the more difficult it is to establish causality as natural variability due to ENSO or circulation anomalies is higher."

**References**

*RC1: The author referred Banglapedia, 2012 as citation in page 2 line 10. However, did not provide in the reference list.*

AC1: We have now added this citation as "Banglapedia: River and Drainage system, Banglapedia- Natl. Encycl. Bangladesh [online] Available from: http://en.banglapedia.org/index.php?title=River_and_Drainage_System, 2012." in the revised bibliography. See **page 12 lines 4-5**

**Other comments:**

*(a) The title of the paper says risks of seasonal extreme rainfall events and presented rainfall extreme using daily and five day mean rainfall. One day max and 5day max would be better presentation of rainfall extreme. It is also necessary to have a better description why daily and five day mean rainfall has been used.*

AC1: This is because our focus is on unusual rare rainfall events with the potential to cause high impacts on the ground in pre-monsoon and monsoon seasons. We have used daily and 5-day running mean rainfall events throughout a season and then looked at events crossing a threshold (e.g. 250mm/day that can trigger floods or, landslides) with a high return period (e.g. 100 years). In particular, we have explored whether or not, and to what extent, the probability of having that same magnitude event (i.e. 250mm/day) changes in that particular season from one forcing scenario to another (e.g. from ACT to HAPPI 1.5).

An explanation for using 1 and 5-day rainfall is given at **page 5 lines 32-39**: "In Bangladesh 1- to 10-day high impact rainfall events can trigger flooding and landslides. For example, at north-east Bangladesh (sub-region 2), more than 150 mm MAM rainfall over a 6-day period can lead to an early flash flood (Ahmed et al., 2017). In contrast, at south-east Bangladesh (sub-region 4), more than 350 mm JJAS rainfall over a 3-day period is enough to cause a landslide (Ali et al., 2014). For a wide-spread river flooding e.g., the Brahmaputra River Basin flooding in August 2017, 10-day extreme rainfall event is considered (Philip et al., 2018). Considering such variations in rainfall magnitudes causing different hazards, we focused on daily and 5-day rainfall events to analyze the potential risks."

*(b) Inconsistent in figure indexing spacing: In the results (e.g. Sect. 3.2) it is needed to be consistent with spacing when referencing to figures. For example, Fig.2 d and (Fig. 2d), (Figs. 8a & 8b) and (Fig. 4 a & c).*

AC1: Figure indexing spacing issue has been resolved. Now this is uniformly done throughout the manuscript using Fig. 1a; Figs. 1a & b; Figs. 1a-d and Figs. 1b & 2b style.

*(c) The Figure captions are too long. The author started to describe the results in some of them (e.g. Fig. 5). The results or discussion should be in text. Caption should be concise and just define what the Figure shows with the necessary information to gather information from it.*

AC1: We have now moved parts of the figure captions to result and discussion section to reduce the length, and made them concise.

*(d) Line 6 (page 4) – Evaluation of the model for the region was conducted by Rimi et al. (2019) and demonstrated a reasonable agreement between model results and observational datasets for extreme rainfall events. What is a reasonable agreement? Which statistical skills show general agreement (e.g. r2, KGE). For example, 60% of stations achieved values greater than 0.6 between modelled and observed data.*

AC1: We present here Fig. AC1.a adapted from Rimi et al., (2019) for reference. In Figure AC1a., the black line represents the model simulated rainfall events; while, red, blue, orange and sky-blue colours indicate APHRODITE, GPCC, CPC gridded observations and TRMM satellite data respectively. Based on these figures, Rimi et al., (2019) reports that the pre-monsoon and monsoon extreme rainfall events with up to 100 years of return periods are adequately well captured by the model over Bangladesh when compared to APHRODITE, GPCC, CPC gridded observations and TRMM satellite data.

Although the observation data sets used in that model evaluation paper had short lengths of records; by fitting a Generalized Extreme Value distribution, the authors demonstrated that the model simulated extreme events are in good agreement with the observations.

In the revised manuscript we added the following lines at **page 4 lines 12-15**: "MAM and JJAS extreme rainfall events with up to 100 years of return periods are adequately well captured by the model over Bangladesh at local sub-regional scales when compared to high resolution gridded observation datasets as well as satellite data (Rimi et al., 2019a). The model is therefore considered to be fit for the purpose of assessing climate change impacts on extreme rainfall events under different forcing scenarios."

*RC1: The discussion article 'Risks of seasonal extreme rainfall events in Bangladesh under 1.5 and 2.0degrees' warmer worlds – How anthropogenic aerosols change the story' by Ruksana H. Rimi et al. is very interesting focused on the extreme rainfall events due climate change particularly 1.5 and 2 degrees warmer world. This is a comprehensive analysis of future projection of multi model rainfall over several sub-region of Bangladesh. The author provides sufficient graphs and maps in the article which explained the results. The major findings of the article are related with the global warming and its implication extreme weather events for Bangladesh. Finally, I suggest that the author will consider the above comments in finalizing the script. The article is recommended to publish with minor correction.*

AC1: The authors highly appreciate your careful review with constructive comments. We have updated the manuscript following most of your suggestions. In case of any disagreement, we have provided our explanation behind that. We hope that now the manuscript is ready to be accepted for publication.

[revised manuscript text omitted]

**Figure AC1.a:** Left panels show 5-day rainfall events in JJA and MAM over Bangladesh while right panels show the same but for sub-region 2. The black line is for the model simulated events, while, red, blue, orange and sky-blue colours indicate APHRODITE, GPCC, CPC gridded observations and TRMM satellite data respectively.

[Figure]

**Figure AC1.b:** Left panel shows South Asia domain of the weather@home regional climate model and right panel shows the four sub-regions of Bangladesh with APHRODITE based mean monsoon rainfall (mm/day).

[Figure]

Figure AC1.c: Return period plots for JJAS daily precipitation over sub-region 2 in different years (2006-2015).

[Figure]

**Figure AC1.d:** The left column shows the old version of the annual cycles of 5-day rainfall over the four sub-regions of Bangladesh. The right column shows the same but uses updated APHRODITE data.

We thank for the constructive comments from the anonymous Reviewer 2. We have carefully revised the manuscript to incorporate necessary amendments as per suggestions. Responses to the Referee Comments 2 (RC2) are presented in the following Author's Comments 2 (AC2):

5  *The authors investigate changes in mean and extreme precipitation in Bangladesh for five different forcing scenarios. They divide Bangladesh in four regions and analyse the magnitude of events with different return times. They find increases in pre-monsoon and monsoon precipitation due to higher global mean temperatures but also due to a decrease in aerosols. While the paper is clear in scope and the analysis in principle straightforward, it needs more work, especially the text. As crucial information is missing from the methods section I can only recommend*
10  *publication of the paper after major revisions.*

**General and Technical Comments**
**Text**

15  *RC2: The text could profit from more work. Some sections seem rather long; while others miss some essential information (see below). Also, there are numerous small mistakes that give the impression of sloppy proofreading. The conclusions, on the other hand are very well written and concise.*

AC2: Thank you for your careful review. We have revised the text as per suggestions and corrected the identified
20  errors in the text.

*RC2: It is very hard to really pin down, but I had the impression that some information is repeated over and over again, see e.g. my comment concerning the abbreviations below. Another example is the first two sentences in your introduction - they are different but they basically say the same. Of course, sometimes it is good to repeat things (e.g.*
25  *in the conclusions), I felt it rather hindered the flow while reading.*

AC2: We have aimed at avoiding repetitions in the manuscript as good as possible now.

*RC2: When you use the word significant do you mean 'statistically significant' or large? Significant is a reserved*
30  *word – please only use it if you conducted a statistical test!*

AC2: We mean statistically significant here. For all risk ratios, we also have calculated the associated error bars (using bootstrapping) to know whether the results are statistically significant or not.

35  *RC2: Similarly, you have to be careful when using the word 'risk'. Risk is often formalized as the combination of exposure and vulnerability to weather and climate events. Therefore please check if 'probability' of 'magnitude' would be more appropriate.*

AC2: We are aware that 'risk' is often defined as the product of vulnerability and exposure. However, in the

probabilistic event attribution, 'risk' is indicative of the range of hazards. According to UNFCCC, "Climate related risks are created by a range of hazards. Some are slow in their onset (such as changes in temperature and precipitation leading to droughts, or agricultural losses), while others happen more suddenly (such as tropical storms and floods)." [Source: https://unfccc.int/topics/resilience/resources/climate-related-risks-and-extreme-events]. In the abstract **(See page 1 line 13)**, to avoid any possibility of misinterpretation, we have added an explanation for the term 'risk'.

*RC2: You introduce abbreviations for the simulations, but you often refer to the simulations by the full name; e.g. on P6: L11 (twice); L15, L16, L21; L30 (twice). There are many more examples throughout the manuscript.*

AC2: Apologies for the irregularity. The model simulations are now uniformly denoted throughout the manuscript.

*RC2: What is the abbreviation for the simulation with current-day GHGs and pre-industrial aerosols? GHGonly? GHG? GHG only? AR? All of them are used throughout the manuscript. Be consistent! Also make sure that it can be differentiated from the abbreviation for greenhouse gas (GHG).*

AC2: Apologies again for such inconsistency. In this model ensemble, anthropogenic aerosols are reduced to pre-industrial levels, and so, the GHGs (at current concentrations) act as the main forcing. Hence this model ensemble consists of ACT GHGs with reduced anthropogenic aerosols (levels of the natural aerosols are unchanged). To keep this uniform throughout the manuscript, this model ensemble is now called 'GHG-only'.

**Methods**

*RC2: You should explicitly write why you use different time periods for your two observational datasets.*

AC2: At the time of analysing data for this study, APHRODITE was only available until 2007; while, CPC was available for the period of 2006-2015. Fortunately, this APHRODITE data is recently updated up till 2015. Therefore, we have updated our analysis with same period for all model and observation data sets.

*RC2: You use bi-linear interpolation to regrid your data. For the future I would recommend to use a conservative remapping scheme to make sure the precipitation amount is conserved.*

AC2: Thank you for the suggestion. We have checked ACT precipitation data over Bangladesh in this regard. As per figure AC2.a, we can argue that changing the method from bilinear interpolation to conservative has no effect on the high intensity precipitation events.

*RC2: The description of the model simulations is very long and overly detailed. I recommend to shorten it.*

AC2: We have tried to make this part shorter by avoiding repeating ensemble information.

*RC2: On the other hand, I miss a description of your statistical analysis, which makes it difficult to assess it. In particular I need the following questions answered:*

*RC2: How do you calculate the return time?*

AC2: "Return time" of an event, also known as the "return period" is the likelihood of an event occurring, defined by a particular variable exceeding a certain threshold during a given time interval. For a variable *X*, if the threshold level is $x_T$, then an extreme event occurs when $X \geq x_T$

10 Now, if *p* is the probability of occurrence of an extreme event, then Return Period *E (τ) = T = 1/p, or, p(X ≥ x_T) = 1/T*. For instance, a "1 in 10 year event" is an event with a 10% chance of occurring. On the contrary, the rarest event is a "1 in 1000 year event", with a 0.1% chance of occurring in a given year.

*RC2: How do you calculate the uncertainty of the return time?*

AC2: The uncertainty of the return period is calculated using bootstrapping. The time series is resampled a 1000 times and what we present are the 95% confidence intervals. We note that structural model uncertainty (such as parameter sensitivity) is not included in our uncertainty estimate.

20 We have explained the calculation method for Return Time and its Uncertainty in **page 6 lines 1-10** of the revised manuscript as: ""Return time" of an event, also known as the "return period" is the likelihood of an event occurring, defined by a particular variable exceeding a certain threshold during a given time interval. If variable X is equal to or greater than an event of magnitude $x_T$, occurs once in T years, then the probability of occurrence $P(X \geq x)$ in a given year of the variable is (Wilks, 2011):

$$P(X \geq x = \tfrac{1}{T}) \ \text{ or, } \ T = \tfrac{1}{1 - P(x \geq x_T)}$$

25 A "1 in 10 year event" is an event with a 10% chance of occurring. On the contrary, the rarest event is a "1 in 1000 year event", with a 0.1% chance of occurring in a given year. The rainfall amounts associated with the 50- or 100-year return periods are extracted from the 98[th] and 99[th] percentiles, respectively, of a fitted distribution (i.e., $[1 - 0.98^{-year}]^{-1} = 50$ years and $[1 - 0.99^{-year}]^{-1}$ =100 years) (Wilks, 1993). The uncertainty of the return period is calculated using bootstrapping method. The time series of each ensemble is resampled a 1000 times using bootstrapping to derive 5 to 95% confidence intervals for return periods. We note that
30 structural model uncertainty (such as parameter sensitivity) is not included in our uncertainty estimate."

*RC2: For the RR, do you assume an Extreme Value distribution? If not how is the probability calculated?*

AC2: In case of the model results, we have large enough an ensemble to calculate the probabilities of occurrence (P)
35 of the event in question explicitly by means of the different forcing scenarios. For example, suppose a 200 mm/day precipitation event has a $P_{actual}$ of 50 years and a $P_{natural}$ of 100 years. The resulting change of probability of that event would simply be a doubling (RR= 2) due to the change in forcing.

*RC2: How do you calculate the uncertainty of RR?*

AC2: To calculate the upper and lower limits of the uncertainty of RR we have used the following formula (based error propagation model for independent contributors):

Upper limit of RR uncertainty $= \sqrt{(a^2 + c^2)}$; and Lower limit of RR uncertainty $= \sqrt{(b^2 + d^2)}$

Where, a = upper limit of uncertainty of $P_{ACT}$,     b = lower limit of uncertainty of $P_{ACT}$,

c = upper limit of uncertainty of $P_{NAT}$,     d = lower limit of uncertainty of $P_{NAT}$.

We have added these responses in the revised manuscript in **page 7 line 1-9.**

*RC2: Do you consider all days or only days with rainfall larger a threshold? This is particularly relevant for MAM.*

AC2: We are considering all days while calculating the return periods. In this way, we can look at low intensity rainfall events with minimum return period of 1 year, and also high intensity rainfall events with up to 980 years return period. However, we focused on rainfall events with high return periods that are relevant for impacts and adaptation planning (e.g., 10-100 year events).

*RC2: Given that you have data from 98 * 10 years, should your maximum return time not be at 980 years instead of 1000 as in Figure 6?*

AC2: The maximum return period is at 980 years. The coloured dots in the figure end at that point but the scale ends at 1000 years. Due to very little difference between them (in comparison to the total length of the scale), it is not clearly visible. We have updated all return time plots with rainfall events with 10-100 year return period because these events are particularly relevant for impact assessments and adaptation planning.

*RC2: Similarly, when analysing the two wettest/ driest years should you not end at 196 years instead of 200?*

AC2: Yes, you are right. This ends at 196 years not 200 years.

*RC2: You might want to mention that you look at SPI in the methods.*

AC2: SPI calculation method is now added to the supplementary material of the manuscript.

**Results**

*RC2: The two observational datasets show quite some differences. Do you have a guess if this is due to the different periods they span, or would they also be different covering the same periods? How different are they during the two overlapping years?*

AC2: The discrepancy between the two observation data sets is due to different periods that they spanned. We have now used the same time period for both data sets because APHRODITE is recently updated and this data version

covered up till 2015. Figure AC2.b demonstrates the improvement in the results; by comparing the old and updated versions of the annual cycles. In the right panels we can see, that the two observation data sets are in better agreement. The model biases are also reduced as a consequence of comparison between the model and observation data spanning the same time period.

*RC2: What is the point of using the two lowest and highest years? It basically says 'during a wet year the magnitude of a 1 in X year's event is larger than during an average year', which is a relatively trivial result. Also, these results are barely used / described in the results sections. My proposition would be to either remove this entirely, or move it to the appendix. This would help to clean up the figures and/ or reduce the amount of required subplots.*

AC2: The use of two wettest and driest years explains how much natural variability of sea surface temperatures (SSTs) contributes to the variability of rainfall intensities over this study area. These are two pairs of individual years with the wettest and driest conditions amongst all model years of ACT ensemble. This model ensemble has same forcings in each year for the historical period of 2006-2015 and this means the only variability playing a role in
15 changing the intensity of rainfall is natural variability of SSTs. This indicates that for Bangladesh natural SST variability can play a role besides the anthropogenic forcings in some cases. This would not be the same case, for example, in a European country context, where natural variability of SSTs would be very small and so it will have minimal contribution compared to other forcings. We agree that this result is not adequately discussed in the manuscript and so we are adding this in the updated version.

See **page 9 lines 29-33** for the added information as: "Overall, the signal-to-noise ratio is higher across all sub-regions, during JJAS compared to that during MAM. During MAM, the highest and lowest signal-to-noise ratio is over sub-region 1 and 3, respectively (Figs. 6a & 7a). On the other hand, during JJAS, we find the highest and lowest signal-to-noise ratio is over sub-region 3 and 1, respectively (Figs. 9a & 8a). The lower the ratio, the more difficult it
25 is to establish causality as natural variability due to ENSO or circulation anomalies is higher."

*RC2: Evaluation is done with 5-day precipitation but return time plots and RR in the main text are with 1-day precipitation – why is that?*

30 AC2: For model evaluation, 5-day running mean precipitation based seasonal cycles only smooths out the variability of daily data based seasonal cycles (not shown) – the overall model evaluation result remains the same. For brevity, we only put the 5-day precipitation based annual cycles in the manuscript. For RRs, we have looked at both 1- and 5-day precipitation events considering their potential to cause flash floods or, landslides. We have presented 1-day precipitation based RRs in the main text because (i) this illustrates the highest variability of the daily precipitation
35 extremes that are probable due to different forcings (ii) daily extreme precipitation events can have sudden and severe impacts on society by affecting agriculture, transport, industry and ecosystem services. The return period plots for 5-day precipitation events are presented at the supplementary materials.

*RC2: You sometimes talk about a linear rainfall response. What do you mean with linear? How can you know it's linear (as you have only 5 data points)?*

AC2: By linear response, we meant steady and gradual increase in the climate change impact on rainfall from one forcing scenario to another due the warming effects starting from NAT to ACT, ACT to HAPPI 1.5 and HAPPI 1.5 to HAPPI 2.0. But we see a non-linear response during JJAS (in Fig. 3) which involves a drying effect from pre-industrial to present climate (Fig. 3a), followed by a large wetting effect from present-day climate to 1.5 degrees warmer world (Fig. 3b) but then again a small wetting effect from 1.5 to 2.0 degrees warmer worlds (Fig. 3c). If we consider the GHG-only scenario, we can explain this non-linear effect, as it demonstrates the extent of aerosol-related rainfall suppression (Fig. 3d). The "data points" are the result of hundreds of model simulations, i.e. they are robust as far as our weather@home model setup is concerned. They might differ amongst models, but since the other HAPPI models do not provide a GHG-only scenario, we can only infer causality of potential non-linear changes between the warming scenarios from weather@home.

We have explained the term linear response in the revised manuscript in **page 8 lines 7-9** by adding the lines**: "By** linear response, we meant steady and gradual increase in the climate change impact on rainfall from one forcing scenario to another due the warming effects starting from NAT to ACT, ACT to HAPPI 1.5 and HAPPI 1.5 to HAPPI 2.0."

**RC2 General Comments on Figures**

*RC2: I had to rasterize the Figures in order to print them. Not sure what the problem is but - please make sure this does not happen for the final paper. However, even in your original pdf the figures are all blurry and don't have a good quality, this makes them very difficult to analyse. Please save them as \*.pdf and ensure fonts are embedded.*

AC2: Apologies for the difficulties that you have faced. Most of the figures are redone to have good quality with high resolution.

*RC2: Avoid mixing green and red in the figures.*

AC2: Thank you, this point is noted.

*RC2: The text (labels, legend) is generally too small.*

AC2: Labels, legends are now larger where possible.

*RC2: The captions are very long and describe results. Please make them shorter and move all results to the main text.*

AC2: The captions are made reasonably shorter by moving some texts to main results section.

*RC2: The naming of simulations is inconsistent in the legend/ labels. E.g. in Figure 1 you call it 'HadRM3P ACT' but 'HAPPI 2.0'. I recommend removing 'HadRM3P'. In Figure 2 and 3 it is called present instead of ACT. In Figure 4 you introduce a new abbreviation!*

5 AC2: Removed HadRM3P, its only ACT now. We have now used uniform names throughout the manuscript for the different forcing scenarios. ACT, NAT, GHG-only, HAPPI 1.5 and HAPPI 2.0 are used for present-day actual; pre-industrial natural; present-day GHG (with pre-industrial levels of anthropogenic aerosols); and additional global warming since pre-industrial period by 1.5 and 2.0 degrees, respectively.

10 *RC2: In the same category: sometimes it is called MAM and sometimes pre-monsoon*

AC2: Now they are kept uniform throughout the manuscript.

*RC2 Specific Comments on Figures*

15 *RC2: Figure 1: Annual -> seasonal*

AC2: Amended as per suggestion.

*RC2: Figures 2 and 3: I recommend to change the title to '(ACT - NAT) / NAT', '(HAPPI1.5 – ACT) / ACT', etc. so it*
20 *is absolutely clear what you are doing.*

AC2: Percent change (PC) is calculated as: $PC_{actual\ relative\ to\ natural} = \{(ACT-NAT)/ACT\} \times 100$. We have added a paragraph explaining how PC is calculated in the supplementary material. Therefore, we are not changing the figure caption.

*RC2: The maps in the top row look a bit distorted – do you use a projection for the map plot?*

AC2: The maps are produced again with high resolution and without any distortion. We show here one updated figure to show the improvements.

*RC2: Why is it only 'approximately' the sub-regions? Remove approximately.*
AC2: Removed

*Figure 6: 'sub-regions of 1 and 2' -> remove of*
35 AC2: Removed

*RC2: Figures 10 and 11: Please use a logarithmic y-axis so that the plot is symmetric with respect to 1. Remove the bars, the RR is not really something that starts a 0. Reverse the order of the bars, start with the 1 in 10 year event.*

AC2: We have now used a logarithmic y-axis. After using a logarithmic y-axis, the issue of starting from 0 is solved. The order of the bars is reversed starting from 10 in 100-year event. See the updated figures AC2. c, & d).

**RC2 Minor Comments**

*RC2: P1 L11: the future*

AC2: Amended

10 *RC2: P1 L15: risk -> probability*

AC2: We want to keep the term as it is. To avoid any possibility of misinterpretation, we have explained clearly what the term 'risk' means in this study.

15 *RC2: P1 L16: 2C global warming*

AC2: We want to keep it as it is because this is an opening statement and so it can be applicable for 1, 2, 3 or even 4 degrees warmer conditions. But we have only focused on the Paris Agreement temperature targets.

20 *RC2: P1 L20: risk -> probability or magnitude*

AC2: We want to keep the term as it is. To avoid any possibility of misinterpretation, we have now explained clearly what the term 'risk' means in this study.

25 *RC2: P1 L23: in terms : : : impact. -> remove, you do not look at impacts*

AC2: By impacts we meant climate change impacts on the probabilities of extreme rainfall events. We have modified the line as "Climate change impacts on the probabilities of extreme rainfall events are found during both pre-monsoon and monsoon seasons, but the level of impacts are spatially variable across the country."

*RC2: P1 L25: GHG abbreviation not introduced*

AC2: GHG abbreviation is now introduced.

35 *RC2: P2 L12: Is it really an increasing trend – do you mean a positive trend?*

AC2: We meant a positive trend. This line is now changed to "The frequencies of observed high-intensity rainfall events are increasing in the recent years."

*RC2: P2 L14: names -> name*

AC2: Corrected

5  *RC2: P2 L15: Currently the sentence reads as if the "low-lying areas" damaged the rice.*

AC2: Modified the line at **page 2 lines 22-23** as "Consequently, vast areas of Haors (local name for lowland wetlands) and low-lying areas were inundated and most of the nearly-harvestable 'Boro' paddy crop (a local high yielding variety of paddy) was damaged (Nirapad, 2017)."

*RC2: P2 L15: Should that be "Boro (...) and paddy crops"? Or is should it be "Boro paddy crops (...)"?*

AC2: It should be Boro paddy crop (….), see the above response.

15  *RC2: P2 L16: Which dataset is this?*

AC2: It is NASA's Integrated Multi-satellitE Retrievals for Global Precipitation Measurement, GPM (IMERG) data. We have now mentioned this in the manuscript at **page 2 lines 24-25.**

20  *RC2: P2 L25: remove 'multi'. Also I would recommend to rewrite this sentence.*

AC2: 'multi' is removed and the sentence is re-written as follows: "According to global climate model (GCM) ensemble based study, By 2090, the north-western part of Bangladesh would experience ~9% and ~18% increase in the pre-monsoon (Mar-May) and monsoon (Jun-Sep) mean rainfall respectively (Kumar et al., 2014)". See **page 2**
25  **lines 34-36.**

*RC2: P2 L25: for the northwestern part of*

AC2: This part is edited, see the above response.

*RC2: P2 L27: the high resolution*

AC2: Amended

35  *RC2: P2 L28: in the global*

AC2: Amended

*RC2: P2 L31: north-eastern*

AC2: Amended

*RC2: P2 L34: remove 'of'*

5  AC2: Removed 'parts of'

*RC2: P2 L35: Even if they did not calculate RRs other studies did look at the influence of anthropogenic climate change -> please reformulate*

10  AC2: By the influence of anthropogenic climate change, we meant the human impacts on climate change due to past GHG emissions (since pre-industrial period). This attribution experiment is done by comparing probabilities of occurrences of events crossing a predefined threshold in (i) present-day climate and (ii) a hypothetical natural climate with pre-industrial levels of GHGs in the atmosphere. This is certainly not done in Kumar et al., (2014), Caesar et al., (2015), and Nowreen et al., (2015). For clarity, we have changed the line as "...; explained whether or not
15  anthropogenic climate change played a role in changing the probabilities of those projected future rainfall events; …...." in **page 2, line 43 to page 3 line1**.

*RC2: P2 L41: runs -> simulations*

20  AC2: Amended as per suggestion.

*RC2: P2 L41: remove 'the'*

AC2: Removed

*RC2: P3 L4-L5: also : : : warning: I don't understand what you mean here.*

AC2: These observation data sets are also used in another paper of the authors. We have removed this line to avoid confusion.

*RC2: P3 L10: remove 'here'*

AC2: Removed

35  *RC2: P3 L11-L14: I recommend to move this sentence to the methods.*

AC2: Agreed and moved to method section.

*RC2: P3 L18: 2.2 -> 3.1*

AC2: Corrected, thank you for pointing this out.

AC2: Section 3.2 is now mentioned.

AC2: Removed

AC2: Removed, this line is rewritten as "APHRODITE is a high-resolution daily gridded rainfall data set for Asia (V1901, available for 1998-2015); created primarily with data obtained from a rain-gauge-observation network." See **page 3 lines 33-34.**

AC2: Corrected

AC2: Moved as per suggestion

AC2: Removed

AC2: Removed

AC2: It is 'GHG-only' not 'GHG'.

AC2: Moved to Result section

*RC2: P4 L10: remove 'the'*

AC2: Removed

5 *RC2: P4 L27: one third*

AC2: Amended as per suggestion.

*RC2: P4 L29: remove 'world'*

AC2: Removed

*RC2: P4 L31: hereinafter?*

15 AC2: Removed hereinafter

*RC2: P4: Formula (i) and (ii): These are unnecessary.*

AC2: Removed

*RC2: P4 L39: force -> forcing*

AC2: Amended as per suggestion.

25 *RC2: P4 L41: the other GCMs -> other GCMs*

AC2: Amended as per suggestion.

*RC2: P5 L5 and L6: I would write this as 3x30x10x98 and 4x30x10x98. In addition, make a remark that all months*
30 *have 30 days in the used model.*

AC2: Agreed and rewritten as per suggestion; see **page 6 lines 15-17**.

*RC2: P5 L15: presented -> present*

AC2: Amended as per suggestion.

*RC2: P5 L15: This sentence does not make sense, please rewrite.*

AC2: Rewritten as "In order to quantify changes in the probability of occurrence of extreme rainfall event, we use Risk Ratio (RR), which is calculated as RR = Pf / Pcf (NAS, 2016). Here Pf denotes the probability of the event in factual climate including climate change (ACT, HAPPI 1.5 and HAPPI 2.0) and Pcf denotes the probability of an event of the same magnitude in a counterfactual climate without anthropogenic climate change (NAT)." see **page 6 lines 36-39**.

*RC2: P5 L17: the probability*

AC2: Amended as per suggestion.

*RC2: P5 L17: remove 'scenarios'*

AC2: Removed

*RC2: P5 L18: an event of the same magnitude*

AC2: Amended as per suggestion.

*RC2: P5 L18-L20: Abbreviations!*

AC2: Rewritten

*RC2: P5 L26 and L34: annual -> seasonal*

AC2: Amended as per suggestion.

*RC2: P5 L35: dataset it is compared with and sub-regions. -> dataset and the sub-region.*

AC2: Amended as per suggestion.

*RC2: P5 L36: at -> in*

AC2: Amended as per suggestion.

*RC2: P5 L36: There is no way to know if the bias is the same for the different scenarios! This is an assumption. It's ok to make the assumption, but 'note' is not the appropriate word here.*

AC2: Removed the word 'note' and rewritten the sentence as "The bias is apparently present in all model scenarios; hence it is unlikely to affect the comparison between model scenarios". See **page 7 line 26**.

*RC2: P6 L2: Significant?*

AC2: Yes, 'significant' because the changes in seasonal mean rainfall are calculated from the result of hundreds of model simulations, i.e. they are robust as far as our weather@home model setup is concerned.

*RC2: P6 L6: although they suggest*

AC2: Amended as per suggestion.

*RC2: P6 L28-L29: Did you show this somewhere in your paper? Do you have a reference for this?*

AC2: We refer to Bollasina at al. (2011) and Zhao et al. (2019) for more theoretical background and model support for our conjecture (which we do not further analyse as it is beyond the scope of this paper) that circulatory changes may have caused the non-linear rainfall percent-change in northern India with warming.

*RC2: P6 L34-L35: Is that global or local warming?*

AC2: It is global, not local warming. We are looking at global warming effects on regional/local weather events.

*RC2: P6 41: Significant?*

AC2:  yes, it refers to 'significant impacts'.

*RC2: P7 L4: remove 'and hence efficiently masked'*

AC2: Removed

*RC2: P7 L6: SPI: abbreviation not introduced*

AC2: Now it is introduced. See **page 8 line 34-35**

*RC2: P7 L15: in all -> in almost all*
AC2: Amended as per suggestion

*RC2: P7 L36: frequencies of occurrence -> magnitude*

AC2: Amended as per suggestion

[Figure]

Figure AC2.a: Comparison between two interpolation methods applied for seasonal mean precipitation during JJAS and MAM over Bangladesh.

Old figure                                                    Updated figure

[Figure]

Figure AC2.b: Comparison between old and updated versions of annual cycles of 5-day precipitation over the four sub-regions of Bangladesh.

[Figure]

Figure AC2.c: Percentage change (PC) in the MAM seasonal mean rainfall between different forcing scenarios. The top row (panels a-d) shows the regional PC over central parts of the South Asia. a. ACT rainfall PC relative to NAT b. ACT rainfall PC relative to HAPPI 1.5°C c. HAPPI 1.5°C rainfall PC relative to HAPPI 2.0°C d. ACT rainfall PC relative to GHG-only. Bottom row (panels e-h) shows the PC in the same way but over Bangladesh. The four boxes (1-4) on top of the panel e represent the four sub-regions of Bangladesh.

[Figure]

Figure AC2.c: The risk ratios of four specific rainfall events with return periods of 10, 20, 50, and 100 years between ACT/NAT, HAPPI 1.5/NAT, HAPPI 2.0/NAT and GHG-only /ACT over the two northern sub-regions 1 and 2 during MAM (top two panels of a. & b.) and JJAS (bottom two panels of c. & d.). The error bars indicate the associated uncertainty range with 95% confidence level for individual event. Same as Figure AC2.c but for MAM and JJAS risk ratios over the two southern sub-regions 3 and 4.

*This study focuses on the investigation of changes in total and extreme precipitation in Bangladesh due to changes in greenhouse gas and aerosol concentrations. Large ensembles of regional climate simulations are used to represent regional dynamics and aerosol effects with sufficient detail and at the same time obtain statistical robust results also for extreme events with long return periods. In my opinion, the research resented in this manuscript is generally sound*

5 *and provides novel insights (although not outstandingly new/innovative) into future rainfall changes in a highly impact-relevant region. However, I think the presentation and discussion of the research needs a substantial revision before the manuscript could be published. In my view, the interpretation of the results is too superficial at several places throughout the manuscript. Furthermore, the language and wording are not always adequate. With respect to the second point, I list a few issues below, but this list is not complete, and actually the native speakers among the co-*

10 *authors should be able to fix this in a better way than I am.*

We thank for the constructive comments from the anonymous Reviewer 3. We have carefully revised the manuscript to incorporate necessary amendments as per suggestions. Responses to the Referee Comments 3 (RC3) are presented in the following Author's Comments 3 (AC3):

**RC3 Specific Comments**

*RC3: Page 1 Line 22: As this is a model study, I'd avoid the term "impacts were observed"*

20 AC3: Rewritten the line as "Climate change impacts on the probabilities of extreme rainfall events are found during both pre-monsoon and monsoon seasons, but the level of impacts are spatially variable across the country." See **page 1 lines 26-27**

*RC3: P 1 L 28: "specifically with respect to...": I was confused when reading this as I though the whole study would*

25 *focus on extreme events. It is not clear from the abstract that also seasonal mean rainfall is analyzed.*

AC3: To make it clear that we have also looked at climate change impacts on seasonal mean rainfall, we have now added the following line in abstract: "Both GHGs and anthropogenic aerosols influence changes in seasonal mean rainfall over this region." See **page 1 lines 23**

*RC3: P 2 L 16: Nirapad (2017) is not in the list of references*

AC3: The reference for Nirapad (2017) is now added to the Bibliography. See **page 15 lines 13-15**

35 *RC3: P 2 L 23: "help to provide...": wording issue*

AC3: Deleted 'to provide'

*RC3: P 3 L 12-14: I think these detailed information regarding the sub-regions would fit better in the methods section.*

AC3: Agreed and moved to method section. See **page 5 lines 27-30**

*RC3: P 3 L27: "observational" appears too often in this sentence*

AC3: Rewritten as "The daily observational data sets that are used as a comparison against model results include: (i) Asian Precipitation Highly Resolved Observational Data Integration Towards Evaluation of Water Resources (APHRODITE) (Yatagai et al., 2012) and (ii) NOAA's Climate Prediction Center (CPC) global 0.5° analysis (Chen et al., 2008a)." See **page 3 lines 31-33**

*RC3: P 3 L 35: This is my only purely methodological comment: I am a bit sceptical with respect to the usage of bi-linear interpolation, as this does not conserve the area-average rainfall amount and also biases the extremes compared to the original grid point values. I would ask the authors to at least test the sensitivity of their approach using a more appropriate conservative interpolation method (see, e.g., Chen and Knutson, 2008, doi:10.1175/2007JCLI1494.1).*

AC3: Thank you for the comment. We have checked our ACT precipitation data over Bangladesh in this regard. As per figure AC2.a, we can argue that changing the method from bilinear interpolation to conservative have no effect on the high intensity precipitation events.

*RC3: P 4: The fact that some experiments are mentioned twice (in the first paragraph and further below) leads to some repetitions.*

AC3: We have now aimed at avoiding repetitions in the manuscript as good as possible.

*RC3: P 4 L 16: I'd mention already here in which way the ensemble members differ from each other.*

AC3: The model ensemble members differ in their initial conditions. They are either slightly perturbed, or the atmospheric field to restart the next model run is slightly different (i.e. it originates from a different member that has been run earlier). In contrast to the scenarios, all forcing parameters are the same. As far as NAT, GHG-only and the HAPPI scenarios are concerned, we use 11 different delta SST pattern (prescribed SSTs to define the lower boundary conditions). Those 11 patterns correspond to the same forcing scenario in CMIP5 (where the delta SSTs are derived from), but they do show slightly different spatial SST anomalies and add therefore additional variability to the weather@home ensemble of the counterfactual and future model scenarios.

*RC3: P 4 L 33-34: This notation is awkward. I'd either write this in text form or as a "real" equation, but not mix these things up.*

AC3: Agreed, we have now removed the equations and only kept text to describe this model ensemble.

*RC3: P L 12-15: This description of the aerosol affect is too short and not very clear. The term "omitted aerosol induced rainfall" should be explained. I am also confused by the sign of the signal and the figure caption: The caption of Fig. 2 says that the figure shows present-day relative to GHG only; positive values would thus mean that the present-day rainfall including the aerosol effect is larger than the rainfall due to GHG only, which is not consistent.*

5    *Finally, before directly linking this result to potential future decreases in the aerosol effect already in the second sentence, the actual content of the figures should be described and explained.*

AC3: The positive values for percent change in MAM mean rainfall shown in Fig. 2d in present-day ACT relative to GHG-only indicates the additional rainfall that could happen in the present-day if only GHGs were the dominant

10   forcing and if anthropogenic aerosols (reduced to pre-industrial levels) were not effecting rainfall. But, instead of such additional rainfall, we see a drying effect in present-day ACT relative to NAT because existing aerosols over-compensate the GHG warming effects over this region.

*RC3: P 6 L 19-28: I think this whole discussion is too superficial. There are many speculations on how*

15   *thermodynamic and dynamic effects could influence the precipitation changes which, in my view, are speculative and should be based on a more quantitative and solid analysis. For instance, I am not sure how an "approximately linear" scaling is deduced from the data presented in this study. If this just refers to the differences between the 1.5 and 2.0 simulations, I could well imagine a case in which precipitation increases due to both increase in the atmospheric moisture content and in the monsoon circulation, and this increase is amplified in the 2.0 case, which*

20   *may also produce a linear change over these simulations. Moreover, is a linear scaling really what we expect thermodynamically? The Clausius-Clapeyron relation is non-linear.*

AC3: We are indeed speculating based on work by others (e.g. Bollasina et al. 2011). It is beyond the scope of this paper to investigate the dynamic response in detail. We are planning on doing that in a more advanced study with

25   additional model simulations from HAPPI, but for now all we do is to "indicate" or "suggest" that a combination of mechanisms might be at play. None of what we say is conclusive, but it provides a potential explanation as to which factors could be at play. We can simply delete this paragraph, yet we believe that this would severely affect the integrity/content of this section. We would therefore pledge to keep the gist of the paragraph. We have added a sentence clarifying that our statements are rather speculative. See **page 8 lines 9-10**.

*RC3: The conclusion that dynamic changes play a secondary role should be manifested in a quantitative way. Also the statement that the thermodynamic response "usually scales with 20-40% of Clausius Clapeyron" is vague and, as such, not comprehensible.*

35   AC3: We did reformulate in order to make a less strong conjecture as to what could be going on. We deleted the last statement (although we are not sure why it is not comprehensible).

*RC3: P 7 L 1-2: I cannot follow here: In the region with the strongest decrease in Fig. 3a, the aerosol effect is small.*

AC3: This sentence is now deleted to avoid confusion.

*RC3: P 7 L6: The abbreviation "SPI" has not been introduced.*

5 AC3: Added now. See **page 8 line 34-35**

*RC3: P 7 L 11-3: Again I cannot follow. For instance, the changes in 5b,d are similar to those in 4b,d*

AC3: This is rewritten as: "Changes in mean absolute rainfall are much more pronounced over sub-regions 1 and 2,
10 where both MAM and JJAS rainfall exhibit clear shifts from one forcing to another forcing scenario (Fig. 4). On the
other hand, over sub-regions 3 and 4, only JJAS rainfall exhibited a robust shift (Fig. 5 b & d)." See **page 8 line 40-42**

*RC3: P 7 L 14: It is difficult to assess the relative change at this point, as the figures show absolute values*

15 AC3: Not sure we can exactly follow their point. Fig. 2 and 3 (as well as S1 and S2 for SPI) show relative percent
changes. We now discuss absolute changes. Can the referee elaborate on what is meant with that comment? We would
highly appreciate that.

*RC3: P 7 L 16-17: This doesn't fit with the interpretation in the figure caption. In general, I find it difficult to shift*
*parts of the discussion to the figure captions.*

AC3: Figure caption amended as "…… aerosol impacts over both sub-regions 1 and 2 are larger in MAM dry season
than that in JJAS wet season." Part of the figure caption is now also discussed here (as opposed to the figure caption).

25 *RC3: P 7 L 18: lapse rate and stability changes are not different feedbacks*

AC3: We agree. Instead we have clarified the point and added effects on boundary layer turbulence. Rewritten as:
"Consequently, direct and indirect aerosol effects, accompanied by feedbacks such as reduced lapse rate, reduced
boundary layer turbulence, or a modified land-sea circulation, remain to be a potent driver for changing monsoonal
30 rainfall amounts." See **page 9 line 4-6**

*RC3: P 7 L 21: Again I don't understand what "linear" increase means here.*

AC3: By linear response, we meant steady and gradual increase in the climate change impact on rainfall from one
35 forcing scenario to another due the warming effects, for example, from ACT to HAPPI 1.5 and HAPPI 1.5 to HAPPI
2.0. In other words, a 'linear' response is when we impacts (e.g. drying, wettening, warming, cooling) continue as a
function of increased warming, i.e. scaling with global mean surface temperature. We have now added a line to
explain this in the revised manuscript as: "By linear response, we meant steady and gradual increase in the climate

change impact on rainfall from one forcing scenario to another due the warming effects starting from NAT to ACT, ACT to HAPPI 1.5 and HAPPI 1.5 to HAPPI 2.0." **See page 8 lines 7-9**

*RC3: Section 3.3, first paragraph: There is an imbalance between the amount of text/discussion and the number of figures. The reader is left alone with much of the material shown in Figs. 6-9. Either expand this discussion, or, if you think that the results are not that important, move parts of the figures to the supplement.*

AC3: This section is expanded to discuss results from figures 6-9.

*RC3: P 8 L 4: "appear to counter": I cannot see how you come to this conclusion.*

AC3: Rephrased: "Might partially" instead of "appear". See **page 9 lines 40**

*RC3: P 8 L 24: "to a lesser extent": Really? Aren't the relative changes larger for the extremes?*

AC3: Revised to: [are projected to increase seasonal mean and extreme rainfall probabilities during] "probabilities". See **page 10 lines 17-18**

*RC3: P 8 L 31: "we conclude that the drier subregions ...": I don't think this has been demonstrated. To show this, the masking effect has to be quantified. Furthermore, it is not clear to me which region and season you're referring to.*

AC3: This part was indeed very confusing. Please accept our apologies. We have revised this paragraph including a more quantitative statement. See **page 10 lines 25-30**

**RC3 Comments on Figures**

*RC3: **Fig. 1**: I think this figure is too busy. I cannot distinguish the different shadings and also the lines of the observations are hard to see.*

AC3: Figure 1 is redone with higher resolution and better visual clarity (see below).

*RC3: **Fig. 3**: shorten caption (as Fig. 2, but for the monsoon season)*

AC3: All figure captions are now reasonably shortened.

*RC3: **Fig. 4**: I cannot follow the interpretation in the caption. For instance, I don't see such large differences in the masking effect between the regions. More in general, it is hard for me to understand how the masking effect is quantified here.*

AC3: You are right; the masking effects vary only with wet and dry seasons. The figure interpretation is rewritten as: "The figure shows that aerosol impacts over both sub-regions 1 and 2 are larger in MAM dry season than that in JJAS wet season." We compare the NAT, ACT and GHG-only (green, gray and orange boxplots) results to quantify the aerosol masking effects for both sub-regions.

*RC3: **Fig. 10**: Again, the interpretation in the caption is unclear (e.g., which region and season are you referring to?)*

AC3: Rewritten the figure caption as follows: Figure AC3.d: Same as Figure AC3.c but for sub-region 3 and 4. During MAM over both sub-regions 3 and 4, aerosol effects suppress the mean rainfall change between NAT and ACT (i.e.,
10 ACT rainfall is lower than NAT). On the other hand, during JJAS over both sub-regions 3 and 4, with lesser aerosol masking effects, ACT has higher mean rainfall than NAT and GHG-only would have noticeably much higher mean rainfall.

**References:**

Bollasina, M.A., Ming, Y. and Ramaswamy, V.: Anthropogenic Aerosols and the Weakening of the South Asian Summer Monsoon, Science 334 (6055), 502-505, doi: 10.1126/science.1204994, 2011.

[Figure]

Figure AC3.a: Comparison between two interpolation methods applied for seasonal mean precipitation during JJAS and MAM over Bangladesh.

[revised manuscript text omitted]

---

## Author Response (AR2)

**Reply to comments to referee 1**

I thank the authors for the revised version of the manuscript. The "Results and Discussion" section and the figure quality has substantially improved. However, the Introduction and "Data and Method" parts are not yet appropriate. Currently, I can not recommend publication.

We wholeheartedly thank the reviewer for their measured and constructive criticism, which we are addressing in detail below.

But first, we owe the editorial team (who have gratefully reopened the review process after we have failed to stick to several deadlines) and the referee(s) an apology for the extremely long delay of the revision. The timing of the reception of the current round of reviewer comments (November 2019) coincided with the end of the PhD thesis of the main author. Ruksana not only had to defend her thesis, but leave the country, suddenly raise two kids on her own given the partner had to stay in the UK, and to get to grips with her new role as assistant professor in Bangladesh. And then, as we all know, came Covid, which complicated things even further. It has only been recently, that things have calmed down a bit, re-enabling Ruksana to start thinking about the paper revision.

As second author, I had to move institutions during Covid as well, along with a major shift in priorities. Therefore, on behalf of the whole author team, I hope you can accept our apology and would be willing to review this set of revisions. We thank everyone involved in advance!

Main Comment
============
* The text (mainly the Introduction and Data and Methods) really need to improve. There is a large number of mistakes and inconsistencies that can be corrected and improvements that can be made. Most of them are small but they add up. I ask all co-authors to work on the text such that it can reach publication-quality.

We have completely revised the manuscript, re-arranged a few paragraphs, deleted confusing formulations and added additional context where necessary. Especially the Data and Methods section has been overhauled extensively. We do admit that the previous version wasn't appropriately revised before submission, for which we apologise as well.

In the following, we address each of the major and minor comments individually. We have revised the text suggestions accordingly, unless the text has been deleted or reformulated altogether.

Major Comments
==============
* It would be very helpful to include a figure (or a table) summarising the global mean temperature and aerosol levels over South Asia for each experiment.

This is an interesting suggestion, but given we are using an RCM (HadRM3P), the global mean would be based on the driving AGCM (HadAM3P). This in itself would still be a useful information, except that ACT (historical) and NAT (counterfactual) RCM scenarios are both based on ACT HadAM3P simulations. The only difference is in the radiative forcing and the delta SSTs (based on CMIP5 (historical minus historicalNat) are different, plus sufficient spin-up so that the atmosphere can respond to the respective scenario. Those delta SST numbers are provided in the HAPPI paper for various AGCMs, but it is not the global mean, which essentially cannot be estimated w/o a dedicated HadAM3P NAT simulation. It is even more complicated with the aerosol levels, as we'd need to diagnose the regional forcing, which would require an entirely new analysis altogether. We could plot the aerosol concentrations, but that would be somewhat misleading as the resulting aerosol forcing patterns are usually very different. Hence we are afraid that this suggestion cannot be sensibly implemented. We hope this detailed explanation is good enough a justification.

* The aerosol levels are "0" (NAT, GHG-ONLY), "1/3" (H1.5, H2.0), and "1" (ACT). Thus, in all comparisons between experiments (except ACT - CO2-ONLY) there is a change in global mean temperature and aerosols. This makes it very hard to disentangle (in my head) the contributions of global mean temperature and aerosols. One way around this (assuming additivity of the responses) would be to conduct a linear regression: (mean) rainfall as a function of global mean temperature and aerosol levels for each season and region. (I think this could be very insightful but I also understand if you deem it beyond the scope of the paper).

As you rightly point out, the only change in ACT vs GHG only is the aerosol level. Therefore, to detect the changes associated with anthropogenic aerosols, it is reasonable to assume that the difference between those two scenarios can be interpreted as the (inverse) net aerosol effect. Both, the associated (regional) temperature and rainfall change would be attributable to aerosols and can be used to qualitatively disentangle the forcing contributions, as we have tried to demonstrate. Any regression analysis would have to take all contribution factors into account, i.e. it would have to be a multiple regression analysis, which is hampered by the same problems that we outlined in the previous reply. Again, it is an interesting suggestion, but at this point it would be beyond what we are able to include in this paper.

* Is my summary of your results correct?: "In general, higher global mean temperatures lead to higher rainfall and higher aerosols to lower rainfall, however, the relative importance of the two varies between the regions"? If so, it would be good to add such a sentence at the beginning of the results as it would help to understand the rest of the paper.

Yes, that is a fair assessment of the results. We have included this sentence in the introduction to put it as prominently as possible.

* I still don't find the notion of a "linear response" fitting. Linear can be defined as "arranged in or extending along a straight or nearly straight line". Given the change in aerosols you would not expect a linear response between the experiments (unless the aerosols do not play a role). I would say "monotonic" or maybe "gradual" would be more appropriate words here.

We agree. The same issue has been raised by the other referee. We have therefore changed the wording to 'monotonic' where ever possible in order to avoid contentious interpretations.

* You still mention results in the figure captions. Remove them.

Amended.

Minor comments
==============
* Is there a reason the regions are called "sub-regions"? I'd recommend just calling them "regions".

We never really thought about it to be honest. In some sense, we consider Bangladesh as our main region (within the larger South Asian regional model domain), with the four smaller regions within Bangladesh as sub-regions. So we decided to leave it for now and hope our argument is convincing.

* P4 L37-L42: "representing the current decade" has a strange ring to it, as it basically is a pre-industrial simulation. Can you reformulate.

We agree. It might have been close to a current decade when we set out to do the analysis, but given how much time has passed since, we have reformulated the text and use 'recent decade' instead.

* In Figure 2 and Figure 3 (also Section 2.3) you compare e.g. "ACT to NAT" - I had more trouble than necessary figuring out which way round this actually is - it would be so much easier if you wrote this as "ACT - NAT". (NOTE: in the supplementary you write this as "ACT relative to NAT", i.e. the other way round).

Valid point. We hope that the revised version of the manuscript is more consistent now. We always show the difference between ACT and NAT, i.e., how has rainfall changed under ACT relative to NAT conditions (with NAT being the baseline).

* You can delete Table S1, all information is contained in the text (once you mention the one missing resolution).

Amended.

* P11 L9-L22: This section belongs to the results.

Amended.

* The boxes indicating the regions in Figure 2 e got lost.

They are corrected now.

* The abbreviation SA (in the figure captions) is not defined.

Amended.

* P7 L29: I don't think this is a confidence interval. I would call that "range".
P7 L37: I am still not happy with this formulation: you do not know whether the bias is present in all scenarios - you assume so (and it is fine to assume this), but you do not know.

We have changed it to uncertainty range and added an additional paragraph to clarify the bias problem for different scenarios.

* Not all colors in the figure captions are correct.

Amended.

* You mix red and green in the figures, which is not colorblind-friendly. Can you make one of them magenta?

We tend to agree that the colour choice of the figures is not always perfect. While not purely rainbow, we would make different choices if we were to restart the whole analysis. Given that we would have to redo all the figures in order to make these small changes, we would hope that we get away with it this time around. A larger version of the figures will be available in the final version of the paper, which should help to increase readability. Also, we will make sure that colour-blind friendly plots are used exclusively in any future paper, presentation or other publication for that matter.

**Reply to comments to referee 2**

In this revision, Rimi and co-authors have improved their manuscript and properly accounted for most of my previous comments. I think the presentation of the results is now clearer, and the paper is almost ready for publication. I only have a few minor remarks for the authors to consider when preparing the final version of the manuscript.

We also thank the reviewer very much for their measured and constructive criticism, which we are addressing below line-by-line.

But as before, we first want to send our sincere apology to the referee(s) for the extremely long delay of the revision. The timing of the reception of the current round of reviewer comments (November 2019) coincided with the end of the PhD thesis of the main author. Ruksana not only had to defend her thesis, but leave the country, suddenly raise two kids on her own given the partner had to stay in the UK, and to get to grips with her new role as assistant professor in Bangladesh. And then, as we all know, came Covid, which complicated things even further. It has only been recently, that things have calmed down a bit, re-enabling Ruksana to start thinking about the paper revision.

As second author, I had to move institutions during Covid as well, along with a major shift in priorities. Therefore, on behalf of the whole author team, I hope you can accept our apology and would be willing to review this set of revisions. We thank everyone involved in advance!

Just as a note: the page/line numbering in the response document is not consistent with the revised paper (at least in my pdf). For instance, the term "might partially" (page 37 in the response document) appears on page 10, line 9 in my pdf, instead of page 9 line 40 as indicated by the authors.

We are not sure why the line numbering was not consistent. Again, our apologies for the inconvenience. Presumably, it is down to Word files acting differently at different Operating Systems upon opening them. I suppose, this could be avoided when switching to pdf format.

I appreciate the effort made by the authors to update the version of the APHRODITE dataset, which has a positive effect on the results, and to include an additional model intercomparison analysis. I just think that describing the results of the latter in the conclusions section is somewhat uncommon. You may consider moving this paragraph to the results section.

We agree. This paragraph should have been in the results section all along. It's incorporated there now.

Two reviewers commented on the use of the term "linear". The authors note that "By linear response, we meant steady and gradual increase…". However, this is not the original meaning of this term, which, in my opinion, leads to confusion (a linear response is more than just steady and gradual). I'd suggest to replace "linear" with "monotonic".

We have now used 'monotonic' where ever possible. Hope this makes our intentions clearer.

My comment on page 7, line 14 in the original manuscript referred to the text on Figs. 4/5 (not 2/3; "yet the relative change is smaller", page 9, line 11 in the revised version). Maybe it would improve the readability if the relative changes were specified more explicitly (add a few numbers in the main text).

We did add a few quantitative estimates which hopefully help to put the results into better perspective.

I'm still slightly confused by Fig. 2. Does panel a show NAT relative to ACT (as stated in the heading) or ACT relative to NAT (as stated in the caption)? In addition: Why do some of the patterns seem to differ between upper and lower row (e.g., panel a vs. e in the southern part)?

Valid point (see above). We hope that the revised version of the manuscript is more consistent now. We always show the difference between ACT and NAT, i.e. how has rainfall changed under ACT

relative to NAT conditions (with NAT being the baseline). But more crucially, you have indeed spotted a problem in Fig 2g. It was inconsistent with Fig 2c, which has of course been amended. All other figures may appear differently, but it is in fact due to the differing range of values. Perhaps not the ideal choice, but the idea was to highlight the details in the smaller region of Bangladesh, which would otherwise overload the figures for the whole South Asia domain in the upper row of Figs 2 and 3.

---

## Author Response (AR3)

I thank the authors for the third version of the manuscript and the open discussion of the reasons for the delay. I am looking forward to seeing the article published! I do unfortunately still have some minor comments on the text but hope they are straightforward to fix.

We are genuinely thankful to reviewer 1 for providing with the constructive comments. We have amended the manuscript following the corrections given here. Hence, we hope that this manuscript version5 will be up to the mark for its final publication.

Minor comment
==============

* It's unfortunate that you cannot include the table summarizing the global mean temperature and aerosol levels of the experiments but I can accept your explanation. However, I would encourage you to mention the aerosol levels for the 1.5°C and 2.0°C experiments if possible in the experiment description on page 4 (e.g. mention that they are about one third of the ones in ACT, or "between ACT and NAT").

We have added this information at page 4 line 31.

* You mention (P5 L6): "The 'return time' of an event, also referred to as the 'return period' is the likelihood of an event occurring during a given period of time." -> The return time of a "1 in 10 year event" is "10 years" so the return time is the inverse of the likelihood.

We have excluded the phrase "also referred to as the 'return period'" to avoid any confusion.

* There are many mentions of something happening "at region X" - I think it's "in region X" (but I am not a native speaker).

We have changed from 'at sub-region' to 'in sub-region' in 4 instances that were found in the manuscript.

* Can I suggest you add % signs for ranges (5-95% -> 5%-95%) and the "E" and "N" all latitudes and longitudes.

We can confirm that "E" and "N" are present in the figures to indicate all latitudes and longitudes. Because it is a range of percentage from 5 to 95, we kept it as it is writtend in the manuscript.

Text
====
P1 L3: increase risk -> increase the risk
P2 L19: in the recent -> in recent
P2 L18: In June 2017, at southeastern parts of Bangladesh heavy rainfall caused devastating -> In June 2017, heavy rainfall in southeastern parts of Bangladesh caused devastating (I think)
P2 L27: flood floods -> floods
P2 L36: to global -> to a global

P2 L37: rainfall respectively -> rainfall, respectively
P2 L39: in the very -> in very
P2 L40: during monsoon season -> during the monsoon season
P2 L40: According to -> According to the
P2 L41: the north-eastern -> north-eastern
P3 L1: part or whole -> part or the whole of
P3 L2: RCM to -> RCM simulations to
P3 L8: to aerosols pollution -> to aerosol pollution
P3 L10: 1.5 -> 1.5°C
P3 L10: 'Paris Agreement' -> Paris Agreement
P3 L11: runs -> simulations
P3 L14: for a counterfactual -> for counterfactual
P3 L18: 'Paris Agreement' -> Paris Agreement
P5 L6: black -> red
P5 L20: Please ensure the $^-1$ is properly formatted for the final paper.
P5 L26: spanning for 30 days -> spanning 30 days (?)
P5 L36: additional four -> four additional (?)
P5 L39: at Supplementary -> in Supplementary
P5 L41: event -> events
P6 L2: In case -> In the case
P6 L11: under five -> under all five
P6 L11: Switch order of sentence and mention the obs products first?
P6 L13: for ACT -> for the ACT
P6 L17: sub-regions -> sub region
P6 L19: over Indian -> over the Indian
P7 L40: 1-2mm -> 1mm - 2 mm
P8 L12: over sea -> over the sea
P8 L30: as -> because
P8 L39: that noticeable -> that a noticeable
P8 L39: are -> is
P9 L5: 2,the -> 2, the
P9 L39: 1 and 2 shows -> 1 and 2 show
P10 L5: aerosols -> aerosol
P10 L20: point -> points

Figure 1. L5: APHRODITE (dark purple) -> APHRODITE (light purple)
Figure 2. L3: of the South Asia -> of South Asia
Figure 4: L5: y-scale range -> y-scale

We can confirm that all text corrections as well as figure caption corrections are done accordingly.